EMBO
Molecular Medicine

# CRISPR screens identify tumor-promoting genes conferring melanoma cell plasticity and resistance

Arthur Gautron[1], Laura Bachelot[1], Marc Aubry[1,2,†], Delphine Leclerc[3,†], Anaïs M Quéméner[1,†], Sébastien Corre[1,†], Florian Rambow[4,5], Anaïs Paris[1], Nina Tardif[1], Héloïse M Leclair[1], Oskar Marin-Bejar[4,5], Cédric Coulouarn[3], Jean-Christophe Marine[4,5], Marie-Dominique Galibert[1,6,*] & David Gilot[1,‡,**]

## Abstract

Most genetic alterations that drive melanoma development and resistance to targeted therapy have been uncovered. In contrast, and despite their increasingly recognized contribution, little is known about the non-genetic mechanisms that drive these processes. Here, we performed *in vivo* gain-of-function CRISPR screens and identified SMAD3, BIRC3, and SLC9A5 as key actors of BRAFi resistance. We show that their expression levels increase during acquisition of BRAFi resistance and remain high in persister cells and during relapse. The upregulation of the SMAD3 transcriptional activity (SMAD3-signature) promotes a mesenchymal-like phenotype and BRAFi resistance by acting as an upstream transcriptional regulator of potent BRAFi-resistance genes such as EGFR and AXL. This SMAD3-signature predicts resistance to both current melanoma therapies in different cohorts. Critically, chemical inhibition of SMAD3 may constitute amenable target for melanoma since it efficiently abrogates persister cells survival. Interestingly, decrease of SMAD3 activity can also be reached by inhibiting the Aryl hydrocarbon Receptor (AhR), another druggable transcription factor governing SMAD3 expression level. Our work highlights novel drug vulnerabilities that can be exploited to develop long-lasting antimelanoma therapies.

**Keywords** Aryl hydrocarbon Receptor; CRISPR-SAM; melanoma; SMAD3; targeted therapy resistance
**Subject Categories** Cancer; Skin

## Introduction

Identifying molecular cancer drivers is critical for precision oncology. Last year, the cancer genome atlas (TCGA) identified 299 driver genes by focusing on point mutations and small indels across 33 cancer types (Bailey *et al*, 2018). It represents the most comprehensive effort thus far to identify cancer driver mutations. Complementary studies are required to elucidate the role of copy-number variations, genomic fusions, and methylation events in the 33 TCGA projects.

Moreover, there is increasing evidence that non-genetic reprogramming leading to cancer cell dedifferentiation, stemness, invasiveness also contribute to tumor growth and therapy resistance (Puisieux *et al*, 2014; Bai *et al*, 2019). Thus, deciphering the signaling pathways that drive such processes may also lead to innovative cancer therapies. Recent gene expression quantifications performed at single-cell level by single-cell RNA sequencing (scRNA-Seq) demonstrated that cancer cells operate a dedifferentiation process, for instance in glioblastoma and melanoma (Patel *et al*, 2014; Tirosh et al, 2016; Rambow *et al*, 2018), promoting tumor growth, stemness, and therapy resistance. This "onco-dedifferentiation" seems to be independent of *de novo* mutations and could offer new targets/strategies to cure cancer. However, these scRNA-Seq studies are mainly descriptive; the tumor growth capability of each gene/RNA is not yet investigated at the genome-scale. Such functional analyses are nowadays feasible using clustered regularly interspaced short palindromic repeats (CRISPR)-Cas9 screens (Shalem *et al*, 2015). The majority of the CRISPR-Cas9 screens is based on the invalidation of coding genes but modulation of gene expression is reachable with the CRISPR-Cas9 synergistic activation mediator (SAM) approach (Konermann *et al*, 2015). It corresponds to an engineered protein complex for the transcriptional activation of endogenous genes. Importantly, SAM can further be combined with a human genome-wide library to activate all

1 CNRS, IGDR (Institut de génétique et développement de Rennes)-UMR 6290, Univ Rennes, Rennes, France
2 Plateforme GEH, CNRS, Inserm, BIOSIT - UMS 3480, US_S 018, Univ Rennes, Rennes, France
3 INSERM U1242, Centre Eugène Marquis, Rennes, France
4 Department of Oncology, KU Leuven, Leuven, Belgium
5 VIB Center for Cancer Biology, VIB, Leuven, Belgium
6 Service de Génétique Moléculaire et Génomique, CHU Rennes, Rennes, France
*Corresponding author. Tel: +33 0 223234705; E-mail: mgaliber@univ-rennes1.fr
**Corresponding author (Lead Contact). Tel: +33 0 223234441; E-mail: david.gilot@univ-rennes1.fr
†These authors contributed equally to this work
‡Present address: INSERM U1242, Centre Eugène Marquis, Rennes, France

known coding isoforms from the RefSeq database (23,430 isoforms) for gain-of-function screening without *a priori*. To date, CRISPR screens are mainly performed *in vitro* using cell lines or primary cultures (Meyers *et al*, 2017). A pan-cancer CRISPR-Cas9 knockout screen was performed *in vitro* (324 human cancer cell lines from 30 cancer types) to identify essential genes for cancer cell fitness (defined as genes required for cell growth or viability) and to prioritize candidates for cancer therapeutics (Behan *et al*, 2019). However, because the contribution of the tumor environment in tumor growth is increasingly recognized, it seems important to perform such screens in the relevant patho-physiological context and, for instance, take advantage of animal models.

We selected cutaneous melanoma as a paradigm since novel therapeutic strategies are critically needed (Bai *et al*, 2019). Targeted therapies such as BRAF inhibitors (BRAFi) initially showed great promise in patients with BRAF(V600)-mutated metastatic melanoma. Unfortunately, the vast majority of patients that initially respond to these drugs, almost inevitably develop resistance. Although combination therapies (BRAF and MEK inhibitors) enhance the response and delay relapse, the overall survival remains unsatisfactory highlighting the need of new therapeutic targets (Ascierto *et al*, 2016).

The mechanisms underlying resistance are numerous and probably not mutually exclusive (Sullivan & Flaherty, 2013; Welsh *et al*, 2016; Song *et al*, 2017). Resistance can be driven by a small preexisting subpopulation, harboring-specific genetic alterations that confer them with resistance to the inhibitors (Wagle *et al*, 2011). Such alterations may also occur *de novo*, during treatment (Welsh *et al*, 2016). In addition, there is increased evidence that non-genetic reprogramming may confer drug-tolerant and/or resistant phenotypes to melanoma cells (Rambow *et al*, 2018; Corre *et al*, 2018; Tsoi *et al*, 2018; Hugo *et al*, 2015a; Talebi *et al*, 2018; Rapino *et al*, 2018; preprint: Marin-Bejar et al, 2020). Earlier works demonstrated that phenotype switching from a proliferative to an invasive/mesenchymal-like state is also likely to contribute to therapy resistance (Hoek & Goding, 2010; Konieczkowski *et al*, 2014; Müller *et al*, 2014; Verfaillie *et al*, 2015; Boshuizen *et al*, 2018). Paradoxically, MITF-induced differentiation into a slow cycling, pigment-producing state was also reported to confer tolerance to BRAFi (Müller *et al*, 2014; Smith *et al*, 2016). It therefore seems that various drug-tolerant subpopulations can emerge under therapeutic pressure and that these cells can provide a pool from which resistance develops. Targeting these populations of persister cells is therefore crucial to achieve effective personalized therapies (Nassar & Blanpain, 2016).

Here, we performed unbiased screens to identify genes promoting tumor growth from persister cells and conferring resistance to BRAFi using CRISPR-Cas9 SAM methodology. We demonstrate that, in addition to promote melanoma development, Mothers against decapentaplegic homolog 3 (*SMAD3*), Baculoviral IAP repeat-containing protein 3 (*BIRC3*), and Sodium/hydrogen exchanger 5 (*SLC9A5*) also support relapse since they promote both BRAFi-resistance and tumor growth capability of persister cells. Their expression levels correlated with BRAFi resistance and relapse. Consequently, their inhibition strongly reduced the number of persister cells. Moreover, we demonstrate that the transcription factor AhR governs *SMAD3* expression levels and in turn SMAD3 drives the expression of a set of genes associated with BRAFi resistance and mesenchymal phenotype. These experiments identify integrated AhR-SMAD3 signaling as a key driver of melanoma growth and relapse, pointing to a new therapeutic vulnerability in melanoma.

# Results

## Identification of tumor-promoting genes by *in vivo* gain-of-function CRISPR screen

Since the tumor environment influences, at least in part, the tumor growth capability of cancer cells, we performed an *in vivo* genome-wide CRISPR-Cas9 SAM screen to identify *in vivo* tumor-promoting genes, defined as genes whose expression support tumor growth (in contrast to driver genes bearing a driver mutation such as BRAF (V600E)). To select the most appropriate cellular model, we classified melanoma biopsies from The Cancer Genome Atlas (TCGA) cohort ($n = 458$) in function of differentiation states according to the most recent melanoma profiling data (Fig 1A; Tsoi *et al*, 2018). As anticipated, the vast majority of these tumors, which are almost all drug naive, exhibited a differentiated profile (89%; melanocytic and transitory). We selected the 501Mel cell line since (i) these cells display a melanocytic differentiation state as the majority of diagnosticated melanoma, (ii) they harbor the BRAF(V600E) mutation as ~50% of cutaneous melanoma, (iii) they are highly sensitive to BRAFi with an $IC_{50}$ value of 0.45 μM to vemurafenib [PLX4032] (Halaban *et al*, 2010; Corre *et al*, 2018), and importantly (iv) they are unable to generate tumor in nude mice (Ohanna *et al*, 2011). This latter characteristic may allow to identify tumor-promoting genes.

To generate the CRISPR-SAM cell library, we modified the 501Mel cells, to express constitutively defective-Cas9 and the required cofactors for CRISPR-SAM technology (Konermann *et al*, 2015). These engineered cells were infected with the single-guide RNA (sgRNA) lentivirus library that contained at least three different guides per coding gene (Konermann *et al*, 2015; Fig 1B). The infection was performed at a multiplicity of infection (MOI) of 0.2 ensuring that only one guide is expressed per infected cell. Infected cells were positively selected using antibiotic selection during 7 days. By DNA sequencing, we observed a normal distribution of the sgRNAs in two cell library replicates (Fig 1C). Only 78 sgRNAs were not detected in our cell library, which validate our protocol and the cell library (>70,100 sgRNAs were detected; Table EV1). Thus, the controls were proper to identify *in vivo* tumor-promoting genes.

The cell library ($30 \times 10^6$ cells) was fractionated and subcutaneously xenografted in 10 nude mice ($3 \times 10^6$ cells/mouse) and tumor growth was monitored using caliper over a 5-month period (Fig 1D and Table EV2). As previously demonstrated (Ohanna *et al*, 2011), we confirmed that parental 501Mel are unable to form tumors in nude mice ($n = 6$). In contrast, seven tumors were obtained from the CRISPR-engineered cells xenografted in 10 mice (Fig 1D). The nature of the sgRNAs, their abundance and occurrence across these 7 tumors were determined by DNA-Seq (Fig 1E and F, and Table EV3).

By comparing the most represented genes (sgRNAs) in each tumor (Tum), we identified 3 common genes (Fig 1F). An enrichment of *SMAD3*, *BIRC3*, and *SLC9A5* sgRNAs was found in tumors when compared to their starting abundance (cell library; orange points; Fig 1G), in contrast to EGFR sgRNAs. Thirty-six other genes were recurrently retrieved in the tumors but not in all (Table EV3). Interestingly, *YAP1* which has already been identified as melanoma growth-promoting gene was also found (Table EV3). This supports

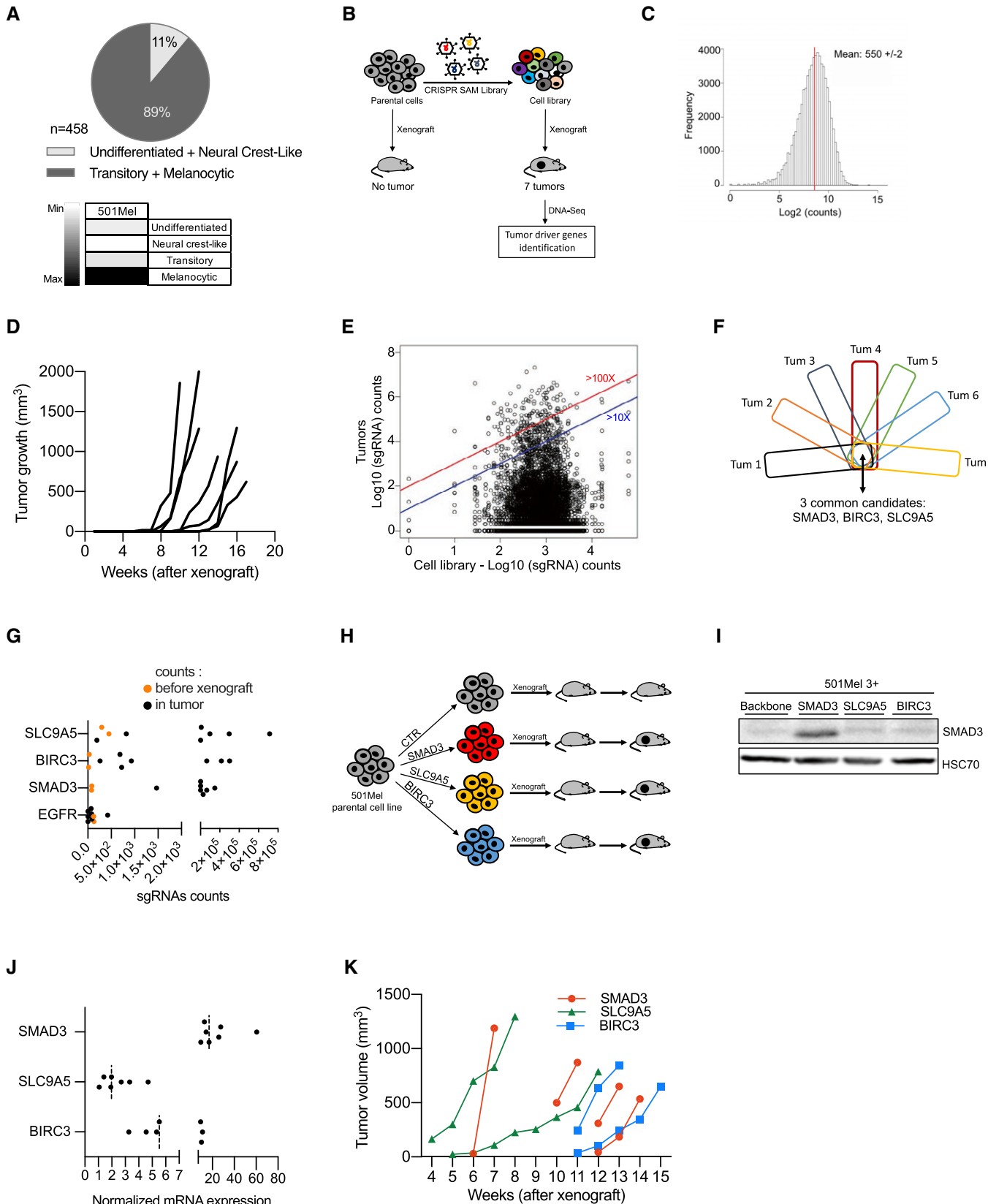

**Figure 1.**

◀

**Figure 1.  Identification of tumor-promoting genes by *in vivo* gain-of-function CRISPR screen.**

A   Determination of differentiation states of skin cutaneous melanoma (SKCM) biopsies from the TCGA cohort ($n = 458$) according to (Tsoi *et al*, 2018). The vast majority of these tumors exhibited a differentiated profile (89%; melanocytic and transitory states). The others display a dedifferentiated profile (11%; neural crest-like cells and undifferentiated states). Human melanoma 501Mel cell line is classified as differentiated melanoma cells (melanocytic cells), according to the gene expression profile (four categories were defined by (Tsoi *et al*, 2018); melanocytic, transitory, neural crest-like, and undifferentiated cells). 501mel cell line was selected for the CRISPR screens.

B   Workflow depicting the *in vivo* CRISPR-SAM screen to identify tumor-promoting genes. Parental cells and CRISPR-SAM cell library were xenografted on nude mice ($3 \times 10^6$ cells/mouse, $n = 6$ and $n = 10$, respectively) and tumor growth was monitored during 5 months. Seven tumors were collected and analyzed by DNA-Seq to identify the sgRNAs (Tables EV1 and EV2).

C   sgRNAs distribution in the cell library. sgRNA mean reaches $550 \pm 2$.

D   Tumor growth curves for the 7 tumors arising from the CRISPR-SAM-engineered cells xenografted in nude mice as detailed in panel B. (No tumor for the parental cell line (501Mel cells)) (Table EV2).

E   Distribution of sgRNAs in cell library and in the 7 tumors ($\log_{10}$(sgRNAs counts)). Blue and red lines indicated the enrichment $\geq 10$ fold or $\geq 100$ fold (tumors *vs* cell library (*in vitro*)) (Table EV1).

F   From the seven tumors, the top hundred genes (enriched) have been selected and common genes are *SMAD3*, *BIRC3*, and *SLC9A5* (Table EV3).

G   sgRNA counts in tumors *versus* in CRISPR-SAM cell library (respectively, black and orange points). Each black point corresponds to one tumor. Two replicates have been shown for CRISPR-SAM cell library (orange points).

H   Workflow depicting the validation step: 501Mel cells overexpressing *SMAD3*, *BIRC3*, or *SLC9A5* (obtained by CRISPR-SAM) were xenografted on nude mice and tumor volume was monitored using caliper. $3 \times 10^6$ cells/mouse. $n = 7, 6,$ and 6 mice, respectively.

I   SMAD3 expression levels in 501Mel cells overexpressing the cofactors for CRISPR-SAM approach and the control sgRNA (501Mel 3 + backbone) or the *SMAD3* sgRNA. *SLC9A5* and *BIRC3* sgRNAs are used as controls to show the specificity of the SMAD3 overexpression. HSC70 serve as loading control.

J   *SMAD3, BIRC3,* and *SLC9A5* mRNA expression levels in melanoma cell lines described in H and I. $n = 7$ independent biological experiments. Dashed lines for medians.

K   Tumor growth curves from 501Mel cells overexpressing *SMAD3*, *BIRC3*, or *SLC9A5*.

Data information: Western blot results are representative of at least two independent experiments. Source data and unprocessed original blots are available in Appendix Fig S1 source data. See also Fig EV1.

Source data are available online for this figure.

the robustness of the screen (Lamar *et al,* 2012; Verfaillie *et al,* 2015; Hugo *et al,* 2015b). The majority of the tumor-promoting genes (Table EV4) identified here are not considered as genes required for cell growth or viability (except the essential genes *YAP1*, *SLC25A41,* and *TGIF1*; Behan *et al,* 2019) and are not frequently altered in melanoma (Fig EV1A). Moreover, the transforming growth factor (TGF)-β pathway seemed well-represented among the tumor-promoting genes (Table EV4). Since our results suggest that a high expression level of these tumor-promoting genes is sufficient to promote melanoma tumor growth, we evaluated the biological consequences of the inhibition of one tumor-promoting gene, *BIRC3*. The chemical inhibitor Birinapant reduced the SKMel28R and Me1402 cell density (Fig EV1B and C), suggesting that the BIRC3 protein is involved in cell proliferation of these melanoma cells as previously demonstrated (Krepler *et al,* 2013; Vetma *et al,* 2017).

Next, we examined *in vivo* the ability of these tumor-promoting genes to promote tumor growth by xenografting cell populations overexpressing the sgRNAs individually. We focused on the top 3 tumor-promoting genes: *SMAD3, BIRC3,* and *SCL9A5* (Fig 1H–K). We generated three new CRISPR-engineered cell lines, and we evaluated the overexpression levels of these genes by Western blot experiments (SMAD3) and RT–qPCR (*SMAD3, BIRC3,* and *SLC9A5*) (Fig 1I and J). Finally, we confirmed that they independently foster tumor development (Fig 1K). Altogether, our results demonstrated that *in vivo* CRISPR-SAM screen identifies new tumor-promoting genes, which may constitute amenable target for melanoma.

### Genome-wide CRISPR activation screen identifies BRAFi-resistance genes in cutaneous melanoma

BRAFi provokes tumor shrinkage in the vast majority of patients with BRAF(V600)-mutated metastatic melanoma but resistance almost inevitably occurs (Bai *et al,* 2019). To examine the genes promoting BRAFi resistance and relapse, we performed an *in cellulo* screen using the same cell library in the presence of BRAFi (Fig 2). Briefly, the CRISPR-SAM 501Mel cell library ($40 \times 10^6$ cells) was treated for 14 days with BRAFi (2 µM), using either the BRAFi used in clinical practice (vemurafenib), the next-generation inhibitor that is still under investigation in clinical trials (PLX8394), or the solvent (dimethyl sulfoxide (DMSO)) as control (Fig 2A). This procedure allows for the enrichment of sgRNAs (genes) conferring resistance. The nature of the sgRNA present in the resistant population and their abundance was determined by DNA-Seq (Fig 2B). The best hit was the Epidermal growth factor receptor gene (*EGFR*), a well-known BRAFi-resistance gene (Sun *et al,* 2014; Shaffer *et al,* 2017). By examining the enrichment of sgRNAs targeting *EGFR* promoter (Fig 2B), we decided to retain genes with at least two sgRNAs among the enriched sgRNAs present in BRAFi-exposed cells (with a false discovery rate, FDR < 0.05) (Tables EV5 and EV6, Appendix Fig S1) since sometimes one of the three sgRNAs designed per gene is not detected or not enriched as observed for *EGFR* and *BIRC3* (Fig 2C). A recent publication confirmed that sgRNAs are not all functional in CRISPRa libraries and it could be interesting to increase the number of sgRNAs per target and to cover more TSS per gene (Sanson *et al,* 2018).

Apart sgRNAs targeting the *EGFR* promoter, the sgRNAs enrichment in BRAFi-exposed cells was unexpectedly weak (Tables EV5 and EV6). Thus, to select the best BRAFi-resistance genes, we examined the gene expression levels of these potential BRAFi-resistance genes identified by CRISPR screen in 12 melanoma cell lines (Fig 2D and E, Appendix Fig S1). We postulated that BRAFi-resistance genes are highly expressed in BRAFi-resistant cells ($n = 6$) as already demonstrated by other approaches for *NRP1*, *AXL*, and *NGFR*. To this end, we confronted CRISPR-SAM candidates (identified in 501Mel cells) to gene expression data from six melanoma cell lines

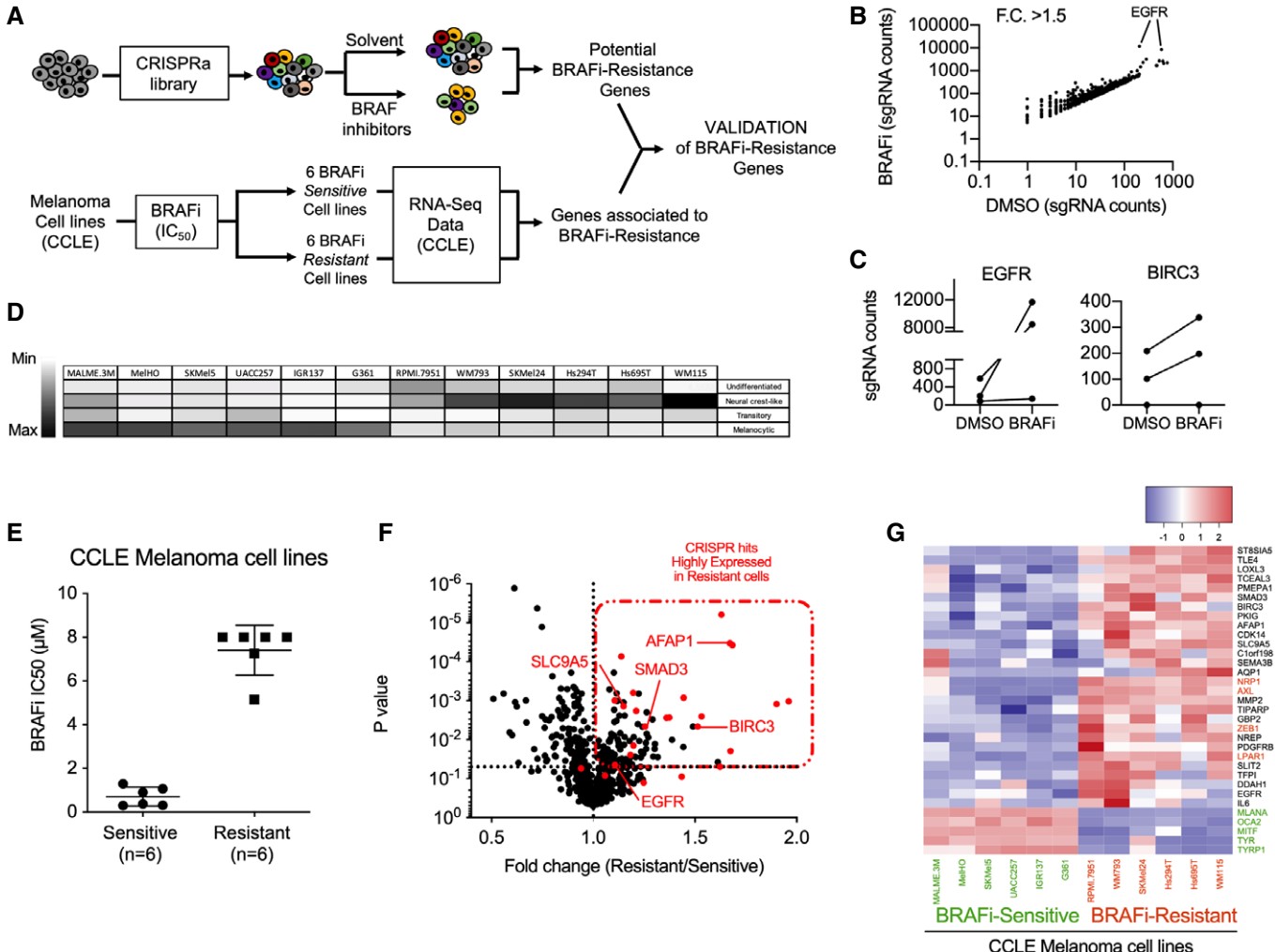

**Figure 2. Genome-wide CRISPR activation screen identifies BRAFi-resistance genes in cutaneous melanoma.**

A CRISPR-SAM workflow. Cell library was exposed to DMSO (solvent of BRAFi) or BRAFi 2 μM (vemurafenib (Vem) or PLX8394 (PB, Paradox Breaker)) during 14 days. $40 \times 10^6$ cells per arm. Experiments were done in duplicate.

B Plot showing sgRNAs detected in BRAFi-resistant cells *versus* in control cells (cell library exposed to DMSO) with a fold change > 1.5. Raw data are available in Table EV5 (BRAFi for PBV). Appendix Fig S1 depicted the comparison Vem vs. PB.

C sgRNAs counts in BRAFi-resistant cells *versus* in DMSO-exposed cells for *EGFR* and *BIRC3* (BRAFi for PBV, Appendix Fig S1)

D Determination of differentiation states of 12 human BRAF(V600) melanoma cell lines from CCLE according to (Tsoi *et al*, 2018).

E BRAF(V600) melanoma cell lines from CCLE were distributed in two groups (Sensitive and "intrinsically" Resistant, $n = 6$ cell lines per group) according to their BRAFi IC$_{50}$ (μM, half maximal inhibitory concentrations) (Barretina *et al*, 2012). Error bars reflect mean ± s.d.

F Volcano plot showing the expression levels of BRAFi-resistant genes (identified in 501Mel by CRISPR-SAM screen) in 12 human melanoma cell lines. The fold change corresponds to the ratio of expression levels found in resistant and sensitive cell lines. In red: selected genes. *EGFR* is considered as positive control and *SMAD3, BIRC3,* and *SLC9A5*; selected as favorite genes for the next steps. *SMAD3, BIRC3, SLC9A5,* and *AFAP1* are BRAFi-resistance genes and potent tumor-promoting genes (Fig 1). Raw data are available in Table EV5 (BRAFi for PBV, Appendix Fig S1)

G Heat map recapitulating the expression levels of the selected hits (red dots in Fig 2F) in BRAFi-resistant and BRAFi-sensitive cell lines. Markers of differentiation (*MITF, MLANA, OCA2, TYRP1, TYR*). In red: resistance genes already published (*NRP1, AXL, ZEB1, LPAR1*). Scale corresponds to Z scores.

Source data are available online for this figure.

that were highly resistant to BRAFi according to the Cancer Cell Lines Encyclopedia (CCLE) versus six sensitive cell lines (Barretina *et al,* 2012; Fig 2E). It is important to note that the CCLE BRAFi-resistant cell lines are therapy-naïve and intrinsically resistant.

We next focused on candidate genes which were both enriched in CRISPR screen and highly expressed in the majority of BRAFi-resistance cell lines (Fig 2F and G). To better evaluate the validity of

our candidate genes identified using this workflow, we added well-established and validated BRAFi-resistant genes (*NRP1, AXL, ZEB1,* and *LPAR1*) (Müller *et al,* 2014; Konermann *et al,* 2015; Rizzolio *et al,* 2018) and five genes associated with melanoma cell differentiation (*MITF, OCA2, MLANA, TYR,* and *TYRP1*) (Levy *et al,* 2006). All BRAFi-resistant cell lines presented a dedifferentiated profile as anticipated. Importantly, our candidate genes including *SMAD3,*

*SLC9A5,* and *BIRC3* (Fig 2G, black color) displayed a similar expression profile than observed for the well-established and validated BRAFi-resistant genes (Fig 2G, red color), strongly suggesting that these genes may also confer BRAFi resistance. *EGFR* and platelet-derived growth factor receptor (*PDGFR)-β* showed high expression only in a few BRAFi-resistant cell lines, as previously observed in patients (Sun *et al*, 2014).

Together, our results confirmed the robustness of the functional *in cellulo* CRISPR-SAM screen.

**Validation of BRAFi SAM-selected resistance genes**

To evaluate the contribution of the CRISPR-SAM-selected genes in BRAFi resistance, we examined the transcriptome of the differentiated cell line M229 (transitory cell state) at different stages during acquisition of resistance (Fig 3A; Song *et al*, 2017). As described by Song *et al* (2017), cell lines were exposed to chronic exposure to BRAFi and analyzed at different days of treatment (P: parental cells, 2D: two days of treatment, DTP: drug-tolerant persister cells, DTPP: drug-tolerant proliferating persister cells, SDR: single-drug-resistant cell). We observed a sequential upregulation in the expression of BRAFi SAM-selected genes: a group of genes (*MMP2, SEMA3B, BIRC3, TIPARP*, etc) being expressed earlier than a 2[nd] group (*IL6, EGFR, AFAP1*, etc). The majority of the candidates were overexpressed while the cells exhibited resistance to a single BRAFi agent (single-drug resistance, SDR). Comparable results were obtained with the M238 melanoma cell line (Figs 3B and EV2A). Importantly, we found that combining the BRAFi with MEKi (DDR) led to comparable upregulation of the BRAFi SAM-selected genes than observed in cells exposed to BRAFi alone (SDR) (Fig 3C). These results indicate that a common gene expression program can confer resistance to inhibitors of MAPK pathway as previously reported (Moriceau *et al*, 2015; Corre *et al*, 2018; Lee *et al*, 2020).

Having shown that combination of BRAFi and MEKi promotes sequential upregulation of BRAFi SAM-selected genes, we monitored their expression levels in an *in vivo* preclinical patient-derived xenograft (PDX) model (Rambow *et al*, 2018). Using scRNA-Seq, our collaborators reported the presence of dedifferentiated drug-tolerant cells exhibiting a neural crest stem cell (NCSC) and invasive profiles at minimal residual disease isolated from the MEL006 PDX model (Rambow *et al*, 2018). Here, we showed that the BRAFi SAM-selected genes were highly and selectively expressed in both of these cell populations at mRNA level (Fig 3D and E). To reinforce these observations, we performed SMAD3 immunostainings in four BRAF-mutant PDXs exposed to BRAF/MEK inhibitors until resistance (recently characterized in (preprint: Marin-Bejar et al, 2020), Appendix Fig S2A and B). Immunostainings showed the emergence of SMAD3[+] cells in Dabrafenib + Trametinib resistant lesions from the MEL003 and MEL006 PDXs in contrast to PDXs characterized by an intrinsic resistance mechanism (MEL007 and MEL037).

Moreover, we found that EGFR-expressing cells sorted from melanoma tumors displayed comparatively high expression levels of the BRAFi SAM-selected genes (Fig EV2B and C). Importantly, these EGFR-positive cells are able to proliferate in the presence of BRAFi and generate BRAFi-resistant colonies (Shaffer *et al*, 2017). In addition, high expression levels of the BRAFi SAM-selected genes have been found in invasive cells when compared to proliferative melanoma cell lines (Verfaillie *et al*, 2015; Fig EV2D). Together, these data confirm the upregulated expression of BRAFi SAM-selected genes in cells shown to contribute to relapse, suggesting their involvement in establishing drug-tolerant and/or resistant phenotypes *in vivo*.

To evaluate the clinical relevance of the SAM-selected BRAFi-resistance genes, we compared their expression levels (median) in two independent BRAFi drug naive/drug-resistant patient cohorts (Fig 3F and G, n = 21 (Hugo *et al*, 2015b) and n = 16 (Rizos *et al*, 2014) patients, respectively). The expression levels of the selected resistant candidate genes increased during relapse in drug-resistant patients (48 and 31%). Notably, none of these genes have been implicated in recurrent gene-amplification events that are sometimes identified in drug-naive lesions (cBioPortal, TCGA; Gao *et al*, 2013) (Fig EV2E–H). These data indicate that the increase in expression of the SAM-selected genes in BRAFi-resistant cells is likely associated with a (non-genetic) dedifferentiation process of melanoma cells induced by the therapy. Together, these *in vitro* and *in vivo* gene expression analyses strongly support a BRAFi-resistance function for the SAM-selected genes.

**BRAFi-resistance genes promote tumor growth**

Long-term effect of BRAFi is reduced by the ability of persister cells to resist to BRAFi but also to promote the tumor growth (relapse). Thus, we investigated the capability of the BRAFi-persister cells

---

**Figure 3.   Validation of BRAFi SAM-selected resistance genes.**

A   Expression levels of our hits in M229 melanoma cells during the BRAFi-resistance acquisition. P: parental cells, 2D: two days of treatment, DTP: drug-tolerant persister cells, DTPP: drug-tolerant proliferating persister cells, SDR: single-drug-resistant cells (BRAFi) (Song *et al*, 2017). Scale corresponds to Z scores.

B   Expression levels of our hits in M238 melanoma cells (Song *et al*, 2017) during the BRAFi-resistance acquisition. Scale corresponds to Z scores.

C   Expression levels of our hits in parental, single-drug-resistant (SDR, BRAFi), or dual drug-resistant (DDR, BRAFi + MEKi) SKMel28 melanoma cell lines (Song *et al*, 2017). Scale corresponds to Z scores.

D   T-distributed Stochastic Neighbor Embedding (t-SNE) plot showing the 4 drug-tolerant states (NCSC (neural crest stem cells), invasive, SMC (starved-like melanoma cells) and pigmented cells) according to single-cell RNA-seq (scRNA-Seq) performed in PDX MEL006 model exposed to BRAFi + MEKi (Rambow *et al*, 2018). Our BRAFi-resistance genes are mainly expressed in NCSC and invasive cells.

E   AUCell score for each drug-tolerant states. ****$P < 0.0001$, Mann–Whitney test.

F   Expression level of our genes (median) in cutaneous melanoma biopsies before the BRAFi treatment (baseline) and during the relapse. Cohort from (Hugo *et al*, 2015b). The expression level of our BRAFi-resistance genes is increased in relapse samples (48%, n = 21).

G   Expression level of our genes (median) in cutaneous melanoma biopsies before the BRAFi treatment and during the relapse. Cohort from (Rizos *et al*, 2014). The expression level of our BRAFi-resistance genes is increased in relapse samples (31%, n = 16).

Data information: See also Fig EV2.
Source data are available online for this figure.

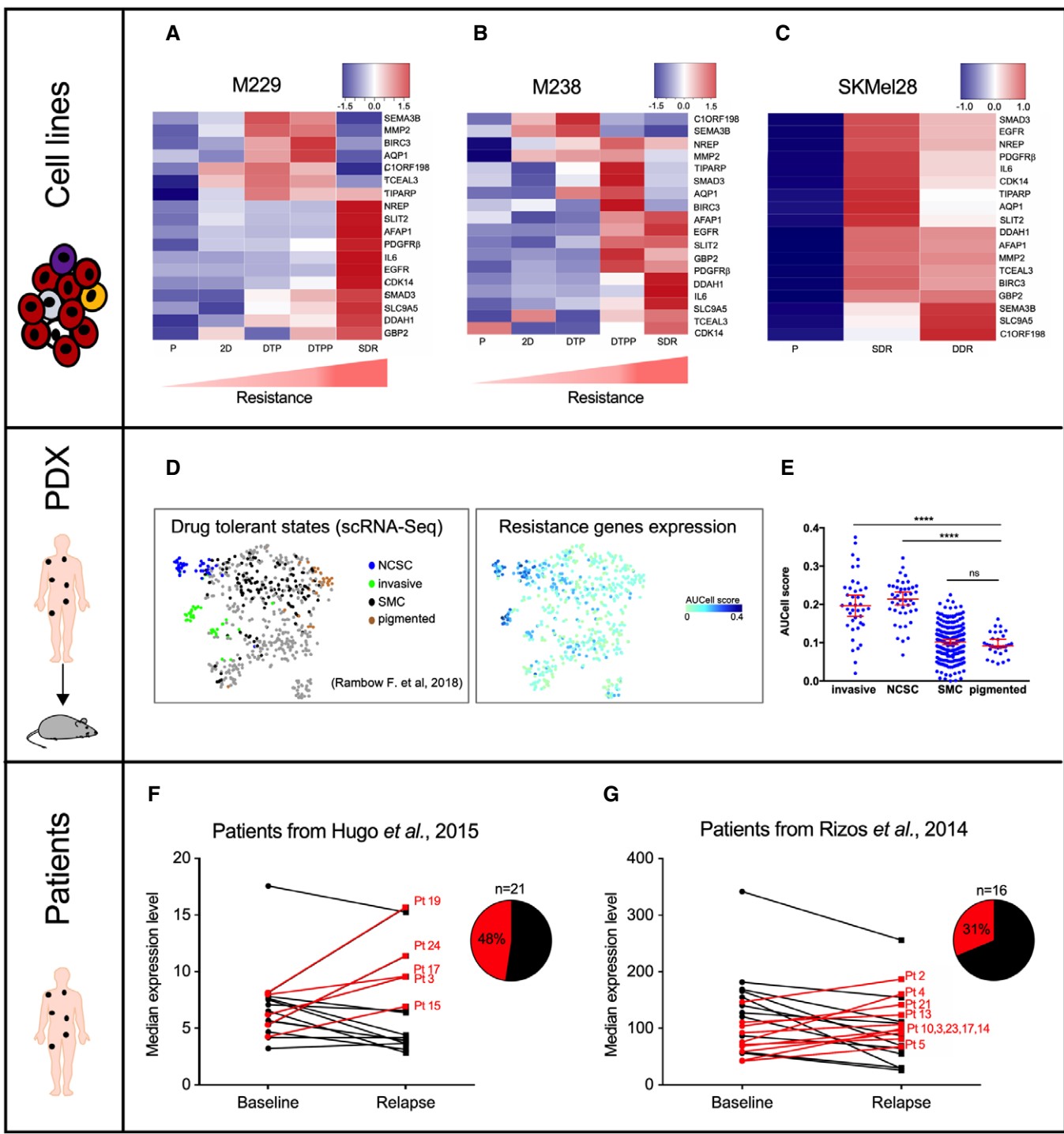

**Figure 3.**

(obtained from the *in vitro* CRISPR screen, Fig 2) to promote tumor growth into nude mice. The subset of BRAFi-resistant/persister cells (Fig 4A) was engrafted (36 × 10⁶ cells, 3 × 10⁶/mouse), and tumor growth was monitored (Fig 4B).

These BRAFi-persister cells formed tumors, in contrast to parental 501Mel cells (Ohanna *et al*, 2011). We determined the nature and abundance of sgRNAs present in each emerging tumor

(Fig 4C and Table EV7), and we detected 97 genes (sgRNAs) detected in all tumors. Interestingly, we recovered the tumor-promoting genes *SMAD3*, *BIRC3,* and *SLC9A5* identified above (Fig 4D). We looked for the enrichment of these sgRNAs in each tumor developed from persister cells (Fig 4E). *EGFR* sgRNA was not frequently enriched in tumors (as previously described for human melanoma tumors (Prahallad *et al*, 2012; Sun *et al*, 2014; Shaffer

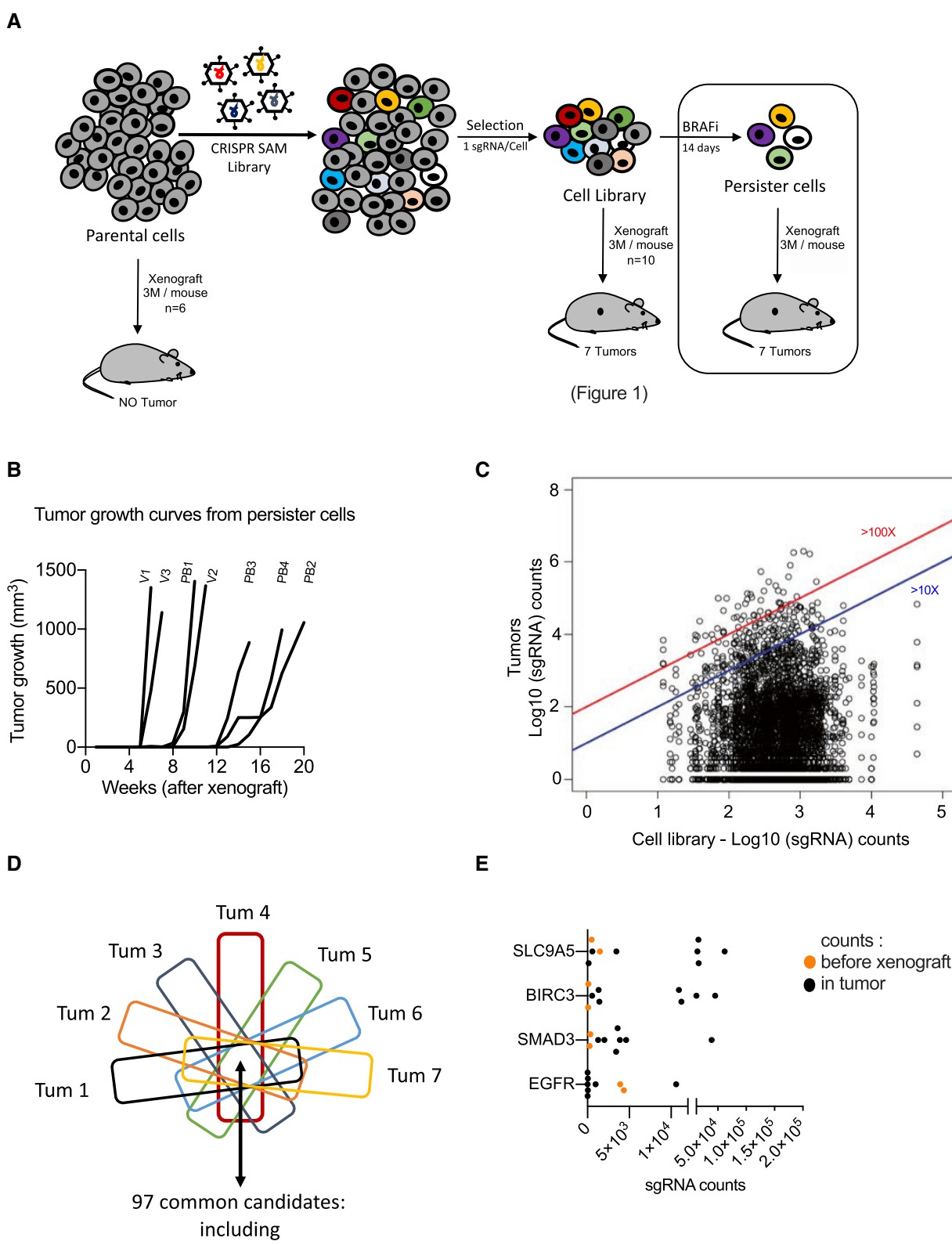

Figure 4.

**Figure 4. BRAFi-resistance genes promote tumor growth.**

A Workflow to identify genes involved in tumor growth from BRAFi-persister cells. Cell populations were subcutaneously grafted on nude mice flanks ($3 \times 10^6$ cells per mouse); Vem-persister cells ($n = 11$ mice) and PLX8394 (PB)-persister cells ($n = 12$ mice) (Table EV2). (BRAFi for PBV, Appendix Fig S1)

B Tumor growth curves from BRAFi-persister cells (monitored during 5 months; Table EV2). V for Vem-resistant cells and PB for PLX8394-resistant cells.

C Distribution of sgRNAs in BRAFi-resistant cells (before xenograft) and in the 7 tumors emerging from the BRAFi-resistant cells ($\log_{10}$(sgRNAs counts)). Blue and red lines indicated the enrichment $\geq 10$ fold or $\geq 100$ fold (tumors *versus* BRAFi-resistant cells (*in vitro*)). Raw data are available in Table EV7.

D From the seven tumors arising from BRAFi-resistant cells, the common genes (enriched) have been extracted. Ninety-seven genes including *SMAD3*, *BIRC3*, and *SLC9A5* are detected in all these 7 tumors. Raw data are available in Table EV7.

E sgRNAs counts in tumors versus sgRNA detected in BRAFi-resistant cell library (*in vitro*) (respectively, black and orange points) for selected candidates. Each black point corresponds to one tumor. *EGFR* was the most potent BRAFi-resistant gene (Fig 2). Two replicates have been shown for CRISPR-SAM cell library (orange points). *SMAD3*, *BIRC3*, and *SLC9A5* were identified as hits in the three screens (Figs 1, 2, and 4).

Source data are available online for this figure.

*et al*, 2017)) in contrast to *SMAD3*, *BIRC3*, and *SLC9A5* (Fig 4E and Appendix Fig S3). Together this suggests that these 3 genes are potential interesting targets for antimelanoma therapy.

### Functional validation of genes involved in BRAFi resistance and relapse

We focused on the transcription factor SMAD3 as a model gene for monitoring BRAFi resistance and relapse due to its critical function downstream of the TGFβ pathway. Although this pathway is known to promote melanoma phenotype switching/dedifferentiation (Sun *et al*, 2014), to support melanoma growth (Berking *et al*, 2001) and metastasis, little is known about the role of SMAD3 in melanoma biology and as a modulator of resistance to targeted therapy.

We confirmed that *SMAD3* mRNA is highly expressed in dedifferentiated cells (Fig 5A–C) and in BRAFi-resistant cells (Fig 2G).

To reinforce the role of SMAD3 in BRAFi resistance, we showed that gain-of-function of SMAD3 significantly increases the BRAFi resistance of melanoma cells when compared to different control cells (parental 501Mel cells and the CRISPR-engineered cells: 501Mel cells expressing dCas9 and HSF1-p65-MS2 (named here 501Mel 2+) and the 501Mel 2+ cells expressing a control guide (named 3+ backbone; Fig 5D). In addition, we showed that gain-of-function of SMAD3 also promotes the three-dimensional (3D) tumor spheroid invasion capability of melanoma cells. Similar results were obtained for *SLC9A5* (Figs 5E and EV3A). These results strongly suggest that a high expression level of *SMAD3* confers BRAFi resistance and invasion capability in melanoma cells.

Thus, we investigated if SMAD3 impairment re-sensitizes cells to BRAF inhibitor, using small-interfering RNA (siRNA) (Fig 5F–I). As the dedifferentiation status correlated with BRAFi resistance, we selected two BRAFi-resistant cell lines, SKMel28 BRAFi-resistant cells (SKMel28R, Fig 5F and G; Hugo *et al*, 2015b) and Me1402 melanoma cells (Fig 5H). In contrast to the SKMel28R, the resistance of which was created by chronic exposure to non-lethal doses of BRAFi, Me1402 cells are intrinsically resistant.

Surprisingly, the single SMAD3 depletion decreased the cell density in a similar magnitude than BRAFi treatment in these BRAFi-resistant cells (Figs 5G and H, and EV3B). The *SMAD3* depletion did not modify the ERK pathway (Fig 5I) in contrast to the BRAFi. The combo (*SMAD3* depletion and BRAFi (5 µM)) efficiently reduced the number of resistant/persister cells in these two cell lines, suggesting that SMAD3 is an interesting target to limit resistance to BRAFi. Similar results were obtained by targeting *BIRC3*, *EGFR*, *IL6*, or *AQP1* (Fig EV3C–J).

To transfer this strategy (SMAD3 inhibition + BRAFi) into clinic, we looked for an efficient inhibitor of SMAD3. We identified the chemical inhibitor SIS3 (SMAD3 inhibitor, SMAD3i; Jinnin *et al*, 2006; Wu *et al*, 2020). Firstly, we validated the inhibitory efficiency of SMAD3i in melanoma cells since it decreased the levels of phospho-SMAD3 Ser423/425 induced by TGFβ (Fig 5J) in accordance with previous studies (Jinnin *et al*, 2006; Chihara *et al*, 2017). Since the SMAD3 regulation by phosphorylation is not fully understood and the phospho-SMAD3 Ser423/425 status is not strictly correlated with SMAD3 transcriptional activity (Ooshima *et al*, 2019), we evaluated the effect of SMAD3i using a reporter assay. We demonstrated that SMAD3i reduced the transcriptional activity of SMAD3 in response to TGFβ exposure in 4 melanoma cell lines (Fig 5K). To know if the combination (SMAD3i + BRAFi) could be broadly used to eradicate persister cells emerging in response to BRAFi treatment, we selected three melanoma cell lines in function of *SMAD3* expression levels (Fig 5L). The differentiated cells (SMAD3^low) are highly sensitive to BRAFi (decrease of cell density: > 80% at 5 µM BRAFi, Fig 5M)) in contrast to dedifferentiated cells (SMAD3^high), which are highly resistant to BRAFi (decrease of cell density: ~50% at 5 µM BRAFi, Fig 5M).

We showed that SMAD3i (SIS3) reduced the cell density of all melanoma cell lines (Fig 5N). These results are in agreement with the SMAD3 knock-down results (Fig 5G and H). Our experiments also indicated that melanocyte survival is weakly affected by SMAD3 inhibition (up to 20 µM), in agreement with the non-toxicity of this inhibitor observed *in vivo* (Tang *et al*, 2017; Wu *et al*, 2020). The inhibitory effect of SMAD3i on melanoma cells seems to be associated with the total SMAD3 expression levels (Fig 5L). Our results might suggest that these melanoma cells could be "addicted" to SMAD3 activity.

We finally investigated the interest to combine BRAFi (Vem, alone or in combination with MEKi (Cobi)) and SMAD3i (SIS3) to eradicate the BRAFi-resistant cell lines (SKMel28R, M229R, and M238R) (Figs 5O and EV3K). We showed that SMAD3 inhibitor alone or in combination with BRAFi (Vem 5 µM) or BRAFi + MEKi (Cobi 1 µM) might be a promising treatment to reduce the amount of persister cells (melanoma). Together, these results identified *SMAD3* as an amenable target to limit resistance to BRAFi and tumor growth.

### The transcription factor AhR drives *SMAD3* expression

Having shown that SMAD3 expression mediates BRAFi-resistance and tumor growth, we explored the transcriptional program promoting its expression in BRAFi-resistant melanoma cells. We recently

reported that the Aryl hydrocarbon Receptor (AhR), a ligand-dependent transcription factor is an upstream central node regulating the expression of BRAFi-resistance genes and melanoma dedifferentiation (Corre *et al*, 2018; Leclerc *et al*, 2021; Paris *et al*, 2021).

We postulated that AhR may govern *SMAD3* expression in BRAFi-resistant cells. We identified three putative canonical binding sites for AhR (XRE for xenobiotic responsive element) in the proximal promoter of *SMAD3* (Fig 6A) in accordance with chromatin

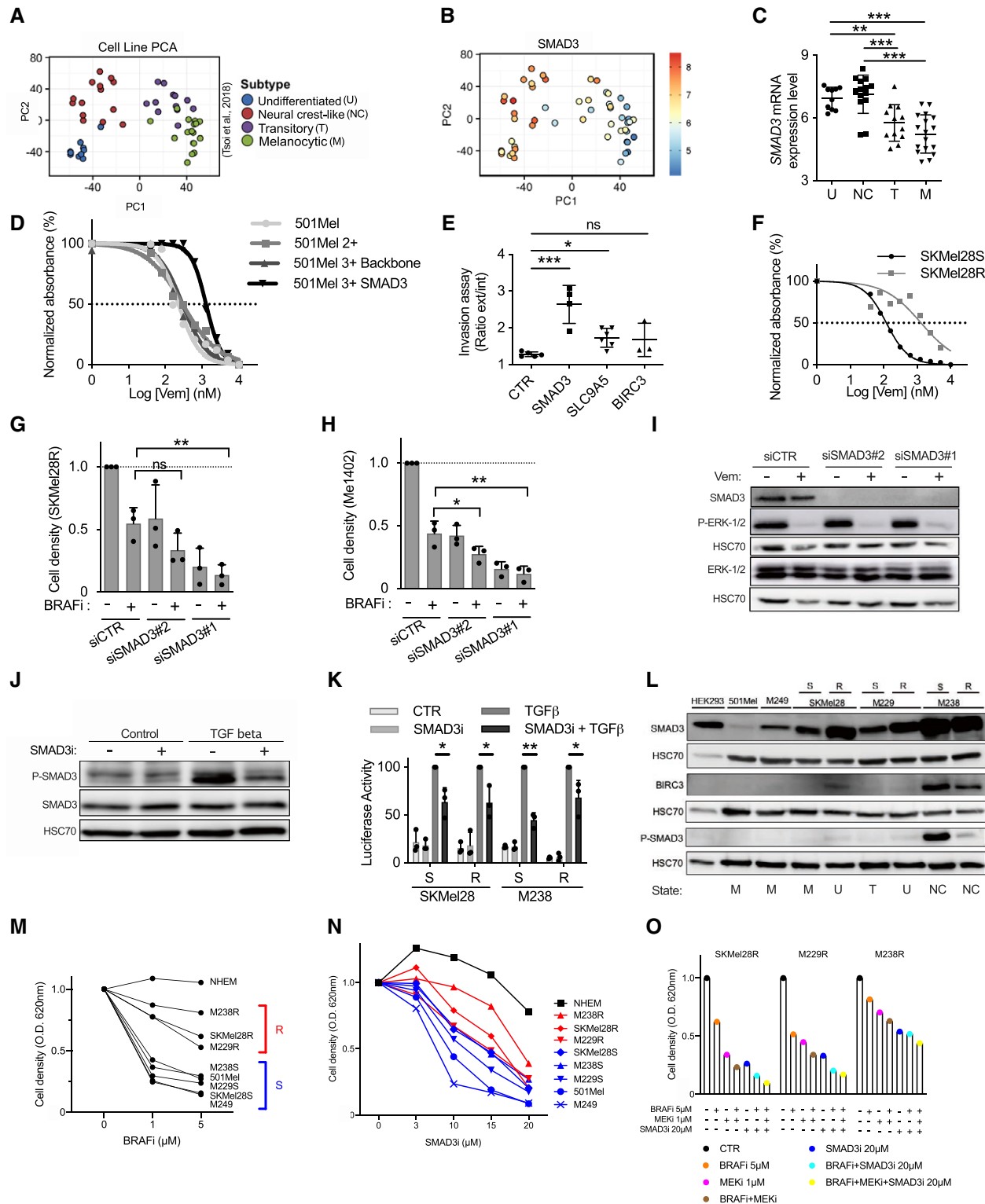

Figure 5.

**Figure 5. Functional validation of genes involved in BRAFi resistance and relapse.**

A   PCA analysis of melanoma cell lines in function of their dedifferentiation states (generated by the webtool http://systems.crump.ucla.edu/dediff/index.php).

B   SMAD3 expression increases with melanoma cell dedifferentiation. PCA analysis of *SMAD3* expression in melanoma cell lines in function of their dedifferentiation states. Scale: red color corresponds to a high *SMAD3* expression level (unit of the scale: $Log_2$ FPKM).

C   *SMAD3* expression in these four subtypes of melanoma cells (U, undifferentiated; NC, neural crest-like; T, transitory; M, melanocytic). Number in each group: U = 10, NC = 14, T = 12, M = 17. Error bars reflect mean ± s.d. Multiple comparisons have been done using ordinary one-way ANOVA **$P < 0.01$, ***$P < 0.001$.

D   SMAD3 gain-of-function increases BRAFi resistance. Determination of BRAFi half maximal inhibitory concentrations ($IC_{50}$ values) for 501Mel cell lines ($Log$[Vem] (nM)). Parental 501Mel cells ($n = 5$), 501Mel cells expressing dCas9 and HSF1-p65-MS2 (named here 501Mel 2+, $n = 5$), the 501Mel 2 + cells expressing a control guide (backbone, $n = 5$) and the 501Mel 2 + cells overexpressing *SMAD3* ($n = 3$ biologically independent experiments). A representative experiment has been chosen. Dotted line is used to determine the $IC_{50}$ values.

E   *SMAD3* and *SLC9A5* gain-of-function increases invasion capability of melanoma cells. Invasion assays for engineered cell lines (melanoma spheroids): CTR; 501Mel 2+, SMAD3; 501Mel 2 + cells overexpressing *SMAD3*. Two other cell lines overexpressing *SLC9A5* or *BIRC3* have been tested. Explanation for ratio calculation is detailed in Fig EV3A. Results obtained from two biologically independent experiments. ($n = 5$, 4, 6, and 3 spheroids, respectively). Error bars reflect mean ± s.d. Multiple comparisons have been done using ordinary one-way ANOVA, *$P < 0.05$, ***$P < 0.001$.

F   Characterization of SKMel28 sensitive and resistant cell lines (SKMel28S and SKMel28R). Determination of BRAFi half maximal inhibitory concentrations ($IC_{50}$ values) ($Log$[Vem] (nM)). A representative experiment has been chosen among two experiments. Dotted line is used to determine the $IC_{50}$ values.

G   *SMAD3* depletion (siRNA#1 & #2) decreased cell density and increased BRAFi effect (vemurafenib) on BRAFi-resistant cells (SKMel28R). CTR for non-targeting siRNA. DMSO for dimethylsulfoxide (solvent of vemurafenib; BRAFi). $n = 3$ biologically independent experiments. Each histogram represents the mean ± s.d.; Multiple comparisons have been done using ordinary one-way ANOVA, **$P < 0.01$.

H   *SMAD3* depletion (siRNA#1 & #2) decreased cell density and increased BRAFi effect (vemurafenib, BRAFi) on BRAFi-resistant cells (Me1402). CTR for non-targeting siRNA. DMSO for dimethylsulfoxide (solvent of vemurafenib; BRAFi). $n = 3$ biologically independent experiments. Each histogram represents the mean ± s.d.; multiple comparisons have been done using ordinary one-way ANOVA, *$P < 0.05$, ** $P < 0.01$.

I   Validation of SMAD3 knock-down by Western blot experiments in Me1402 cells exposed or not to vemurafenib (BRAFi (Vem) 5 μM, 2 days). Cells were exposed to BRAFi 24 h after siRNA transfection. Vemurafenib inhibitory effect on mutated BRAF was evaluated by analyzing the phospho-ERK1/2 levels. ERK and HSC70 serves as loading control.

J   Validation of SMAD3 inhibitor (SIS3, SMAD3i). Effect of SMAD3i (10 μM) on the level of phospho-SMAD3 Ser423/425 in response to TGFβ (2 ng/ml, 1 h) (or solvent: HCl 4 mM + Bovine serum albumin 1 mg/ml) in SKMel28R cells (expressing a high endogenous level of *SMAD3* mRNA). Serum starved cells (500,000 per well) were pretreated with SMAD3i 10 μM (or control solvent) during 2 h before TGFβ addition. SMAD3 and HSC70 serve as loading control for Western blot experiments.

K   Inhibitory effect of SMAD3i (SIS3) on the SMAD-responsive luciferase activity. Vector encodes the Firefly luciferase reporter gene under the control of a minimal (m) CMV promoter and tandem repeats of the SMAD Binding Element (SBE). Cells (10,000 per well) were pretreated with SMAD3i 10 μM or control solvent during 1.5 h, and next cells were exposed to TGFβ 10 ng/ml (or solvent: HCl 4 mM + Bovine serum albumin 1 mg/ml) for 6 h. $n = 3$ biologically independent experiments. Each histogram represents the mean ± s.d.; Bilateral Student test (with non-equivalent variances); TGFβ *us* TGFβ+SMAD3i, *$P < 0.05$, **$P < 0.01$.

L   SMAD3, Phospho-SMAD3 Ser423/425, and BIRC3 expression levels in melanoma cell lines. Four subtypes of melanoma cells (U, undifferentiated; NC, neural crest-like; T, transitory; M, melanocytic) have been compared. 501Mel and M249 cells are melanocytic cells in contrast to dedifferentiated BRAFi-resistant cells (R). Three couples of melanoma cell lines (Sensitive (S) and (R)) have been used to illustrate the SMAD3 and BIRC3 increases in BRAFi-resistant cell lines. HEK293 cells are used as control (kidney). HSC70 serves as loading control for Western blot experiments. Each antibody has been evaluated on individual membrane (explaining the three HSC70, loading controls).

M   The vemurafenib decreased melanoma cell density. Cell lines have been exposed to Vem (1 or 5 μM, 84 h) to define two groups of cell lines (sensitive (blue) and "resistant" (red) cell lines). Normal human melanocytes (NEHM) have been used to evaluate the effect of Vem on normal cells (mean of 3 independent donors). Representative values (mean of biological triplicates) of $n = 3$ biologically independent experiments.

N   The SMAD3 inhibitor decreased melanoma cell density. Cell lines have been exposed to SMAD3i (0, 3, 10, 15, or 20 μM, 84 h). The cell lines defined as S and R to BRAFi in the item 5 M are indicated in blue and red. Normal human melanocytes (NEHM) have been used to evaluate the effect of SMAD3i on normal cells ($n = 3$ independent donors). The SMAD3i effect on NHEMs is weak and manageable for these normal cells (NHEM). Representative values (mean of biological triplicates) of $n = 2$ biologically independent experiments.

O   The chemical inhibition of SMAD3 by SIS3 (SMAD3i) improved current therapy effect (BRAFi + MEKi; Vem 5 μM and Cobi 1 μM) on BRAFi-resistant cells (SKMel28R, M229R & M238R). Cells have been treated as detailed for panel M. Representative values (mean of biological triplicates) of $n = 2$ biologically independent experiments.

Data information: Western blot results are representative of at least two independent experiments. Source data and unprocessed original blots are available in Appendix Fig S2 source data. See also Fig EV3.
Source data are available online for this figure.

immunoprecipitation coupled to massively parallel DNA sequencing data (AhR ChIP-Seq) showing AhR binding on SMAD3 promoter (Lo & Matthews, 2012; Yang *et al*, 2018). To demonstrate the *SMAD3* induction by AhR, 501Mel cells were exposed to the most potent and well-known AhR ligands (TCDD for 2,3,5,7-tetrachlorodibenzo-dioxin; ITE for 2-(1*H*-Indol-3-ylcarbonyl)-4-thiazolecarboxylic acid methyl ester). These AhR ligands increased *SMAD3* expression, in an AhR-dependent manner (Fig 6B). Comparable results were obtained with a canonical target gene of AhR; the TCCD-induced poly(ADP ribose) polymerase gene (*TIPARP*), supporting the role of AhR in regulating *SMAD3* expression (Fig 6C). The need of an activated AhR-promoting *SMAD3* expression was further confirmed by the use of an AhR antagonist (CH-223191) in SKMel28 cells

(Fig 6D). Long-term chemical inhibition of AhR activity reduced *SMAD3* and *TIPARP* expression levels. In accordance with these results, *SMAD3* expression levels decreased in AhR KO SKMel28 cells (Fig 6E).

Together, our results support the hypothesis that AhR activity drives *SMAD3* expression along with the acquisition of BRAFi resistance.

**SMAD3 drives phenotype switching and resistance to melanoma therapies**

To explore the mechanism underlying therapy sensitization upon SMAD3 inhibition, we examined the transcriptional program

regulated by SMAD3 (Fig 7). We hypothesized that the transcription factor SMAD3 may regulate the expression levels of several resistant genes and thereby induce a multifactorial effect, in which multiple drug resistance pathways are activated. We compared the SMAD3 ChIP-Seq (Ramachandran *et al*, 2018) data to BRAFi-resistance gene sets established from three different sources, namely the enclosed screen, the screen performed in A375 (Konermann *et al*, 2015), and other established BRAFi-resistance genes such as *AXL* or *NRP1* (Fig 7A). We established a list of SMAD3-regulated genes, which includes *SLIT2, RUNX2, NRP1, MMP2, JUNB, ITGB5, AXL, AFAP1,*

and *EGFR* (SMAD3-signature). We next showed that basal SMAD3-signature is higher in the three BRAFi-resistant cell lines when compared to parental cell lines (SKMel28S, M229S, and M238S) (Fig 7B). The stimulation of the TGFβ-SMAD3 pathway by the recombinant TGFβ promoted the SMAD3 signature in these melanoma cell lines (Fig 7C). The inducibility is higher in parental cell lines (S) since the basal expression level of genes forming the SMAD3 signature is weak in parental cells (Fig 7B). Next, we examined the SMAD3 signature in a large panel of melanoma cell lines representative of the distinct differentiation states (U - NC -T - M)

**A**

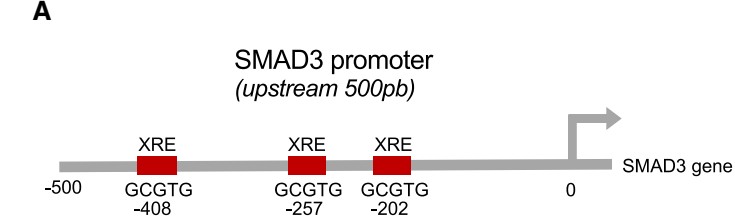

**B** *SMAD3* **C** *TIPARP*

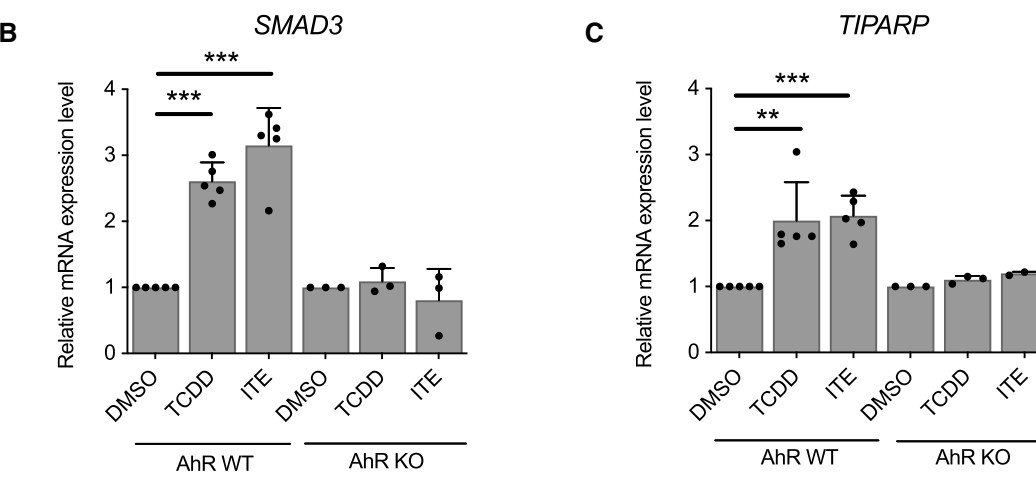

**D** **E**

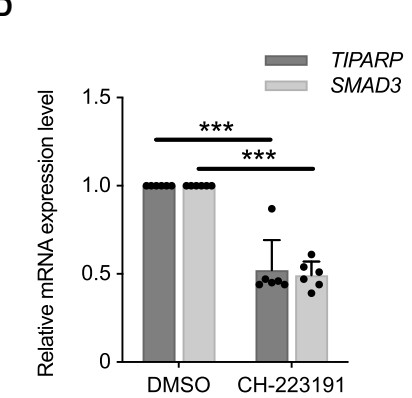
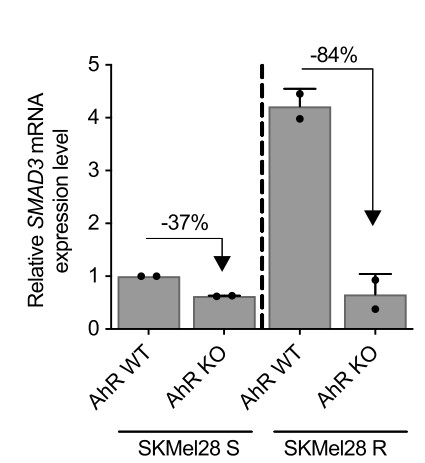

**Figure 6.**

 

**Figure 6.   The Transcription Factor AhR Drives *SMAD3* Expression.**

A    AhR binding sites (xenobiotic responsive element (XRE); GCGTG) in human *SMAD3* proximal promoter.
B    AhR activation by exogenous and endogenous ligands promotes *SMAD3* induction. 501Mel cells AhR wild-type or knockout have been exposed to exogenous and endogenous AhR ligands; TCDD (5 nM) or ITE (10 μM) or the solvent (DMSO) during 10 days. *n* = 5 biologically independent experiments for AhR WT cells and *n* = 3 for AhR KO cells. Each histogram represents the mean ± s.d.; multiple comparisons have been done using ordinary one-way ANOVA, ****P* < 0.001.
C    AhR activation by exogenous and endogenous AhR ligands promotes *TIPARP* induction. 501Mel cells have been treated as described in B. *n* = 5 biologically independent experiments for AhR WT cells and *n* = 3 for AhR KO cells. Each histogram represents the mean ± s.d.; multiple comparisons have been done using ordinary one-way ANOVA, ***P* < 0.01, ****P* < 0.001.
D    AhR antagonist (CH-223191) reduces *SMAD3* and *TIPARP* expression levels. SKMel28 cells (AhR wild type) have been exposed to CH-223191 (5 μM) or the solvent (DMSO) during 7 days. *n* = 6 biologically independent experiments. Each histogram represents the mean ± s.d.; Bilateral Student test (with non-equivalent variances): ****P* < 0.001.
E    Loss of AhR reduces *SMAD3* expression levels. SMAD3 expression has been investigated in SKMel28 cells AhR wild-type (WT) or knockout (KO). SKMel28R has been obtained from SKMel28S by chronic exposure to non-lethal doses of BRAFi (Hugo *et al*, 2015b). R for BRAFi-resistant SKMel28 cells and S for sensitive. *n* = 2 biologically independent experiments. Each histogram represents the mean ± s.d.

Source data are available online for this figure.

(*n* = 53) (Fig 7D) and cutaneous melanoma (*n* = 118, TCGA cohort, tumors not exposed to targeted therapy; Fig 7E). The SMAD3 signature correlated with a dedifferentiation status as suggested above (Figs 2G and 5B, and EV4A). Importantly, a subset of BRAF(V600E) melanoma patients (~20%) expressing the SMAD3-signature was identified (Fig 7E), suggesting that these tumors contained dedifferentiated melanoma cells with potential intrinsic resistance to BRAFi. In addition, SMAD3 signature could be useful to clinicians to propose immune checkpoint immunotherapy to their patients since SMAD3 signature identified anti-PD1 non-responders (prior the selection of the treatment) (Fig 7F). Therefore, these results strongly suggest that the SMAD3-signature may be useful to identify a population of pre-existing drug-resistant cells within drug-naive lesions. To further illustrate the clinical relevance of our results, we assessed the expression levels of the SMAD3 signature in drug-naive and BRAFi-resistant patients using publicly available dataset. SMAD3 signature increased in 14/16 patients exposed to BRAFi (baseline vs. relapse, Fig 7G). Altogether, these results indicated that the SMAD3 signature is associated with resistance to both current melanoma therapies.

The analyses of the SMAD3 signature in tumor samples highlighted the tumor heterogeneity (mRNA expression level of genes forming the SMAD3 signature is variable between patients, and all genes forming the SMAD3 signature are not high in each tumor.). As expected, this heterogeneity has been retrieved in melanoma cell lines (Fig 7H). In response to SMAD3 depletion, *AXL* and *EGFR* decreased in the two models in contrast to other genes such as *MMP2,* which decreased in only one model. Altogether, our results indicate that the SMAD3 signature assessment is probably more appropriate than quantification of specific mRNAs such as *EGFR* or *SLIT2* to track pre-existing drug-resistant cells within drug-naive lesions.

As the shift toward the mesenchymal-like state confers broad resistance to therapies (Redfern *et al*, 2018), we postulated that the SMAD3 signature may be associated with this particular dedifferentiated phenotype. Comparing the SMAD3 signature with the classical mesenchymal-like signature (Mak *et al*, 2016) of melanoma (TCGA cohort) highlighted a significant correlation (Figs 7I and EV4A). To further confirm the link between the SMAD3 signature and a mesenchymal state, we searched for similarities with other mesenchymal states identified in two other cancers (glioblastoma [GBM] and hepatoma). As described for the cutaneous melanoma, different differentiation states have been characterized for GBM

(proneural, classical, and mesenchymal GBM) (Jin *et al*, 2017). We found that the SMAD3 signature overlaps with the mesenchymal GBM signature (Fig 7J). Mesenchymal GBM is the most aggressive GBM usually associated with poor overall survival (Patel *et al*, 2014; Jin *et al*, 2017). The SMAD3 signature was also associated with epithelial–mesenchymal transition (EMT) in hepatoma (Fig EV4B and C).

Altogether, these results indicate that the transcription factor SMAD3 and its downstream target genes confer resistance to targeted therapies by promoting a mesenchymal-like phenotype. Our work identifies AhR-SMAD3 axis as a target to overcome therapy resistance of melanoma.

# Discussion

Targeted therapy and immunotherapy have greatly improved the prognosis of patients with cancer, but resistance to these treatments restricts the overall survival of patients. Increasing evidence indicates that transcriptomic reprogramming is associated with persister cell emergence (Puisieux *et al*, 2014; Bai *et al*, 2019) but the mechanism underlying resistance from this pool of cells remains elusive. Targeting tumor-promoting genes leading reprogramming could therefore constitute an attractive approach to prevent relapse, at least in some specific contexts (Bailey *et al*, 2018). Here, using a whole genome approach, we searched for pathways that trigger the transcriptional reprogramming of persister cells into drug-resistant cells. Based on CRISPR screen, data mining, and *in vivo* experiments, we identified and validated three genes (*SMAD3*, *BIRC3*, and *SLC9A5*) able to promote both BRAFi-resistance and tumor growth. Our work expands our understanding of the biology of persister cells and highlights novel drug vulnerabilities that can be exploited to develop long-lasting antimelanoma therapies.

Even if CRISPR-SAM screen is a leading-edge genetic tool, several concerns must be considered. As observed for all screening approaches, false positives and false negatives are engendered rendering the validation experiments a crucial step. In this study, we clearly showed that the number of sgRNAs per target is an important parameter. For our best hit, *EGFR*, only two sgRNAs were enriched in BRAFi-treated cells. Thus, it is highly likely that we missed interesting BRAFi-resistance genes (false negatives) due to the number of sgRNA/gene (at least 3 sgRNAs/gene in this library).

A recent publication confirmed that sgRNAs are not all functional in CRISPRa libraries and it could be interesting to increase the number of sgRNAs per target and to cover more TSS per gene (Sanson *et al*, 2018). Interestingly, the sgRNA library used in our study displays for several genes up to 27 sgRNAs. These sgRNAs target different isoforms (or TSS) of these genes. By examining the sgRNAs targeting *SMAD3*, we found that 2 sgRNA are enriched (BRAFi resistance)

(Fig EV5). These two sgRNAs promote the expression of the longest *SMAD3* isoform; the *SMAD3* mRNA expressed in our model (501Mel cells) (Fig EV5A–E). It is important to note that the nine other sgRNAs targeting *SMAD3* are not able to confer the BRAFi resistance since they target other SMAD3 isoforms. Based on these observations, we believe that mRNA isoforms identified by RNA sequencing should be considered during the sgRNAs selection for each model.

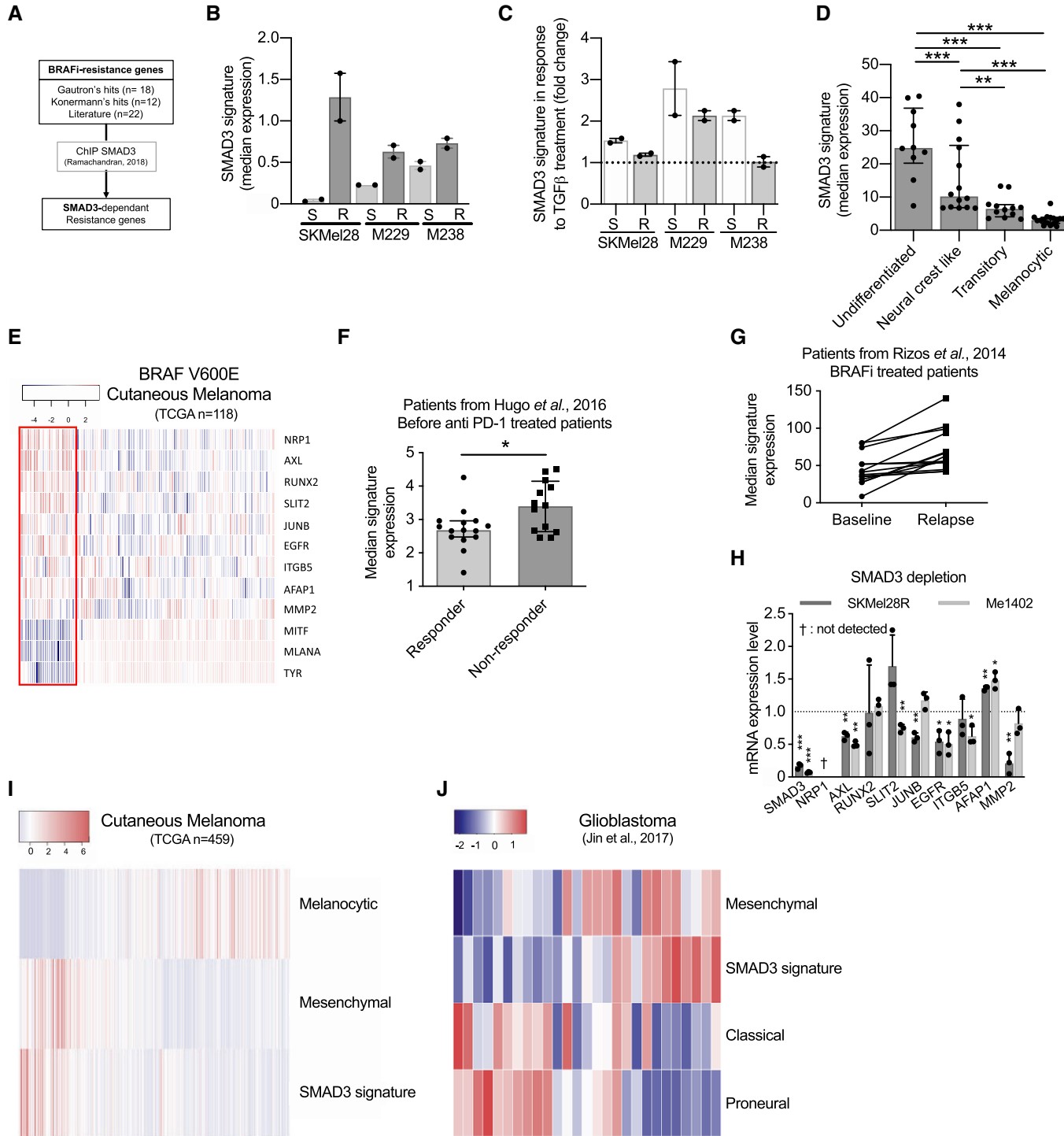

**Figure 7.**

**Figure 7. SMAD3 drives phenotype switching and resistance to melanoma therapies.**

A SMAD3 signature has been established by comparing BRAFi-resistance genes and SMAD3-regulated genes identified by chromatin immunoprecipitation followed by DNA sequencing (ChIP-Seq) (Ramachandran et al, 2018).

B Basal SMAD3 signature in six melanoma cell lines (S for sensitive to BRAFi and R for Resistant). n = 2 biologically independent experiments. Each histogram represents the median with interquartile range.

C Inducibility of SMAD3-signature in six melanoma cell lines exposed to TGF-β (10 ng/ml, 48 h). Data were normalized to cell lines exposed to solvent (4 mM HCl + 1 mg/ml human BSA). Values obtained with the TGF-β stimulated M238R cell line have been set to 1. n = 2 biologically independent experiments. Each histogram represents the median with interquartile range.

D SMAD3 signature discriminates differentiation states of 53 melanoma cell lines (Tsoi et al, 2018). The SMAD3 signature in four subgroups of melanoma cell lines. Each point represents a cell line (n = 17, 12, 14, and 10, respectively, for melanocytic, transitory, neural crest-like, and undifferentiated cell lines); each histogram represents the median with interquartile range; multiple comparisons have been done using ordinary one-way ANOVA; **P < 0.01, ***P < 0.001.

E Heat map depicting mRNA levels of SMAD3 signature in BRAF (V600E) non-treated melanoma patients (dataset from TCGA; SKCM, BRAF(V600E) mutated: n = 118). Three pigmentation genes (MITF, MLANA, and TYR) have been added to highlight the differentiation states of tumors. ~20% of tumors are considered as dedifferentiated tumors (red box) with a high SMAD3 signature. Scale corresponds to Z scores.

F SMAD3 signature in pre-treatment biopsies from responders and non-responders to anti-PD-1 treatment (from Hugo's cutaneous melanoma cohort, n = 15 and 13, respectively) (Hugo et al, 2016). Each histogram represents the median with interquartile range, one-tailed Mann–Whitney test; *P = 0.0324.

G SMAD3 signature in two groups (before BRAFi treatment or during relapse) of V600E patients from Rizos's cohort (Rizos et al, 2014) (14 on 16 patients displayed an upregulation of SMAD3 signature during relapse).

H SMAD3 depletion decreases expression of SMAD3-regulated genes. SMAD3 knock-down by siRNA decreased mRNA expression of BRAFi-resistance genes in SKMel28R and Me1402. NRP1 mRNA was not detected in our experimental conditions. n = 3 biologically independent experiments. Each histogram represents the mean ± s.d.; Bilateral Student test (with non-equivalent variances) *P < 0.05, **P < 0.01, *** P < 0.001. Dotted line highlights the value of 1.

I SMAD3 signature overlapped with mesenchymal signature in melanoma tumors (TCGA, n = 459) (Akbani et al, 2015). Melanocytic signature highlights the differentiation states of tumors (Corre et al, 2018). SMAD3 signature and mesenchymal signature correlate in melanoma tumors (TCGA, n = 459) (Mak et al, 2016). Scale corresponds to Z scores.

J The SMAD3 signature overlaps with the glioblastoma mesenchymal subtype (Jin et al, 2017). Scale corresponds to Z scores.

Source data are available online for this figure.

By this way, it would be easy to disqualify a part of the sgRNAs of the library targeting weakly expressed isoforms. This attitude could increase the score per isoform. In other words, the use of the z-score calculated per gene may discard interesting candidates. So, an analysis based on RNA isoforms could reduce the false negatives.

The second lesson of this CRISPR screen is the weak sgRNAs enrichment in BRAFi-exposed cells. Except for *EGFR*, the sgRNAs enrichment is about 2. Despite these values, we showed that these candidates are robust as for *SMAD3* (in vitro, in vivo, and in patients). It is tempting to explain this fact by the duration of treatment (BRAFi exposure) and the dose (2 μM). By increasing the dose (i.e., 5 μM), we showed that BRAFi killed more than 90% of 501Mel cells in four days (Fig 5M). This protocol is not achievable because it would induce too many false negatives. The other option consists to increase the duration of BRAFi treatment (and keep a low BRAFi concentration, i.e., 2 μM). However, a recent publication demonstrated that a long-term BRAFi exposure promotes a dedifferentiation process conferring BRAFi resistance (Tsoi et al, 2018; Bai et al, 2019). This alternative protocol could be perilous by inducing cell resistance to BRAFi independently of the sgRNA expression. Here, we selected a melanoma cell model exhibiting a differentiated profile as the vast majority of metastatic melanoma tumors (89% in the TCGA cohort), a short period of treatment (14 days) and an intermediate dose of BRAFi (2 μM). In fact, we followed, except the cell line, the protocol established by Feng Zhang's laboratory, who developed the CRISPR-SAM library (Konermann et al, 2015). The differentiation status of the cell line could be important since the transactivation mediated by CRISPR-SAM is possible only for active promoter in basal condition and the magnitude of transactivation relies on the basal expression level. Here, we showed that 501Mel cells express low level of *SMAD3* in basal condition and the transactivation obtained by CRISPR-SAM is substantial (Fig 1I). Moreover, SMAD3 is a transcription factor

which promotes the expression of various genes including potent BRAFi-resistance genes (*AXL* and *EGFR*). The robust transactivation of genes encoding a transcription factor, a transporter, or an enzyme is more inclined to be enriched with our protocol, especially if the basal gene expression is low. Here, we identified and validated in vitro and in vivo the transcription factor SMAD3 and the transporter SLC9A5, validated in vitro and in vivo.

BRAFi resistance relies, at least in part, on the phenotypic plasticity of melanoma cells (Tsoi et al, 2018; Rambow et al, 2018; Corre et al, 2018). These cells may escape the deleterious effect of drug combinations such as BRAFi + MEKi. Among these cells, those harboring a mesenchymal-like phenotype (usually named invasive cells) display high intrinsic resistance to MAPK therapeutics (Konieczkowski et al, 2014; Müller et al, 2014; Verfaillie et al, 2015; Shaffer et al, 2017). Enrichment in AXL[high] subpopulation (considered as invasive and mesenchymal-like cells) is a common feature of drug-resistant melanomas (Müller et al, 2014; Konieczkowski et al, 2014). Targeting mesenchymal-like cells using an antibody-drug conjugate, AXL-107-MMAE, showed promising effects in a preclinical model of melanoma (Boshuizen et al, 2018). The emergence of AXL[high] cells is currently explained by the decrease in *MITF* activity, but the mechanism of resistance to MAPK therapeutics remains unclear. Here, we demonstrate that the AhR-SMAD3 axis governs the expression levels of potent BRAFi-resistance genes, including *AXL*, *EGFR*, and *MMP2*.

The dedifferentiation process, conferring BRAFi resistance, requires transcriptomic reprogramming by transcription factors. Retinoic acid receptor gamma (RXRγ) was identified as a crucial transcription factor that promotes the emergence of the drug-tolerant subpopulation of NCSCs (Rambow et al, 2018). An increase in AXL[high]-positive cell population was reported following MAPK inhibition in the presence of an RXRγ antagonist. This increase may explain why this co-treatment only delays but does not completely

prevent relapse in PDXs (preprint: Marin-Bejar et al, 2020). This confirms the need to develop strategies that prevent melanoma dedifferentiation during BRAFi treatment. Thus, our data strongly suggest that SMAD3 is a key transcriptional factor involved in the emergence of drug-resistant mesenchymal-like cells in response to MAPK and identify a clinically compatible approach (SMAD3i) that might abrogate such a trajectory. Other transcription factors have been associated to BRAFi resistance such as JUN (Titz et al, 2016) and AhR (Liu et al, 2017; Corre et al, 2018). We recently showed that a sustained AhR activation promotes the dedifferentiation of melanoma cells and the expression of BRAFi-resistance genes (Corre et al, 2018). As proof-of concept, we demonstrated that differentiated and BRAFi-sensitive cells can be directed toward an AhR-dependent resistant program using AhR agonists. To abrogate the deleterious AhR sustained-activation, we identified Resveratrol, a clinically compatible AhR antagonist. Combined with BRAFi, Resveratrol reduces the number of BRAFi-resistant cells and delays relapse. Recently, an independent team confirmed that AhR inhibition is reachable in vivo using other AhR antagonists (Kyn 101, Ikena Oncology, or the CH-223191; Campesato et al, 2020) to improve the melanoma therapy. AhR antagonists validated in mice are currently evaluated in clinical trials (NCT04200963, Ikena Oncology) and (NCT04069026, Bayer). In addition, a water-soluble SMAD3 inhibitor has been recently published for the in vivo treatment (Wu et al, 2020), suggesting that clinical trials should start soon. Another option overcoming the potential pharmacological caveat would be to use antisense oligonucleotides (ASO) targeting AhR or SMAD3 (Leclerc et al, 2021). We and others demonstrated in melanoma that ASO strategy is feasible in vivo by targeting SAMMSON mRNA and the lncRNA TYRP1 (Leucci et al, 2016; Gilot et al, 2017).

In this study, we propose an AhR-SMAD3 impairment as a strategy to overcome melanoma resistance. Recently, conditional deletion of Smad7, a negative regulator of TGF-β/SMAD pathway, led to sustained melanoma growth and at the same time promoted metastasis formation (Tuncer et al, 2019), confirming that TGF-β/SMAD pathway is a promising target for melanoma (Javelaud et al, 2007). In addition, Rizos's team further illustrated the link between the TGF-β and melanoma therapy resistance. They showed that TGF-β promotes a dedifferentiation phenotype, which is a common mechanism of resistance to PD-1 inhibitors (Lee et al, 2020).

Several questions remain unsolved. We previously reported that activated AhR reprograms the transcriptome of melanoma cells mediating BRAFi resistance. In this study, we demonstrate that a SMAD3-regulated gene expression program promotes therapy resistance in cutaneous melanoma and EMT. Importantly, SMAD3 expression levels during resistance acquisition are dependent, at least in part, on AhR. Thus, it would be interesting to precisely define the role of AhR and SMAD3 in the induction of each BRAFi-resistance gene. To date, no physical interaction between AhR and SMAD3 proteins has been reported, suggesting that AhR acts as an upstream regulator of SMAD3 axis. It is noteworthy that the increased expression levels of SMAD3 by AhR expands the possibility of fine tuning gene expression since SMAD3 interacts with SMAD2 but also with JUN, TEADs, and YAP1 (Zhang et al, 1998; Fujii et al, 2012; Piersma et al, 2015). Because these three transcription factors have also been associated with therapies resistance in melanoma (Nallet-Staub et al, 2014; Ramsdale et al, 2015; Verfaillie

et al, 2015; Hugo et al, 2015b), our results suggest that regulation of BRAFi-resistance genes expression is multiparametric and probably more sophisticated than initially though. Nonetheless, the elucidation of these transcriptional programs and networks governing BRAFi-resistance genes and relapse is important for optimal target selection and the development of rationale and effective combination strategies.

BRAFi resistance may be achieved through the exposure of melanoma cells to TGF-β, demonstrating that transcriptome reprogramming may confer resistance without the need for pre-existing or de novo mutations (Viswanathan et al, 2017). The TGF-β pathway promotes a shift toward the mesenchymal state (Antony et al, 2019). The resulting dedifferentiation modifies the expression of the adhesion molecules in the cell, supporting a migratory and invasive behavior. Together, our results strongly indicate that the SMAD3-regulated genes are critical players in melanoma resistance to therapies by promoting an EMT-like process. EMT reversal represents a powerful approach, as it may reduce the invasive behavior of cancer cells and favor re-differentiation, synonymous of a decrease in BRAFi-resistance gene expression (Giannelli et al, 2014; Rodón et al, 2015). By combining anti-EMT drug and targeted therapy such as SMAD3i and BRAFi, we should efficiently reduce amount of persister cells. We anticipate that SMAD3 inhibition should limit the risk of resistance to therapies, since a decrease of expression levels of several BRAFi-resistance genes is obtained with SMAD3i (SIS3). SMAD3 inhibition is expected to be more efficient than inhibitors targeting single downstream targets such as AXL or EGFR. In conclusion, our work highlights novel drug vulnerabilities that can be exploited to develop long-lasting antimelanoma therapies.

Given the plasticity of melanoma cells and the capability of tumor microenvironment to produce TGF-β (Chakravarthy et al, 2018), our work also warrants further investigation of the source of TGF-β as another approach to prevent acquisition of the therapy-resistant mesenchymal phenotype.

# Materials and Methods

## Reagents

- DMSO—Sigma-Aldrich (D8418)
- BRAF inhibitors: Vemurafenib (PLX4032)—Selleckchem (RG7204); Paradox Breaker (PLX8394)—MedChemExpress (HY-18972)
- SMAD3 inhibitor: SIS3—Santa Cruz Biotechnology (sc-222318)
- MEK inhibitor: Cobimetinib—Selleckchem (GDC-0973)
- 2,3,7,8-TCDD (TCDD)—Sigma-Aldrich (48599)
- 2-(1H-Indol-3-ylcarbonyl)-4-thiazolecarboxylic acid methyl ester (ITE)—MedChemExpress (HY-19317)
- CH-223191—Selleckchem (S7711)
- TGF-β recombinant—Santa Cruz Biotechnology (240-B-010)
- BIRC inhibitor: Birinapant—Selleckchem (S7015)

## Cell lines and culture conditions

501Mel, Me1402, and HEK293T cell lines were obtained from ATCC. SKMel28 S & R cell lines were obtained from J.C Marine's laboratory at VIB Center for Cancer Biology, VIB, Leuven, Belgium. 501Mel

and SKMel28 AhR knockout cell lines have been established as previously described (Corre *et al*, 2018). M229S, M229R, M238S, M238R, and M249 were obtained from Thomas Graeber's laboratory at department of Molecular and Medical Pharmacology, University of California, Los Angeles, USA. All melanoma cell lines were grown in humidified air (37°C, 5% $CO_2$) in RPMI-1640 medium (Gibco BRL, Invitrogen, Paisley, UK) supplemented with 10% fetal bovine serum (FBS) (PAA cell culture company) and 1% penicillin–streptomycin (PS) antibiotics (Gibco, Invitrogen). HEK293T was grown in DMEM (Gibco BRL, Invitrogen, Paisley, UK) supplemented as melanoma cell lines media. SKMel28R, M229R, and M238R are cultivated in presence of 0.1 µM vemurafenib. All cell lines have been routinely tested for mycoplasma contamination (Mycoplasma contamination detection kit; rep-pt1; InvivoGen—San Diego—CA).

## CRISPR-SAM screens

A detailed protocol is available as supplementary Methods. Briefly, lentiviral productions have been performed as recommended (http://tronolab.epfl.ch), using HEK293T cells, psPAX2, pVSVG, and vectors required for CRISPR-SAM according to Zhang lab (Joung *et al*, 2017). Infections were performed overnight in presence of 4 or 8 µg of polybrene per ml. All vectors have been provided by Addgene. SgRNA Library was amplified and prepared as described by Zhang Lab (Joung *et al*, 2017). 501Mel cells were transduced to stably express dCAS-VP64 (cat. no. 61425) and MS2-P65-HSF1 (cat. no. 61426), before to transduce them with SAM sgRNA library (lentiSAMv2, 3-plasmid system, cat. no. 1000000057) at a MOI of 0.2. Infected cells have been selected using antibiotics: Blasticidin (2 µg/ml, 5 days), Hygromycin B (200 µg/ml, 5 days), and Zeocine (600 µg/ml, 5 days). *In vitro* CRISPR-SAM screens (Fig 2) were conducted as described by Zhang laboratory (Joung *et al*, 2017). The 501Mel cells expressing the sgRNA library were split into three groups: DMSO (solvent for BRAFi), vemurafenib (PLX4032, 2 µM, Selleckchem), and Paradox Breaker (PLX8394, 2 µM, MedChemExpress). After 14 days, resistant cell populations have been amplified. A minimum of $36 \times 10^6$ cells has been pellet and further sgRNA enrichment analysis. Cell libraries have been cryopreserved at −80°C for further *in vivo* experiments. The results presented are pooled data from two independent screens. To generate 501Mel cells overexpressing individually *SMAD3*, *BIRC3*, or *SLC9A5*, 501Mel expressing dCAS-VP64 and MS2-P65-HSF1 were transduced to stably express specific sgRNAs (Table EV8). Infected cells have been selected using zeocin (600 µg/ml, 5 days). Manipulations of lentivirus were performed in the biosafety level 3 containment laboratory core facility of the Biology and Health Federative Research Structure of Rennes (Biosit).

## Xenograft

Nude mice (4 weeks) were purchased from Janvier Labs and maintained under specific pathogen-free conditions in our accredited animal house (A 35_238_40). The animal study follows the 3R (replace_reduce_refine) framework and has been filed with and approved by the French Government Board (No. 04386.03). Animal welfare is a constant priority: animals were thus euthanized under anesthesia. No blinding was done for animal studies. The animals were randomly allocated into different groups.

To identify tumor-promoting genes *in vivo* (Fig 1D), two cell populations were subcutaneously xenografted on female NMRI nude mice flanks ($3 \times 10^6$ cells per mouse); 501Mel (6 mice) and 501Mel CRISPR-SAM cell library (10 mice, Table EV2) for Fig 1. For xenograft of other libraries (Fig 4): Vem-persister cells ($n = 11$ mice) and (PLX8394)-persister cells ($n = 12$ mice) (Table EV7). Tumor volume was assessed according to the formula $V = (W(2) \times L)/2$, during 20 weeks. After mice sacrifice, tumors were sampled and conserved at −80°C for further sgRNA enrichment analysis.

For individual validation of tumor-promoting genes (Fig 1H–K): $3 \times 10^6$ of 501Mel cells overexpressing either *SMAD3*, *BIRC3*, or *SLC9A5* were subcutaneously xenografted on female NMRI nude mice flanks (6 mice per group). In each group, mice were injected on both flanks: sgRNA 1 on right flank and sgRNA 2 on left flank (see Table EV8 for sgRNA sequences). Tumor growth was assessed as previously described (Gilot *et al*, 2017) until the endpoint (600 mm³).

In order to identify BRAFi-resistance and tumor growth genes *in vivo* (Fig 4), three cell populations were subcutaneously xenografted on female NMRI nude mice flanks ($3 \times 10^6$ cells per mice); 501Mel (6 mice), 501Mel CRISPR-SAM vemurafenib resistant (10 mice), and 501Mel CRISPR-SAM Paradox Breaker resistant (12 mice). Tumor growth was assessed as previously described (Gilot *et al*, 2017), during 20 weeks. After mice sacrifice, tumors were sampled and conserved at −80°C for further sgRNA enrichment analysis.

These experiments are compliant with all relevant ethical regulations regarding animal research.

## sgRNA enrichment analysis

Genomic DNAs from cell pellets ($>36.10^6$ cells) and tumors (> 400 mg) were extracted using the Zymo Research Quick-gDNA MidiPrep according to the manufacturer's protocol. PCR amplifications and quality controls have been done as described by Zhang laboratory (Joung *et al*, 2017).

## sgRNA sequencing

Sequencing was performed by the Human & Environmental Genomics core facility of Rennes on a HiSeq 1500 (Rapid SBS kit v2 1x100 cycles, Illumina). Base Calling was performed with Illumina's CASAVA pipeline (Version 1.8).

## Bioinformatic analysis of sgRNA and gene hits

Data processing was conducted using the MAGeCK v0.5.6 software (Li *et al*, 2014). Briefly, read counts from different samples are first median-normalized to adjust for the effect of library sizes and read count distributions (mageck count with option: norm-method median). Then, in an approach similar to those used for differential RNA-Seq analysis, the variance of read counts is estimated by sharing information across features and a negative binomial model is used to test whether sgRNA abundance differs significantly between the treatment conditions and the DMSO control. Positively or negatively selected sgRNA are ranked according to adjusted *P*-values (false discovery rate) and gene log fold changes computed with the modified robust ranking aggregation algorithm implemented in MAGeCK (mageck test with

options: norm-method median, gene-lfc-method alphamedian, andadjust method fdr).

## RNA interference

All siRNAs were transfected at 66 nM using Lipofectamine RNAiMAX (Invitrogen). For survival assay, 4,000 cells were seeded in 96-well plates, in quadruplicates. The following day, cells have been FBS-starved (1% FBS overnight). Thirty-six hours after transfection, cells were exposed to DMSO or BRAFi (vemurafenib (PLX4032); Selleckchem; 1 μM). After 84 h of treatment, cell density was measured by methylene blue assay, as previously described (Gilot et al, 2011). For RNA analysis, 50,000 cells were seeded in 12-well plates. Cells were harvested 48 h after transfection. All siRNAs were purchased from IDT DNA (Table EV8).

## Treatment experiments

8,000 (SKMel28R) or 12,000 (Me1402) cells were seeded in 96-well plates, in quadruplicates. For SMAD3i + vemurafenib combination treatment: 5,000 (501Mel) or 8,000 (M249, SKMel28R) cells were seeded in 96-well plates, in triplicates. Cells were exposed, 6 h after seeding, to either DMSO, BRAFi (vemurafenib; 5 μM), or MEKi (Cobi, 1 μM) or combination BRAFi (5 μM) + SMAD3i (0, 3, 10, 15, or 20 μM or solvent) ± MEKi (1 μM) for 4 days. 8,000 (SKMel28R) or 12,000 (Me1402) cells were seeded in 96-well plates, in quadruplicates. For Birinapant treatment (96 h): cells were exposed, 6 h after seeding, to either DMSO or Birinapant (100 nM). Cell density was evaluated using methylene blue assay, as previously described (Gilot et al, 2011).

## Half maximal inhibitory concentrations (IC$_{50}$ values)

Cell sensitivity to vemurafenib or SMAD3i (SIS3) has been established by cell density measurement and calculation of the IC$_{50}$ using GraphPad (PRISM8.0®) as previously described (Corre et al, 2018). 5,000 (501Mel) or 8,000 cells (other cell lines) were plated and exposed to inhibitor(s) at indicated concentrations for 4 days. Cells have been exposed to inhibitors 6 h after plating.

## Melanoma spheroids

Spheroids were prepared using the liquid overlay method. Briefly, 500 μl of melanoma cells (10,000 cell/ml) were added to a 24-well plate coated with 1.5% agar (Invitrogen). Plates were left to incubate for 72 h; by this time, cells had organized into 3-dimensional (3D) spheroids. Spheroids were then harvested and added into 1 ml of a solution of collagen I (2 mg/ml – Corning) with MEM 1X (Gibco), acetic acid 0.02N and neutralization buffer (HEPES 200 mM pH 7.4; sodium bicarbonate 2.2%; NaOH 0.2 N). The suspension was then added to a 12-well plate coated with 1.5% agar. Normal medium was overlaid on top of the solidified collagen after 2 h of incubation. After 48 h, medium was renewed. Pictures of the invading spheroids were monitored at different times using a Zeiss microscope.

## RNA extraction and RT–qPCR expression

Experiments have been done as previously described (Gilot et al, 2017). Primers used for RT–qPCR experiments are available in Table EV8.

## Western blot experiments

Experiments were performed as previously described (Gilot et al, 2017). Membranes were probed with suitable antibodies and signals were detected using the LAS-3000 Imager (Fuji Photo Film). The primary and secondary antibodies are described in Table EV8.

## SMAD-luciferase assay

Experiments are based on the SMAD-responsive element: Cignal Lenti SMAD Reporter (luc) (CLS-017L from Qiagen) and as control the Cignal Lenti Negative Control (luc). The SMAD-responsive luciferase vector encodes the Firefly luciferase reporter gene under the control of a minimal (m)CMV promoter and tandem repeats of the SMAD Binding Element (SBE). Melanoma cells line were infected as previously described (Gilot et al, 2017), and infected cells were selected using antibiotic selection (puromycin 2 μg/ml, 7 days). Cells were exposed to TGFβ ± SMAD3i as detailed in Fig 5K legend. Luciferase assays were then performed with a Promega kit according to the manufacturer's instructions. Data were expressed in arbitrary units, relative to the value of luciferase activity levels found in TGFβ-exposed cells, arbitrarily set at 100%. Firefly luciferase activity was normalized to protein content using Bicinchoninic Acid Kit from Sigma-Aldrich® and measured with using a luminometer (Centro XS3 LB960, Berthold Technologies).

## Immunohistochemistry

Tissue samples from representative lesions (preprint: Marin-Bejar et al, 2020) were collected and fixed in 4% paraformaldehyde for 24 h and then processed for paraffin embedding (HistoStar™ Embedding Workstation). Sections of 5 μm of thickness obtained from the paraffin-embedded tissues (Thermo Scientific Microm HM355S microtome) were mounted on SuperFrost™ Plus Adhesion slides (Thermo Scientific).

## Immunofluorescence with and without tyramide signal amplification

Antibody targeted SMAD3 was used for detecting the following protein: (rabbit, 1:100, Cell Signaling Technologies, #9513, incubation for 30 min at RT).

Furthermore, the Akoya Opal Polaris 7 Color Automation IHC Detection Kit (Akoya, NEL871001KT) was used for the tyramide signal amplification according to the manufacturer's protocol. All dewaxing and staining steps were performed using the Leica Bond RX slide stainer with the standard settings of the Opal 7-color (v5.2 plus) IHC protocol provided by the manufacturer. The steps specific for the staining is as follows: Antigen retrieval was performed for 20 min at 100°C with Bond TM Epitope Retrieval 1 solution (Leica, AR9961). Tissue was blocked for 15 min using a blocking buffer (TBS with 1% BSA (VWR, 22013, bovine serum albumin (BSA), fraction V, biotium (50 g))) with 10% normal goat serum (Invitrogen, 10000C). For introduction of the secondary-HRP, the Envision+/HRP goat anti-Rabbit (Dako Envision + Single Reagents, HRP, Rabbit, Code K4003) was used for the antibody raised in rabbit (SMAD3). The protein SMAD3 was detected using the OPAL 690 reagent.

After the staining in the Leica Bond RX, slides were washed twice for 2 min in TBS-Tween-20 (0.5%) and were then mounted using ProLong Diamond Antifade Mountant (Thermo Fisher, P36961). Numbers of count cells are indicated in Appendix Fig S2 source data file.

### *In silico* analyses

Heat maps were generated with R-packages heatmap3 (Zhao *et al*, 2014).

Gene Set Enrichment Analysis was performed using the Broad Institute software.

SKCM TCGA expression data were obtained using OncoLNC portal (http://www.oncolnc.org; (Anaya, 2016)). Cell state categorization into four differentiation states (Undifferentiated, Neural crest-like, Transitory, Melanocytic) of SKCM TCGA tumors and were performed using expression data of previously established gene sets (Tsoi *et al*, 2018).

Genomic alterations of SKCM TCGA tumors were analyzed using cBioPortal (http://www.cbioportal.org).

CCLE cell lines expression data and $IC_{50}$ were obtained from GSE36139 and from the original publication (Barretina *et al*, 2012), respectively.

Expression data of 501Mel were obtained from our previously published RNA-seq (GSE95589) (Gilot *et al*, 2017).

Expression data of M229, M238, and SKMel28 at different resistance steps were obtained from GSE75313 (Song *et al*, 2017).

Patient's median expression of our 18 hits in baseline *vs* relapse (BRAFi) is based on expression data from GSE65186 (Hugo *et al*, 2015b) and GSE50509 (Rizos *et al*, 2014).

Expression data of invasive *vs* proliferative cell lines were obtained from GSE60666 (Verfaillie *et al*, 2015).

Analysis of our 18 hits expression in EGFR-positive sorted cells was realized based on GSE97682 (Shaffer *et al*, 2017).

Analysis of *SMAD3*, *BIRC3*, and *EGFR* expression among 52 cell lines previously categorized as Undifferentiated, Neural crest-like, Transitory, or Melanocytic were performed using http://systems.c rump.ucla.edu/dediff and GSE80829.

SMAD3 ChIP-Seq data have been obtained from GSE92443 (Ramachandran *et al*, 2018).

The comparison of the median SMAD3 signature in anti-PD-1 responsive *vs* resistant patients has been performed *via* expression data from GSE78220 (Hugo *et al*, 2016).

SMAD3 signature median expression was compared to melanocytic one (*MITF, OCA2, MLANA, TYR, DCT*) and mesenchymal one (52 genes; (Mak *et al*, 2016)) *via* expression data from SKCM TCGA obtained from OncoLNC.

The comparison between median expression of SMAD3 signature with classical, proneural, and mesenchymal ones in a cohort of glioblastoma patients was based on expression data from GSE103366 (Jin *et al*, 2017).

Gene Set Enrichment Analysis on Huh28 ± TGF-β was realized based on GSE102109 (Merdrignac *et al*, 2018). Gene Set Enrichment Analysis on 3sp cells (mesenchymal) *versus* 3p cells (epithelial) was realized based on GSE26391 (van Zijl *et al*, 2011).

Single-cell RNA-seq data of a BRAF-mutant PDX model undergoing BRAF and MEK inhibition were retrieved (GSE116237). The 674 single cells were projected into a two-dimensional space using

**The paper explained**

**Problem**

Precision medicine has greatly improved the survival of patients with cutaneous melanoma, but relapse limits the long-lasting responses. Relapse is explained by the capability of persister cells to survive despite targeted therapies; however, the molecular mechanism conferring resistance is not fully understood

**Results**

We performed a genetic screen (CRISPR activation) to identify genes conferring resistance to BRAF inhibitors, promoting survival of persister cells and relapse. The current study finds that genes that enable tumor growth of therapy-naïve melanoma cells are also enriched during acquisition of resistance to BRAFi. SMAD3 targeted genes (SMAD3-signature) could be useful to find subpopulations of pre-existing BRAFi-resistant cells within therapy-naïve lesions. Moreover, chemical inhibition of SMAD3 decreases persister cells population.

**Impact**

We established in a clinical context that a high expression level of several tumor-promoting genes fuels cancer cell plasticity, therapy resistance, and relapse. By highlighting the role of tumor-promoting genes, this work expands our understanding of the biology underlying tumor growth. Moreover, we identify novel drug vulnerabilities that can be exploited to develop long-lasting anticancer therapies.

t-distributed stochastic neighbor embedding (tSNE) and colored according to their drug-tolerant state (DTC) identity (Rambow *et al*, 2018). In a second step, the activity of the resistance gene expression signature (18 genes) was quantified per cell using the AUCell algorithm (Aibar *et al*, 2017) resulting into an AUCell score (0 < range<0.4), which was used to gradient-color the tSNE plot. Finally, the activity of the resistance gene signature was quantified per DTC population using GraphPad (****$P < 0.0001$, Mann–Whitney test).

### Statistical analyses

Data are presented as mean ± s.d. unless otherwise specified, and differences were considered significant at a *P*-value of less than 0.05. In the box plots, the line within the box is the median, the bottom and top of the box are, respectively, the first and the third quartiles, and the whiskers represent the minimum and maximum of all the data points. Comparisons between groups normalized to a control were carried out using bilateral Student's test (with non-equivalent variances). When at least two factors were compared between two groups, a two-way ANOVA (without adjustment) was used as specified in figure legends. The statistical significance between two independent groups of patient samples was examined using the Mann–Whitney test. All statistical analyses were performed using Prism 8 software (GraphPad, La Jolla, CA, USA) or Microsoft Excel software. All experiments were performed three or more times independently under similar conditions, unless otherwise specified in the figure legends and statistics table containing the raw data (Appendix Fig S2–S3 source data) and Appendix Table S1 (*P*-values).

# Data availability

The datasets generated during current study are available: https://www.ebi.ac.uk/arrayexpress/experiments/E-MTAB-8595/.

**Expanded View** for this article is available online.

## Acknowledgements
The authors thank the Gene Expression and Oncogenesis team from the CNRS UMR6290 especially M. Migault, A. Forestier, and L. Boussemart; the Rennes FHU CAMIn team; and E. Watrin and C. Hitte for providing scientific expertise. The authors acknowledge the SFR Biosit core facilities of Rennes 1 University with the ARCHE animal housing facility, the cell culture L3 facility, and the Human and Environmental Genomics platform for their help and support. This study received financial support from the following: Fondation ARC pour la Recherche (ARC labellized team); Ligue Nationale Contre le Cancer (LNCC); Départements du Grand-Ouest; Région Bretagne; University of Rennes 1; CNRS; SFR Biosit, Association Vaincre le Cancer. Further support was provided by a "Ligue Nationale Contre le Cancer" (LNCC) Grand Ouest fellowship (AG) and from the Région Bretagne (AG), and Fondation ARC pour la Recherche (AG) and the LNCC (AQ) and from French Ministry of Research (NT and DL) and from Institut National contre le Cancer (INCa) (AP). The authors are grateful to Feng Zhang for providing the Human CRISPR 3-plasmid activation pooled library (SAM) (Addgene).

## Author contributions
Conceptualization: AG and DG. Methodology: AG, LB, AMQ, MA, and DG. Software: AG, MA, SC, CC, and FR. Formal analysis: AG, LB, AMQ, MA, DL, SC, OM-B, CC, and FR. Investigation: AG, LB, AMQ, AP, MA, DL, SC, AP, HML, OM-B, NT, and DG. Writing-original draft: AG and DG. Writing-review and editing: AG, DG, LB, AMQ, SC, M-DG, FR, MA, CC, and J-CM. Visualization: AG, LB, AMQ, MA, SC, FR, CC, and DG. Supervision: DG. Project Administration: DG and M-DG. Funding: DG, SC, and M-DG.

## Conflict of interest
The authors declare that they have no conflict of interest.

## For more information
- The Graeber laboratory provides a webtool to investigate the expression profile of your favorite RNAs in 53 melanoma cell lines, classified in four categories based on the differentiation status.
- https://systems.crump.ucla.edu/dediff/

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
