## [Review Process File · EMBO Molecular Medicine]

CRISPR screens identify tumor-promoting genes conferring melanoma cell plasticity and resistance

Arthur Gautron, Laura Bachelot, Marc Aubry, Delphine Leclerc, Anais Quemener, Sébastien Corre, Florian Rambow, Anais Paris, Nina Tardif, Heloise Leclair, Oskar Martin-Bejar, Cedric Coulouarn, Jean-Christophe Marine, Marie-Dominique Galibert, and David Gilot

DOI: [10.15252/emmm.202013466](https://doi.org/10.15252/emmm.202013466)

Corresponding authors: David Gilot (david.gilot@univ-rennes1.fr) , Marie-Dominique Galibert (mgaliber@univ-rennes1.fr)

Review Timeline:

Transfer from Review Commons:	16th Sep 20
Editorial Decision:	18th Sep 20
Revision Received:	15th Jan 21
Editorial Decision:	29th Jan 21
Revision Received:	9th Feb 21
Accepted:	11th Feb 21

Editor: Lise Roth

Transaction Report:

Review
COMMONS

This manuscript was transferred to EMBO Molecular Medicine following peer review at Review Commons.

Review #1

1. How much time do you estimate the authors will need to complete the suggested revisions:

Estimated time to Complete Revisions (Required)

(Decision Recommendation)

Between 3 and 6 months

2. Evidence, reproducibility and clarity:

Evidence, reproducibility and clarity (Required)

Here Gautron and colleagues performed an in vivo gain-of-function CRISPR screen to identify genes that enable xenograft growth of an otherwise poorly tumorigenic melanoma cell line (501Mel). By using published transcriptomes, authors found that some of these genes (SMAD3, BIRC3, SLC9A5) are also modulated (at the mRNA level) during acquisition of resistance to BRAF inhibitors (BRAFi), which is associated with dedifferentiation in some cases. These genes are also enriched in BRAFi-persister cells in vitro and further enriched after growth of these cells in mice. Authors then focus on SMAD3, since they find that an SMAD3 transcriptional signature promotes a mesenchymal phenotype by upregulating expression of known BRAFi-resistant genes (EGFR, AXL). This SMAD3 signature is found in dedifferentiated melanoma cell lines and in a subset (20%) of treatment-naïve melanoma patients and most BRAFi-resistant human tumours. Chemical inhibition of SMAD3 combined with BRAFi impairs survival of BRAFi-resistant cells in vitro. This can also be achieved through inhibition of aryl hydrocarbon receptor (AhR, described previously by the authors to contribute to BRAFi-resistance (Corre 2018 Nat Commun)), since AhR drives SMAD3 expression. The study is very interesting and with potential preclinical implications. In general, the data is well presented and, in most cases, adequately replicated. Most of the work is performed in a limited number of cell lines (mostly 501Mel, in addition to SKMEL28 and Mel1402), even though authors make use of published expression data to support most of the conclusions. However, there are a number of limitations that should be considered. The current study focuses almost exclusively on BRAFi and BRAFi-resistance, although currently the standard of care is combined BRAFi+MEKi. Even though some analyses are performed on published transcriptomes of BRAFi+MEKi-resistant tumours (Fig.3), the study would be more preclinically relevant if validating key conclusions using the combination. Authors propose that expression of SMAD3 signature as a potential biomarker of resistance. The study suggests that this SMAD3-signature could be useful to find sub-populations of pre-existing BRAFi-resistant cells within therapy-naïve lesions. However, it does not provide evidence on stainings of relevant hits on human samples. The manuscript hints that SMAD3 inhibitor eliminates BRAFi-resistant cells, especially when combined with BRAFi. However, it appears that this SMAD3 inhibitor as monotherapy is much more potent in decreasing survival of

BRAFi-sensitive cells than BRAFi-resistant cells (Fig.5M), which raises concerns on potential toxicities. It is unclear whether this therapy would affect normal cells (melanocytes for instance). Whether this treatment would work in vivo to eradicate MAPKi-resistant tumours is also unknown. **Specific comments:** 1.Fig.1G: there is only 1 measurement "before xenograft" but for "in tumor" there are quite a few from the different xenografts. Is this measurement of sgRNA counts in vitro reproducible/consistent enough to consider only one? Similar comment for Fig.4E. 2.Fig.2: since authors used 2 BRAFi (paradox breaker and vemurafenib), hits are common genes in both treatments? It is not clear from legend or Methods. Were the 2 treatments very different in terms of hits? As said below, information in Supplementary Tables is a bit difficult to interpret due to lack of some important details. 3.Fig.3A: text says that majority of candidates were over-expressed in SDR. However, BIRC3 is not overexpressed in the SDR of M229 and M238 (while it was upregulated in the DTP and DTPP states), although it is upregulated in SKMEL28-SDR and DDR. This should be discussed. Also, how about in the other cell lines from the Song 2017 study (M395, M397, M249) for which there is 2day, DTP, DTPP, SDR data? Did authors analyze protein levels of candidates (BIRC3..) as they did for SMAD3 in Fig.5L? 4.Fig.3F-G: graph should show all the patients (as in S2D-E), not only the ones in which there is an increase, otherwise it is misleading since it seems that all patients in the cohort had upregulation. Text says "majority of drug resistant patients", but it would be more accurate to state the percentage of patients with upregulation in resistant sample vs baseline. Last sentence of first paragraph in pag.12 seems a bit too strong when it appears to conclude from just a correlation between expression and resistance state ("supporting their involvement in establishing drug tolerant and/or resistant phenotypes in vivo") 5.Fig.4A: which BRAFi was used here? It is not clear. Legend says 12 mice were xenografted. However, in Methods pag.48, authors mention different BRAFi and different mouse cohorts "...three cell populations were subcutaneously xenografted on female NMRI nude mice flanks (3x10⁶ cells per mice); 501Mel (6 mice), 501Mel CRISPR-SAM vemurafenib resistant (10 mice), 501Mel CRISPR-SAM Paradox Breaker resistant (12 mice)". 6.Fig.4B: despite the interesting result, it appears that these tumours were grown in the absence of BRAFi. Even if the cell lines were resistant in vitro to BRAFi, it should have been demonstrated that they are able to grow in vivo under continuous BRAFi treatment, showing persistent resistance. Especially since tumours took quite a long time to develop, between 1 and 4 months, similar to the therapy-naïve setting (Fig.1D). Graph in 4B should also number the tumours (as they are identified in Table S7), it would be informative to analyze if tumours arising at 4 weeks had a different sgRNA enrichment vs tumours arising later (i.e. 12-16 weeks). 7.When referring to Mel1402, authors say in page 14 that they are intrinsically resistant to BRAFi, compared to acquired resistance of SKMEL28R. However, graphs 5G-H show that in both cases BRAFi decreased cell density by 50% in both cases, which is quite substantial. Was this 5uM? It is not stated in legend, but it is quite high. Was this the concentration used to make them resistant? It is not clear from Methods. As said above for original M229 and M238-resistant derivatives, 1uM was used (Nazarian et al 2010 Nature), as in other papers in the field (Obenauf et al 2015 Nature, Wang et al 2018 Cell, among other studies). If BRAFi treatment was 84h, it should be stated also in the legend (also in other figures). 8.Fig.5I: this is in mel1402, but the knockdown in SKMEL28R (Fig.5G) is not shown. Same for Fig.S3F and S3G and corresponding knockdown. 9.Fig.5J: text says that SMAD3i "strongly" decreased levels of p-SMAD3 induced by TGFbeta. However, from the blot it does not seem very strong. Do authors have a

quantification from different blots? In line with this, authors should show p-SMAD3 blots from experiment in Fig.5K, and p-ERK and p-SMAD3 for Fig.5M to show that drugs are acting on-target. 10.Fig.5L: was p-SMAD3 also increased? Or is it all due to increased total protein? 11.Fig.5M: these experiments need a proper quantification (dose-response, IC50, synergy analyses) and including more BRAFi-resistant lines, since in the current experiment there is only one (SKMel28R). Comparing the pairs that authors have (M229S/R, M238S/R, SKMEL28S/R) would be much more informative, since they usually are sensitive=differentiated, resistant=undifferentiated. And as mentioned above, with current data, SMAD3 seems to be more potent in reducing viability of BRAFi-sensitive than BRAFi-resistant when used as monotherapy, which raises concerns over specificity and dependency of resistant cells on this pathway. Also, 5uM BRAFi is quite high and may obscure synergy effects. For example, Fig.5A used 2uM, as in Lito 2012 Cancer Cell. Or even 1uM, as said above original M229 and M238 resistant derivatives (used in this study) were isolated after 1uM vemurafenib chronic treatment (Nazarian 2010 Nature). 12.Fig.6B: here authors show induction of SMAD3 after treatment with AhR ligand for 10 days. Fig.6C-D experiments were for 7 days. It is unclear why such a long time was needed to show induction of transcription. In their previous paper Corre 2018 Nat Commun, authors showed transcriptomic changes after 48 h TCDD treatment in 501Mel cells. Was SMAD3 upregulated here? 13.Fig.7B: text says that SMAD3 depletion further validated MMP2, AXL, EGFR and JUNB as SMAD3-regulated genes. Since the list was comprised of 9 genes, to clarify, were the other genes not tested or not regulated upon SMAD3 knockdown? 14.Fig.7D: SMAD3-signature inducibility is higher in differentiated cells However, basal expression is higher in dedifferentiated cells, which should be discussed. However, dedifferentiated cells here only include the M238 pair (S and R). Did authors compare S and R of the pairs in which dedifferentiation increases from S to R (SKMEL28 and M229 pair, Fig.5L)? It would be a more appropriate comparison. 15.Regarding statistical analyses, some graphs should be revised and adjusted for multiple comparisons instead of t-test (using ANOVA or equivalent), like 5G-H, 6B-D, 7C,E. Also, in some cases like 6B-D, 7C,E it is unclear which comparison is being made, since there is a line on top of 2 bars. For example, 6B, line on top of TCDD and ITE, 2 asterisks vs dms0, does this mean that each comparison DMSO vs TCDD and DMSO vs ITE is 2 asterisks? While methods say that Anova was used in some cases and specified in figure legends, this is not found in legends. **Minor comments:** Some Supplementary Tables lack an explanation/legend of what the table and data show and the reader has to guess. For example, Table S7, what do numbers in each column mean? Is PB paradox breaker and V vemurafenib? Or Table S6, what is 0, 1, 2..? sgRNA counts? The other tables should be revised accordingly. Pag.7: when speaking about 501mel are unable to generate tumors in mice, it should say nude mice (used in ref.29) instead of immunocompromised. Pag.11: differentiated cell line M229 (melanocytic).. However, Fig.5L says "T" for transitory. Fig.1K. BIRC3 tumors seem to be delayed (day 11) compared to the other two, especially SLC9A5 (day 4-5). This should be discussed given the strong phenotype of SLC9A5 even with a moderate overexpression (average 2fold, Fig.1J) compared to the other two (6- and 20-fold). Legend should specify how many mice were injected with each cell line. Fig.2E: when referring to this dataset, it should be mentioned that these lines are therapy-naïve (never treated with drug) but they are intrinsically (partially) resistant to BRAFi when exposed to the drug, to distinguish from acquired resistance models (M229, M238..). Fig.2F: the axis is missing the numbers. Perhaps this should go in supplementary, or just indicate in the legend or elsewhere which quadrant of the graph shows the hits. Second

paragraph in page 14 seems out of context, it should go earlier in page 13 when describing that SMAD3 high in dedifferentiated cells. Pag.17: when describing Hugo 2016 Cell patient samples, it should be specified that these are pre-treatment biopsies from responders and non-responders to anti-PD-1 treatment. Fig legend should be corrected too ("melanoma exposed to PD-1 therapy".. these samples were not exposed to anti-PD-1, they were collected before treatment). Therefore, higher SMAD3 signature would identify anti-PD1 non-responders Fig.7I: text mentions that "comparing the Smad3-signature with the classical mesenchymal-like signature of melanoma (TCGA cohort) highlighted a significant correlation" but there is no correlation analysis here. So currently it would be an "overlap" or similar. In discussion p.20 when referring to the transactivation obtained by CRISPR-SAM as "massive" (Fig.1I) authors should avoid using these subjective adjectives, especially since expression levels were not quantified. SLC9A5 is defined in p.21 but this should go the first time SLC9A5 is described in the paper. Perhaps Table S8 is not needed since the list of genes (9) is already mentioned in the text (page 16) Methods should specify formula used to calculate tumour volume. Authors should be commended for the detailed CRISPR-SAM protocol in Supplementary methods, it will be very useful in order for others to replicate/use this technology. All relevant prior studies are properly referenced.

3. Significance:

Significance (Required)

The study builds upon previous findings of the group describing the role of AhR in BRAFi-resistance (Corre 2018 Nat Commun). The current study finds that the same genes that enable tumour growth of therapy-naïve melanoma cells are also enriched during acquisition of resistance to BRAFi, given that SMAD3 in particular is a target of AhR. The study suggests that this SMAD3-signature could be useful to find sub-populations of pre-existing BRAFi-resistant cells within therapy-naïve lesions, which should be evaluated. This study would be important for researchers working on melanoma and also potentially on other cancers driven by EMT phenotype switching. The study hints that this SMAD3-signature could be operative in subsets of glioblastoma tumours, and also in the context of anti-PD-1 in melanoma. Field of expertise Melanoma, therapy resistance, invasion and metastasis, cytoskeleton.

Review #2

1. How much time do you estimate the authors will need to complete the suggested revisions:

Estimated time to Complete Revisions (Required)

(Decision Recommendation)

Between 1 and 3 months

2. Evidence, reproducibility and clarity:

Evidence, reproducibility and clarity (Required)

Gautron and colleagues search for novel genes that promote phenotypic plasticity and drug resistance in melanoma, discovered by in vivo CRISPR screens. The subject is highly topical, the technology is cutting edge, the presentation is excellent, and the work is technically sound. My critiques are relatively minor. Side effects and toxicity: A key finding of this paper is SMAD3 as a potential therapeutic target in BRAFi-resistant melanoma. However I did not find any discussion of how SMAD3-inhibition might result in toxicities or side effects in a treated individual. How widely-expressed is SMAD3? Is anything known about the effects of inhibition to a human or mouse? Could any strategies mitigate such toxicity? Figure 1C: If I understand correctly, the authors present here the correlation of sgRNA counts between replicates. This is not a valid measure of experiment quality. The authors should present the correlation of FOLD CHANGES. See Hanna and Doench (PMID: 32284587) Box2 for an explanation why. p8 "Thirty-six other genes were recurrently retrieved in the tumors but not in all". Here and elsewhere the authors are rather tough with candidates that do not appear universally across all replicates. Were any of these candidates validated? It could be a rich source of additional drug targets, and perhaps even ones for where small molecule inhibitors already exist and may give less side effects than SMAD3 etc. Degree of activation by CRISPRa: The method gives highly variable degrees of activation. This is shown in the literature where fold changes range from 1.5 to many hundreds. Can the authors comment on how much this might affect their results? Figure 1J shows that SMAD3 gives very high activation. Could this partially explain why it consistently appears in the screens? Could this also explain why the ignored other candidates (in above comment) appear in less than all the screens? Could it also explain why some of the 3 sgRNAs for each candidate do not appear as hits in the screen (because they do not activate expression strongly enough)? I'd encourage the authors to discuss this in the paper somewhere, and even consider a few more validation experiments (qRTPCR) to look at fold activation by the various sgRNAs for the target genes, and check if it correlates with those sgRNAs enrichment in the screen. Targeting AhR and SMAD3 by small molecules: The authors discuss some issues with the small molecule inhibitors they used. Why not simply design antisense oligonucleotides / gapmers targeting these mRNAs? Significance of AhR: I was a little puzzled about how AhR fits with the SMAD3 story. Likely this could be fixed by some explanation at an appropriate point in the paper, about the motivation for looking into AhR. For example, is it because AhR would have some advantage as a drug target over SMAD3 itself? Or is it simply because the authors studied this gene previously?

3. Significance:

Significance (Required)

My expertise: cancer and CRISPR screening.

Reply to the reviewers

Rebuttal_ Preprint RC-2020-00368

We thank the editor for handling our manuscript and both reviewers for their constructive critiques. We provide below a detailed list of results already available and experiments we propose to perform to address the reviewers' comments and improve the quality of our manuscript.

Reviewer #1 (Evidence, reproducibility and clarity (Required)):

Here Gautron and colleagues performed an in vivo gain-of-function CRISPR screen to identify genes that enable xenograft growth of an otherwise poorly tumorigenic melanoma cell line (501Mel). By using published transcriptomes, authors found that some of these genes (SMAD3, BIRC3, SLC9A5) are also modulated (at the mRNA level) during acquisition of resistance to BRAF inhibitors (BRAFi), which is associated with dedifferentiation in some cases. These genes are also enriched in BRAFi-persister cells in vitro and further enriched after growth of these cells in mice.

Authors then focus on SMAD3, since they find that an SMAD3 transcriptional signature promotes a mesenchymal phenotype by upregulating expression of known BRAFi-resistant genes (EGFR, AXL). This SMAD3 signature is found in dedifferentiated melanoma cell lines and in a subset (20%) of treatment-naïve melanoma patients and most BRAFi-resistant human tumours. Chemical inhibition of SMAD3 combined with BRAFi impairs survival of BRAFi-resistant cells in vitro. This can also be achieved through inhibition of aryl hydrocarbon receptor (AhR, described previously by the authors to contribute to BRAFi-resistance (Corre 2018 Nat Commun)), since AhR drives SMAD3 expression.

The study is very interesting and with potential preclinical implications. In general, the data is well presented and, in most cases, adequately replicated. Most of the work is performed in a limited number of cell lines (mostly 501Mel, in addition to SKMEL28 and Mel1402), even though authors make use of published expression data to support most of the conclusions. However, there are a number of limitations that should be considered.

We thank this reviewer for his/her positive comments on our work and for the suggestions made to improve its relevance

Review#1 point 1: *The current study focuses almost exclusively on BRAFi and BRAFi-resistance, although currently the standard of care is combined BRAFi+MEKi. Even though some analyses are performed on published transcriptomes of BRAFi+MEKi-resistant tumours (Fig.3), the study would be more preclinically relevant if validating key conclusions using the combination.*

Response 1.1. We agree that the combo (BRAFi+MEKi) is the gold-standard. So, in the revised version, we will validate the key conclusions using this combo (BRAFi+MEKi), especially in Figure 5.

Review#1 point 2: *Authors propose that expression of SMAD3 signature as a potential biomarker of resistance. The study suggests that this SMAD3-signature could be useful to find sub-populations of pre-existing BRAFi-resistant cells within therapy-naïve lesions. However, it does not provide evidence on stainings of relevant hits on human samples.*

Response 1.2. We will perform the SMAD3 staining in therapy-naïve lesions as suggested in agreement with our collaborative's skills (J.C. Marine). We hope to add these data in the revised version. Tumor slices are already available (VIB) and antibodies are ordered. David Nittner, from the VIB-Leuven IHC/histopathology expertise center, will perform the SMAD3 staining.

Review#1 point 3: *The manuscript hints that SMAD3 inhibitor eliminates BRAFi-resistant cells, especially when combined with BRAFi. However, it appears that this SMAD3 inhibitor as monotherapy is much more potent in decreasing survival of BRAFi-sensitive cells than BRAFi-resistant cells (Fig.5M), which raises concerns on potential toxicities. It is unclear whether this therapy would affect normal cells (melanocytes for instance). Whether this treatment would work in vivo to eradicate MAPKi-resistant tumours is also unknown.*

Response 1.3. We agree that an *in vivo* toxicity of SMAD3i (alias SIS3) on normal cells may impair the use of this inhibitor for therapy. We selected this inhibitor accordingly to the literature. The most convincing manuscript, published in Nature Communications in 2015 (Smad3 promotes cancer progression by inhibiting E4BP4-mediated NK cell development), demonstrated that inhibition of Smad3 prevents cancer progression. Tang et al., used the B16F10-luc melanoma model. These cells were s.c. inoculated into mice and mice were exposed to various dosages of SIS3 (0, 2.5, 5.0 or 10 mg.g-1.day-1, i.p.). Interestingly, the antitumoral effect of SIS3 is due to the decrease of B16F10-luc cell proliferation (dose-dependent effect of SIS3) and an increase of NK cell anti-cancer cytotoxicity. Authors did not report toxicities despites 15 days of treatment (SMAD3i).

In addition, an upgraded SIS3 version was published in 2020 (Discovery of a novel selective water-soluble SMAD3 inhibitor as an antitumor agent). So, we believe that this second generation of SMAD3i will be more appropriate for *in vivo* usage. This molecule is not yet commercially available. Authors did not report toxicities for these two generations of SMAD3i despites 20 days of treatment (SMAD3i).

However, to evaluate a potential effect of SMAD3i on normal cells, we will expose normal melanocytes in primary culture to SMAD3i (dose -response) to define the therapeutic dose-range. These experiments will be added to the revised version. NHEMs will be purchased and inhibitors are available.

Here, we use the SIS3 inhibitor to challenge our hypothesis. The *in vivo* ability of SMAD3 inhibitor to reduce the tumor growth and or the relapse will be investigated in another study (BRAFi+MEKi +/- SMAD3i). This type of experiments has to demonstrate that relapse is delayed or suppressed as published by Rambow F. in Cell (2018) and others. It is important to keep in mind that melanoma PDX-exposed to BRAFi+MEKi relapse after ~50 days (and we must monitor the relapse at least ~100 (120) days to observe the protective effect of the tri-therapy). Moreover, a preliminary study is needed to establish the best *in vivo* protocol and the evaluation of SIS3 toxicity (increasing range, acute toxicity). In addition, the water-soluble SMAD3 inhibitor is not yet commercially available.

To conclude, we believe that this *in vivo* experiment is not compatible with a rapid publication of our manuscript identifying AhR-SMAD3 pathway as a critical pathway in BRAFi-resistance. **We hope to submit the revised version of this manuscript before the end of the 2020.**

****Specific comments:****

Review#1 point 4: *1.Fig.1G: there is only 1 measurement "before xenograft" but for "in tumor" there are quite a few from the different xenografts. Is this measurement of sgRNA counts in vitro reproducible/consistent enough to consider only one? Similar comment for Fig.4E.*

Response 1.4. As mentioned in the manuscript, we performed two experiments and we merged the sgRNA counts (corresponding to “the before xenograft = cell library”). In the revised version, we will show the two experiments in these two items (data available).

Review#1 point 5: 2.Fig.2: since authors used 2 BRAFi (paradox breaker and vemurafenib), hits are common genes in both treatments? It is not clear from legend or Methods. Were the 2 treatments very different in terms of hits? As said below, information in Supplementary Tables is a bit difficult to interpret due to lack of some important details.

Response 1.5. We recognize that it can be difficult to compare the results obtained with these two inhibitors. Data are already available (Table S5, page 9 of Supplementary information). In the revised version, we will add a clear comparison (supplementary figure).

Review#1 point 6: 3.Fig.3A: text says that majority of candidates were over-expressed in SDR. However, BIRC3 is not overexpressed in the SDR of M229 and M238 (while it was upregulated in the DTP and DTPP states), although it is upregulated in SKMEL28-SDR and DDR. This should be discussed. Also, how about in the other cell lines from the Song 2017 study (M395, M397, M249) for which there is 2day, DTP, DTPP, SDR data? Did authors analyze protein levels of candidates (BIRC3..) as they did for SMAD3 in Fig.5L?

Response 1.6. In the revised version of the manuscript, we will discuss the fact that BIRC3 is not overexpressed in all the samples as observed for different genes by Song (2017)(Fig. 3A). In fact, BIRC3 is induced in M229 in response to BRAFi at SDR stage (FC = 2.87 but it is difficult to read the blue scale). So, we will add a graph showing the fold-change of BIRC3 expression (Resistant versus Parental cells; we will separate the Rx and Ra cell lines). This graph will include the other available data as suggested.

In addition, we attempted to detect BIRC3 protein by western-blot but we failed with the antibody raised against c-IAP2 clone (58C7) (rabbit ref 3130S). Since the protein samples are already available, we will try another antibody to show BIRC3 expression levels in Fig.5L.

Review#1 point 7: 4.Fig.3F-G: graph should show all the patients (as in S2D-E), not only the ones in which there is an increase, otherwise it is misleading since it seems that all patients in the cohort had upregulation. Text says “majority of drug resistant patients”, but it would be more accurate to state the percentage of patients with upregulation in resistant sample vs baseline. Last sentence of first paragraph in pag.12 seems a bit too strong when it appears to conclude from just a correlation between expression and resistance state (“supporting their involvement in establishing drug tolerant and/or resistant phenotypes in vivo”)

Response 1.7. We agree with the reviewer. These data were available in the submitted version of the manuscript (supp Fig. 2). We will move these items on Fig. 3 and we will indicate the percentage of patients with an upregulation of the signature as suggested. Moreover, we will remove this overstatement (page 12) in the revised version.

Review#1 point 8: 5.Fig.4A: which BRAFi was used here? It is not clear. Legend says 12 mice were xenografted. However, in Methods pag.48, authors mention different BRAFi and different mouse cohorts “...three cell populations were subcutaneously xenografted on female NMRI nude mice flanks (3x10⁶ cells per mice); 501Mel (6 mice), 501Mel CRISPR-SAM vemurafenib resistant (10 mice), 501Mel CRISPR-SAM Paradox Breaker resistant (12 mice)”.

Response 1.8. We will clearly indicate the BRAF inhibitor used in each experiment to clarify the revised version of the manuscript. Moreover, we will correct the number of mice per group.

Review#1 point 9: 6.Fig.4B: despite the interesting result, it appears that these tumours were grown in the absence of BRAFi. Even if the cell lines were resistant in vitro to BRAFi, it should have been demonstrated that they are able to grow in vivo under continuous BRAFi treatment, showing persistent resistance. Especially since tumours took quite a long time to develop, between 1 and 4 months, similar to the therapy-naïve setting (Fig.1D). Graph in 4B should also number the tumours (as they are identified in Table S7), it would be informative to analyze if tumours arising at 4 weeks had a different sgRNA enrichment vs tumours arising later (i.e. 12-16 weeks).

Response 1.9. We thank this reviewer for his/her positive comments on our work and for the suggestions made to improve its relevance. In absence of BRAFi, the tumor-growth monitoring took 5 months. So, we believe that in presence of BRAFi, the tumor growth will be slower, suggesting that the advised experience could require at least 6 months (and probably more) according to the literature (cf Response 1.3.). In conclusion, we think that this experience is not compatible with a rapid publication of our manuscript (we hope to submit the revised version of this manuscript before the end of the 2020.)

As suggested, we will analyze, in the revised version, the different sgRNA enrichment in the two groups of tumors. (Data are already available, Table S7).

Review#1 point 10: 7.When referring to ME1402, authors say in page 14 that they are intrinsically resistant to BRAFi, compared to acquired resistance of SKMEL28R. However, graphs 5G-H show that in both cases BRAFi decreased cell density by 50% in both cases, which is quite substantial. Was this 5µM? It is not stated in legend, but it is quite high. Was this the concentration used to make them resistant? It is not clear from Methods. As said above for original M229 and M238-resistant derivatives, 1µM was used (Nazarian et al 2010 Nature), as in other papers in the field (Obenaus et al 2015 Nature, Wang et al 2018 Cell, among other studies). If BRAFi treatment was 84h, it should be stated also in the legend (also in other figures).

Response 1.10 SKMEL28R cell line has been established by the team of Pr D. PEEPER (NKI) (a gift to JC Marine's lab). The parental cell line was exposed to PLX-4720 during several months (up to 3 µM). Cells are routinely cultivated with 1µM of PLX-4720. ME1402 model is BRAFi-tolerant as well as SKMEL28R cells. The 5µM (BRAFi) was an arbitrarily choice for the treatment of these two cells lines (84h, Fig. 5). In the revised version, we will improve the legends by indicating the treatment conditions. Moreover, for another query (Review#1 point 14.), we will perform BRAFi dose-response in several cell lines (including ME1402) to define a synergy between SMAD3i and BRAFi. These data will be added in the revised version.

Review#1 point 11: 8.Fig.5I: this is in ME1402, but the knockdown in SKMEL28R (Fig.5G) is not shown. Same for Fig.S3F and S3G and corresponding knockdown.

Response 1.11. All the knock-down validations will be included in the revised version of the manuscript (data are already available).

Review#1 point 12: 9.Fig.5J: text says that SMAD3i "strongly" decreased levels of p-SMAD3 induced by TGFbeta. However, from the blot it does not seem very strong. Do authors have a quantification from different blots? In line with this, authors should show p-SMAD3 blots from experiment in Fig.5K, and p-ERK and p-SMAD3 for Fig.5M to show that drugs are acting on-target.

Response 1.11. We agree with the comment. We will quantify this effect (phospho-SMAD3 decrease in presence of SMAD3i on the two western-blot (already available). A quantification will be add to supplementary information and we will modify the sentence.

Review#1 point 13. *10.Fig.5L: was p-SMAD3 also increased? Or is it all due to increased total protein?*

Response 1.13. We will perform this experiment (Phospho-SMAD3 _ Fig. 5L, antibody already available). The western-blot experiments will be added to this figure.

Review#1 point 14. *11.Fig.5M: these experiments need a proper quantification (dose-response, IC50, synergy analyses) and including more BRAFi-resistant lines, since in the current experiment there is only one (SKMel28R). Comparing the pairs that authors have (M229S/R, M238S/R, SKMEL28S/R) would be much more informative, since they usually are sensitive=differentiated, resistant=undifferentiated. And as mentioned above, with current data, SMAD3 seems to be more potent in reducing viability of BRAFi-sensitive than BRAFi-resistant when used as monotherapy, which raises concerns over specificity and dependency of resistant cells on this pathway.*

Also, 5uM BRAFi is quite high and may obscure synergy effects. For example, Fig.5A used 2uM, as in Lito 2012 Cancer Cell. Or even 1uM, as said above original M229 and M238 resistant derivatives (used in this study) were isolated after 1uM vemurafenib chronic treatment (Nazarian 2010 Nature).

Response 1.14. To further illustrate the capability of SMAD3i for target cutaneous melanoma, we will perform these suggested experiments in more cell lines (3 pairs). We will determine the IC₅₀ and a possible synergy effect between BRAFi and SMAD3i (as seen in Fig. 5M (5µM BRAFi and 3µM SMAD3i). We agree that 5µM is probably too high to observe a clear synergy effect. We will test different concentrations. Cell lines and drugs are already available in the lab.

Before to conclude about the role of SMAD3i on BRAFi-sensitive and BRAFi-resistant cell lines, it seems important to compare the SMAD3 expression levels in each cell line (Fig. 5L). So, it is quite intuitive that a cell line expressing a low level of SMAD3 requires a lower SMAD3i concentration to be effective than a cell line expressing a huge amount of SMAD3 (i.e. 501Mel and SKMel28R). In contrast to the reviewer 1, we believe that it would be an advantage/opportunity that melanoma cell lines are sensitive to SMAD3. We showed (in Fig. 1A) that cutaneous melanomas are mainly differentiated (low SMAD3 expression levels suggesting a low SMAD3i dose might be effective to treat these tumors).

Review#1 point 15. *12.Fig.6B: here authors show induction of SMAD3 after treatment with AhR ligand for 10 days. Fig.6C-D experiments were for 7 days. It is unclear why such a long time was needed to show induction of transcription. In their previous paper Corre 2018 Nat Commun, authors showed transcriptomic changes after 48 h TCDD treatment in 501Mel cells. Was SMAD3 upregulated here?*

Response 1.15. In our previous study (Corre et al., 2018), we demonstrated that AhR activation by the exogenous ligand of AhR (TCDD (dioxin) promoted the transcriptional activation of AhR in 501Mel. We observed a significant induction of SMAD3 mRNA in these experimental conditions (48h; log FC = 0.43 ; pvalue 0.00213). We observed a stronger induction for a longer treatment (7 days). This point will be discussed in the revised version.

Review#1 point 16. 13.Fig.7B: text says that SMAD3 depletion further validated MMP2, AXL, EGFR and JUNB as SMAD3-regulated genes. Since the list was comprised of 9 genes, to clarify, were the other genes not tested or not regulated upon SMAD3 knockdown?

Response 1.16. We arbitrarily selected the most “important genes” from this SMAD3-signature (MMP2, AXL, EGFR & JUNB). In the revised version, we will quantify the 9 genes as suggested (samples are available).

Review#1 point 17. 14.Fig.7D: SMAD3-signature inducibility is higher in differentiated cells. However, basal expression is higher in dedifferentiated cells, which should be discussed. However, dedifferentiated cells here only include the M238 pair (S and R). Did authors compare S and R of the pairs in which dedifferentiation increases from S to R (SKMEL28 and M229 pair, Fig.5L)? It would be a more appropriate comparison.

Response 1.17. We agree. This comparison will be performed for the revised version. However, it is important to keep in mind that the differentiation status of S & R cell lines is subtle: Sensitive cell lines are not all annotated as “melanocytic” (SKMel28S, M229S and M238S are respectively M, T and NC). Conversely the Resistant cell lines are not all undifferentiated (U) ((SKMel28R, M229R and M238R are respectively U, U and NC). These results are in accordance to Fig 5B and 5C showing the SMAD3 mRNA expression levels in 53 cells lines (in function of the 4 differentiation states).

Review#1 point 18. 15.Regarding statistical analyses, some graphs should be revised and adjusted for multiple comparisons instead of t-test (using ANOVA or equivalent), like 5G-H, 6B-D, 7C,E. Also, in some cases like 6B-D, 7C,E it is unclear which comparison is being made, since there is a line on top of 2 bars. For example, 6B, line on top of TCDD and ITE, 2 asterisks vs dms0, does this mean that each comparison DMSO vs TCDD and DMSO vs ITE is 2 asterisks? While methods say that Anova was used in some cases and specified in figure legends, this is not found in legends.

Response 1.18. We agree that our graphical representation explaining the multiple comparison is not clear. We will modify these representations as suggested and we will improve the figure legends (stats).

****Minor comments.****

Review#1 point 19. Some Supplementary Tables lack an explanation/legend of what the table and data show and the reader has to guess. For example, Table S7, what do numbers in each column mean? Is PB paradox breaker and V vemurafenib? Or Table S6, what is 0, 1, 2..? sgRNA counts? The other tables should be revised accordingly.

Response 1.19. We thank the reviewer for this comment. We apologize. In the revised version, we will more explain the tables (and the columns).

Review#1 point 20. Pag.7: when speaking about 501mel are unable to generate tumors in mice, it should say nude mice (used in ref.29) instead of immunocompromised.

Response 1.20. We thank the reviewer for this comment. In the revised version, we will indicate the mouse strain (nude).

Review#1 point 21. *Pag.11: differentiated cell line M229 (melanocytic)..*" However, Fig.5L says "T" for transitory.

Response 1.21. We thank the reviewer for this comment. This point will be fixed in the revised version of the manuscript.

Review#1 point 22. *Fig.1K. BIRC3 tumors seem to be delayed (day 11) compared to the other two, especially SLC9A5 (day 4-5). This should be discussed given the strong phenotype of SLC9A5 even with a moderate overexpression (average 2fold, Fig.1J) compared to the other two (6- and 20-fold). Legend should specify how many mice were injected with each cell line.*

Response 1.22. We thank the reviewer for this comment. In the Fig. 1K, we performed a proof-of-concept experiment showing that the increase of *SMAD3* or *BIRC3* or *SLC9A5* promotes the tumor growth in contrast to parental cell line (n=6 mice/group). It is interesting to keep in mind that *SLC9A5* encodes a transporter. Thus, a moderate mRNA expression increase could significantly enhance the global activity of this transporter. This point will be discussed in the revised version of the manuscript and the legends will be improved.

Review#1 point 23. *Fig.2E: when referring to this dataset, it should be mentioned that these lines are therapy-naïve (never treated with drug) but they are intrinsically (partially) resistant to BRAFi when exposed to the drug, to distinguish from acquired resistance models (M229, M238..).*

Response 1.23. We thank the reviewer for this comment. This point will be fixed in the revised version of the manuscript.

Review#1 point 24. *Fig.2F: the axis is missing the numbers. Perhaps this should go in supplementary, or just indicate in the legend or elsewhere which quadrant of the graph shows the hits.*

Response 1.24. We thank the reviewer for this comment. This point will be fixed in the revised version of the manuscript, as suggested (supplementary information).

Review#1 point 25. *Second paragraph in page 14 seems out of context, it should go earlier in page 13 when describing that SMAD3 high in dedifferentiated cells.*

Response 1.25. We thank the reviewer for this comment. This paragraph will be moved according to the suggestion in the revised version.

Review#1 point 26. *Pag.17: when describing Hugo 2016 Cell patient samples, it should be specified that these are pre-treatment biopsies from responders and non-responders to anti-PD-1 treatment. Fig legend should be corrected too ("melanoma exposed to PD-1 therapy".. these samples were not exposed to anti-PD-1, they were collected before treatment). Therefore, higher SMAD3 signature would identify anti-PD1 non-responders*

Response 1.26. We apologize. This point will be fixed in the revised version of the manuscript (results and discussion).

Review#1 point 27. *Fig.7I: text mentions that "comparing the Smad3-signature with the classical mesenchymal-like signature of melanoma (TCGA cohort) highlighted a significant correlation" but there is no correlation analysis here. So currently it would be an "overlap" or similar.*

Response 1.27. We will provide the data (already available) illustrating the correlation in the revised version of the manuscript.

Review#1 point 28. *In discussion p.20 when referring to the transactivation obtained by CRISPR-SAM as "massive" (Fig.1I) authors should avoid using these subjective adjectives, especially since expression levels were not quantified.*

Response 1.28. We will modify this sentence in the revised version of the manuscript.

Review#1 point 29. *SLC9A5 is defined in p.21 but this should go the first time SLC9A5 is described in the paper.*

Response 1.29. We will revise this mistake in the revised version of the manuscript.

Review#1 point 30. *Perhaps Table S8 is not needed since the list of genes (9) is already mentioned in the text (page 16).*

Response 1.30. We will remove this superfluous table S8 in the revised version of the manuscript.

Review#1 point 31. *Methods should specify formula used to calculate tumour volume.*

Response 1.31. We will add the formula in the revised version of the manuscript (Methods).

Authors should be commended for the detailed CRISPR-SAM protocol in Supplementary methods, it will be very useful in order for others to replicate/use this technology. All relevant prior studies are properly referenced.

We thank this reviewer for his/her positive comments on our manuscript.

Reviewer #1 (Significance (Required)):

The study builds upon previous findings of the group describing the role of AhR in BRAFi-resistance (Corre 2018 Nat Commun). The current study finds that the same genes that enable tumour growth of therapy-naïve melanoma cells are also enriched during acquisition of resistance to BRAFi, given that SMAD3 in particular is a target of AhR. The study suggests that this SMAD3-signature could be useful to find sub-populations of pre-existing BRAFi-resistant cells within therapy-naïve lesions, which should be evaluated.

This study would be important for researchers working on melanoma and also potentially on other cancers driven by EMT phenotype switching. The study hints that this SMAD3-signature could be operative in subsets of glioblastoma tumours, and also in the context of anti-PD-1 in melanoma.

Field of expertise: Melanoma, therapy resistance, invasion and metastasis, cytoskeleton.

Reviewer #2 (Evidence, reproducibility and clarity (Required)):

Gautron and colleagues search for novel genes that promote phenotypic plasticity and drug resistance in melanoma, discovered by in vivo CRISPR screens. The subject is highly topical, the technology is cutting edge, the presentation is excellent, and the work is technically sound. My critiques are relatively minor.

We thank this reviewer for his/her positive comments on our manuscript.

Review#2 point 1. *Side effects and toxicity: A key finding of this paper is SMAD3 as a potential therapeutic target in BRAFi-resistant melanoma. However, I did not find any discussion of how SMAD3-inhibition might result in toxicities or side effects in a treated individual. How widely-expressed is SMAD3? Is anything known about the effects of inhibition to a human or mouse? Could any strategies mitigate such toxicity?*

Response 2.1. We thank the reviewer for this comment. This point will be discussed in the revised version of the manuscript. The possible “side effects and toxicity” have been also highlighted by Reviewer 1 (cf **Review#1 point 3.**).

We agree that an eventual *in vivo* toxicity of SMAD3i (alias SIS3) may impair the use of this inhibitor for therapy. We selected this inhibitor accordingly to the literature ¹⁻⁴. The most convincing manuscript, published in Nature Communications in 2015 (DOI: 10.1038/ncomms14677), demonstrated that inhibition of Smad3 prevents cancer progression. Tang et al., used the B16F10-luc melanoma model. These cells were s.c. inoculated into mice and mice have been treated with various dosages of SIS3 (0, 2.5, 5.0 or 10 mg.g-1.day-1, i.p.). Interestingly, the antitumoral effect of SIS3 is due to the decrease of B16F10-luc cell proliferation (dose-dependent effect of SIS3) and an increase of NK cell anti-cancer cytotoxicity. **Authors did not report toxicities despite 15 days of treatment (SMAD3i).**

In addition, an upgraded SIS3 version was published in 2020 (Discovery of a novel selective water-soluble SMAD3 inhibitor as an antitumor agent). So, we believe that this second generation of SMAD3i will be more appropriate for an *in vivo* usage. This molecule is not yet commercially available. **Authors did not report toxicities for these two generations of SMAD3i despite 20 days of treatment (SMAD3i).**

In the first version of the manuscript, we provided (Fig. S2G) expression levels of SMAD3 in the TCGA dataset. In the revised version, we will improve this figure with publicly available datasets (such as Human Protein Atlas) to illustrate the SMAD3 expression pattern.

To evaluate a potential effect of SMAD3i on normal cells, we will expose normal melanocytes in primary culture to SMAD3i (dose-response) to define the therapeutic dose-range. These experiments will be added to the revised version.

Review#2 point 2. *Figure 1C: If I understand correctly, the authors present here the correlation of sgRNA counts between replicates. This is not a valid measure of experiment quality. The authors should present the correlation of FOLD CHANGES. See Hanna and Doench (PMID: 32284587) Box2 for an explanation why.*

Response 2.2. We agree with the reviewer. This point will be fixed in the revised version of the manuscript (data available).

Review#2 point 3. p8 "Thirty-six other genes were recurrently retrieved in the tumors but not in all". Here and elsewhere the authors are rather tough with candidates that do not appear universally across all replicates. Were any of these candidates validated? It could be a rich source of additional drug targets, and perhaps even ones for where small molecule inhibitors already exist and may give less side effects than SMAD3 etc.

Response 2.3. We thank this reviewer for his/her sagacious comments. In the first version of the manuscript, we investigated if our melanoma growth-promoting genes (Table S3) are essential genes (their inhibition trigger cell death). In fact, only YAP1, SLC25A41 and TGIF1 are described as essential genes, suggesting that the inhibition of our melanoma growth-promoting genes (except YAP1, SLC25A41 and TGIF1) may be safe (and confirming the recent publications demonstrating that SMAD3 inhibition is feasible *in vivo* without toxicity). In the revised version, we will scrutinize the current available data banks to explore if small molecule inhibitors already exist for, at least, IL-6, CDK14, MMP2 and BIRC3.

Review#2 point 4. Degree of activation by CRISPRa: The method gives highly variable degrees of activation. This is shown in the literature where fold changes range from 1.5 to many hundreds. Can the authors comment on how much this might affect their results? Figure 1J shows that SMAD3 gives very high activation. Could this partially explain why it consistently appears in the screens? Could this also explain why the ignored other candidates (in above comment) appear in less than all the screens? Could it also explain why some of the 3 sgRNAs for each candidate do not appear as hits in the screen (because they do not activate expression strongly enough)? I'd encourage the authors to discuss this in the paper somewhere, and even consider a few more validation experiments (qRT-PCR) to look at fold activation by the various sgRNAs for the target genes, and check if it correlates with those sgRNAs enrichment in the screen.

Response 2.4. We thank this reviewer for his/her comments. In the first version of the manuscript, we already (in part) discussed this point (relationship between fold transactivation and hits, please see Discussion section, second paragraph). For our best candidate, the SMAD3 expression levels (protein) is really weak in basal condition (501Mel). The transactivation is really "impressive" using CRISPR-SAM but not for all the sgRNAs targeting this promoter. Our preliminary analysis suggests that sgRNAs targeting the longest isoform of SMAD3 are more efficient to transactivate the SMAD3 promoter. To conclude, we will perform an additional experiment: we will evaluate the ability of 6 sgRNA to transactivate SMAD3 gene (in function of their location in the promoter). SMAD3 expression will be quantify by RT-qPCR as suggested. Thus, we will clone these sgRNA and we will evaluate their capability to transactivate SMAD3 gene in the revised version of the manuscript and we will update the discussion in function of these results.

Review#2 point 5. Targeting AhR and SMAD3 by small molecules: The authors discuss some issues with the small molecule inhibitors they used. Why not simply design antisense oligonucleotides / gapmers targeting these mRNAs?

Response 2.5. We thank this reviewer for his/her comments. Since the first version of the manuscript, a study clearly demonstrated that AHR inhibition is feasible and efficient *in vivo* (mice and B16 cells) using a new AhR-antagonist (Kyn 101, Ikena Oncology) and CH-223191 +/- anti-PD-1 (Campedato et al., Nature Comms 2020). To the best of our knowledges, gapmer targeting AhR has not been yet used *in vivo* to reduce the tumor growth. This strategy is promising since we and other demonstrated that ASOs can efficiently reduce tumor growth by targeting RNAs such as TYRP1 or SAMMSON. In the first version of the manuscript, we demonstrated that AhR silencing abrogate SMAD3 expression, suggesting that AhR knockdown is a promising strategy.

In the revised version of the manuscript, we will discuss these two points (AhR antagonists for therapy and ASO targeting *AhR* or *SMAD3*).

Review#2 point 6. *Significance of AhR: I was a little puzzled about how AhR fits with the SMAD3 story. Likely this could be fixed by some explanation at an appropriate point in the paper, about the motivation for looking into AhR. For example, is it because AhR would have some advantage as a drug target over SMAD3 itself? Or is it simply because the authors studied this gene previously?*

Response 2.6. We thank this reviewer for his/her comments. In our previous study (Corre, 2018), we identified AhR as a potential target for melanoma therapy. However, our AhR antagonist (Resveratrol) was poorly efficient *in vivo* due to its poor bioavailability. Thus, we performed these CRISPR screenings to identify a new druggable-target associated to the AhR pathway.

Moreover, in function of the cell type, sustained AhR activation or its inhibition could be safe or deleterious. It is also important to keep in mind that AHR plays a role in the modulation of the adaptive and innate immune systems and AHR is downregulated in autoimmune diseases (cf review Aryl hydrocarbon receptor ligands in cancer: friend and foe (PMID: 25568920)). In addition, AhR antagonism demonstrates tumor cell intrinsic AHR dependence in certain cancers (melanoma and Glioblastoma). The AhR antagonist promotes the T cell expansion (as well as IL2 and IFN- γ), and reduces functional regulatory T-cells stimulated by kynurenine, and decreases suppressive cytokines (and function of myeloid-derived suppressor cells).

Due to the duality of AhR signaling pathway, we thought that it might “be risky” to inhibit AhR in human with cutaneous melanoma. Interestingly, AhR antagonist validated in mice are currently evaluated in clinical trials (NCT04200963, Ikena Oncology) and (NCT04069026, Bayer). So, we don’t know the safety of these antagonists in human. Thus, we looked for a downstream target of AhR instead of targeting AhR itself. We identified SMAD3 (and other candidates). According to studies performed in murine model by two independent teams, SMAD3 inhibition seems safe and efficient to decrease tumor growth. These two strategies seem exploitable.

In the revised version of the manuscript, the AhR-SMAD3 inhibition will be discussed accordingly to these new elements.

Reviewer #2 (Significance (Required)):

My expertise: cancer and CRISPR screening.

18th Sep 2020

Dear Dr. Gilot,

Thank you for the submission of your research manuscript to our editorial offices. I have now had the opportunity to read your manuscript, as well as the referees' reports and your rebuttal letter, and to discuss them with the other members of our editorial team.

We agree with your revision plan, and thus encourage you to submit a revised version of your manuscript along these lines. Acceptance of the manuscript will entail a second round of review. EMBO Molecular Medicine encourages a single round of revision only and therefore, acceptance or rejection of the manuscript will depend on the completeness of your responses included in the next, final version of the manuscript. For this reason, and to save you from any frustrations in the end, I would strongly advise against returning an incomplete revision.

When submitting your revised manuscript, please carefully review the instructions that follow below. Failure to include requested items will delay the evaluation of your revision:

- 1) A .docx formatted version of the manuscript text (including legends for main figures, EV figures and tables). Please make sure that the changes are highlighted to be clearly visible.
- 2) Individual production quality figure files as .eps, .tif, .jpg (one file per figure).
- 3) A .docx formatted letter INCLUDING the reviewers' reports and your detailed point-by-point responses to their comments. As part of the EMBO Press transparent editorial process, the point-by-point response is part of the Review Process File (RPF), which will be published alongside your paper.
- 4) A complete author checklist, which you can download from our author guidelines (<https://www.embopress.org/page/journal/17574684/authorguide#submissionofrevisions>). Please insert information in the checklist that is also reflected in the manuscript. The completed author checklist will also be part of the RPF.
- 5) Before submitting your revision, primary datasets produced in this study need to be deposited in an appropriate public database (see <https://www.embopress.org/page/journal/17574684/authorguide#dataavailability>). Please remember to provide a reviewer password if the datasets are not yet public. The accession numbers and database should be listed in a formal "Data Availability " section (placed after Materials & Method). Please note that the Data Availability Section is restricted to new primary data that are part of this study.

- 6) We would also encourage you to include the source data for figure panels that show essential data. Numerical data should be provided as individual .xls or .csv files (including a tab describing the

data). For blots or microscopy, uncropped images should be submitted (using a zip archive if multiple images need to be supplied for one panel). Additional information on source data and instruction on how to label the files are available at

7) Our journal encourages inclusion of *data citations in the reference list* to directly cite datasets that were re-used and obtained from public databases. Data citations in the article text are distinct from normal bibliographical citations and should directly link to the database records from which the data can be accessed. In the main text, data citations are formatted as follows: "Data ref: Smith et al, 2001" or "Data ref: NCBI Sequence Read Archive PRJNA342805, 2017". In the Reference list, data citations must be labeled with "[DATASET]". A data reference must provide the database name, accession number/identifiers and a resolvable link to the landing page from which the data can be accessed at the end of the reference. Further instructions are available at .

8) We replaced Supplementary Information with Expanded View (EV) Figures and Tables that are collapsible/expandable online. A maximum of 5 EV Figures can be typeset. EV Figures should be cited as 'Figure EV1, Figure EV2' etc... in the text and their respective legends should be included in the main text after the legends of regular figures.

- Additional Tables/Datasets should be labeled and referred to as Table EV1, Dataset EV1, etc. Legends have to be provided in a separate tab in case of .xls files. Alternatively, the legend can be supplied as a separate text file (README) and zipped together with the Table/Dataset file. See detailed instructions here:

9) The paper explained: EMBO Molecular Medicine articles are accompanied by a summary of the articles to emphasize the major findings in the paper and their medical implications for the non-specialist reader. Please provide a draft summary of your article highlighting

10) For more information: There is space at the end of each article to list relevant web links for further consultation by our readers. Could you identify some relevant ones and provide such information as well? Some examples are patient associations, relevant databases, OMIM/proteins/genes links, author's websites, etc...

11) Every published paper now includes a 'Synopsis' to further enhance discoverability. Synopses are displayed on the journal webpage and are freely accessible to all readers. They include a short stand first (maximum of 300 characters, including space) as well as 2-5 one-sentences bullet points that summarizes the paper. Please write the bullet points to summarize the key NEW findings. They should be designed to be complementary to the abstract - i.e. not repeat the same text. We

encourage inclusion of key acronyms and quantitative information (maximum of 30 words / bullet point). Please use the passive voice. Please attach these in a separate file or send them by email, we will incorporate them accordingly.

Please also suggest a striking image or visual abstract to illustrate your article. If you do please provide a png file 550 px-wide x 400-px high.

12) As part of the EMBO Publications transparent editorial process initiative (see our Editorial at <http://embomolmed.embopress.org/content/2/9/329>), EMBO Molecular Medicine will publish online a Review Process File (RPF) to accompany accepted manuscripts.

In the event of acceptance, this file will be published in conjunction with your paper and will include the anonymous referee reports, your point-by-point response and all pertinent correspondence relating to the manuscript. Let us know whether you agree with the publication of the RPF and as here, if you want to remove or not any figures from it prior to publication.

I look forward to receiving your revised manuscript.

Yours sincerely,

Lise Roth

Lise Roth, PhD
Editor
EMBO Molecular Medicine

To submit your manuscript, please follow this link:

Link Not Available

Photos 400-800 DPI

*Additional important information regarding figures and illustrations can be found at <https://embomolmed.embopress.org/authorguide#figures>

Rev_Com_number: RC-2020-00368

New_manu_number: EMM-2020-13466

Corr_author: Gilot

Title: Gain-of-function CRISPR screens identify tumor-promoting genes conferring melanoma cell plasticity and therapy-resistance

Reply to the reviewers

Rebuttal_ EMM-2020-13466-V2

We thank the editor for handling our manuscript and both reviewers for their constructive evaluations. We provide below a detailed list of corrections and experiments performed to address the reviewers' comments and improve the quality of our manuscript.

Reviewer #1 (Evidence, reproducibility and clarity (Required)):

Here Gautron and colleagues performed an in vivo gain-of-function CRISPR screen to identify genes that enable xenograft growth of an otherwise poorly tumorigenic melanoma cell line (501Mel). By using published transcriptomes, authors found that some of these genes (SMAD3, BIRC3, SLC9A5) are also modulated (at the mRNA level) during acquisition of resistance to BRAF inhibitors (BRAFi), which is associated with dedifferentiation in some cases. These genes are also enriched in BRAFi-persister cells in vitro and further enriched after growth of these cells in mice.

Authors then focus on SMAD3, since they find that an SMAD3 transcriptional signature promotes a mesenchymal phenotype by upregulating expression of known BRAFi-resistant genes (EGFR, AXL). This SMAD3 signature is found in dedifferentiated melanoma cell lines and in a subset (20%) of treatment-naïve melanoma patients and most BRAFi-resistant human tumours. Chemical inhibition of SMAD3 combined with BRAFi impairs survival of BRAFi-resistant cells in vitro. This can also be achieved through inhibition of aryl hydrocarbon receptor (AhR, described previously by the authors to contribute to BRAFi-resistance (Corre 2018 Nat Commun)), since AhR drives SMAD3 expression.

The study is very interesting and with potential preclinical implications. In general, the data is well presented and, in most cases, adequately replicated. Most of the work is performed in a limited number of cell lines (mostly 501Mel, in addition to SKMEL28 and Mel1402), even though authors make use of published expression data to support most of the conclusions. However, there are a number of limitations that should be considered.

We thank this reviewer for his/her positive comments on our work and for the suggestions made to improve its relevance

Review#1 point 1: *The current study focuses almost exclusively on BRAFi and BRAFi-resistance, although currently the standard of care is combined BRAFi+MEKi. Even though some analyses are performed on published transcriptomes of BRAFi+MEKi-resistant tumours (Fig.3), the study would be more preclinically relevant if validating key conclusions using the combination.*

Response 1.1. We agree that the combo (BRAFi+MEKi) is the gold-standard. So, in the revised version, we validated the key conclusions using this combo (BRAFi+MEKi), especially in Figure 5.

Review#1 point 2: *Authors propose that expression of SMAD3 signature as a potential biomarker of resistance. The study suggests that this SMAD3-signature could be useful to find sub-populations of pre-existing BRAFi-resistant cells within therapy-naïve lesions. However, it does not provide evidence on stainings of relevant hits on human samples.*

Response 1.2. As suggested, we performed SMAD3 immunostainings in four BRAF-mutant PDXs exposed to BRAF/MEK inhibitors until resistance (recently characterized in reference

21: Marin-Bejar et al. **preprint 2020** (<https://doi.org/10.1101/2020.12.15.422929>). In Appendix Fig. S2A-B, immunostainings showed the emergence of SMAD3⁺ cells in Dabrafenib+Trametinib resistant lesions from the MEL003 and MEL006 PDXs in contrast to PDXs characterized by an intrinsic resistance mechanism (MEL007 and MEL037). These results are in accordance with the increase of SMAD3 mRNA expression during the acquisition of BRAFi-resistance (Fig. 3A-E). We also detected SMAD3⁺ cells in therapy-naïve lesions (especially for MEL003), suggesting that SMAD3 immunostaining might be useful to clinicians. It is important to note that we also showed that a subset (20%) of treatment-naïve melanoma patients has a clear SMAD3 signature (Fig. 7E, TCGA cohort).

Review#1 point 3: *The manuscript hints that SMAD3 inhibitor eliminates BRAFi-resistant cells, especially when combined with BRAFi. However, it appears that this SMAD3 inhibitor as monotherapy is much more potent in decreasing survival of BRAFi-sensitive cells than BRAFi-resistant cells (Fig.5M), which raises concerns on potential toxicities. It is unclear whether this therapy would affect normal cells (melanocytes for instance). Whether this treatment would work in vivo to eradicate MAPKi-resistant tumours is also unknown.*

Response 1.3. We thank the reviewer for this comment.

Due to the fact that SMAD3 is expressed in many cell types (Fig EV 2H), we agree that an eventual *in vivo* toxicity of SMAD3i (alias SIS3) may impair the use of this inhibitor for therapy. We selected this inhibitor accordingly to the literature¹⁻⁴. The most convincing manuscript, published in Nature Communications in 2015 (DOI: 10.1038/ncomms14677), demonstrated that inhibition of Smad3 prevents cancer progression. Tang et al., used the B16F10-luc melanoma model. These cells were s.c. inoculated into mice and mice have been treated with various dosages of SIS3 (0, 2.5, 5.0 or 10 mg.g⁻¹.day⁻¹, i.p.). Interestingly, the antitumoral effect of SIS3 is due to the decrease of B16F10-luc cell proliferation (dose-dependent effect of SIS3) and an increase of NK cell anti-cancer cytotoxicity. **Authors did not report toxicities despite 15 days of treatment (SMAD3i).**

In addition, an upgraded SIS3 version was published in 2020 (Discovery of a novel selective water-soluble SMAD3 inhibitor as an antitumor agent). So, we believe that this second generation of SMAD3i will be more appropriate for an *in vivo* usage. This molecule is not yet commercially available. **This independent team did not report toxicities for these two generations of SMAD3i despite 20 days of treatment (SMAD3i).**

As suggested, we evaluated a potential effect of SMAD3i on normal cells. We thus exposed normal melanocytes in primary culture to SMAD3i (dose-response) to define the “therapeutic window”. These experiments clearly indicated that melanocyte survival is weakly affected by SMAD3 inhibition (up to 20µM), in agreement with the non-toxicity of this inhibitor observed *in vivo* (DOI: 10.1038/ncomms14677). In addition, we confirmed that BRAF inhibition (Vemurafenib 1 or 5µM) did not alter NHEM cell density as expected (Fig. 5). These latest results have been added to the revised version. Altogether, our data and previous reports strongly suggest that SMAD3 inhibition is relevant for cutaneous melanoma and the toxicity is manageable (see NHEMs, Fig. 7).

Here, we use the SIS3 inhibitor to challenge our hypothesis. The *in vivo* ability of SMAD3 inhibitor to reduce the tumor growth and or the relapse will be investigated in another study (BRAFi+MEKi +/- SMAD3i). This type of experiments has to demonstrate that relapse is delayed or suppressed as published in Cell by Rambow F. (2018) and others. It is important to keep in mind that melanoma PDX-exposed to BRAFi+MEKi relapse after ~50 days (and we must monitor the relapse at least ~100 (120) days to observe the protective effect of the tri-therapy). Moreover, a preliminary study is needed to establish the best *in vivo* procedure and the evaluation of SIS3 toxicity (increasing range, acute toxicity). In addition, the water-soluble SMAD3 inhibitor is not yet commercially available.

To conclude, we believe that this *in vivo* experiment is not compatible with a rapid publication of our manuscript identifying AhR-SMAD3 pathway as a critical pathway in BRAFi-resistance.

****Specific comments:****

Review#1 point 4: 1.Fig.1G: there is only 1 measurement "before xenograft" but for "in tumor" there are quite a few from the different xenografts. Is this measurement of sgRNA counts *in vitro* reproducible/consistent enough to consider only one? Similar comment for Fig.4E.

Response 1.4. As specified in the manuscript, we performed two experiments and we merged the sgRNA counts (corresponding to "the before xenograft = cell library"). In the revised version, we showed the two experiments in these two items.

Review#1 point 5: 2.Fig.2: since authors used 2 BRAFi (paradox breaker and vemurafenib), hits are common genes in both treatments? It is not clear from legend or Methods. Were the 2 treatments very different in terms of hits? As said below, information in Supplementary Tables is a bit difficult to interpret due to lack of some important details.

Response 1.5. We recognize that it can be difficult to understand the workflow in the submitted version even if the data were available (Tables EV5 & EV6, page 9 of Supplementary information). In the revised version, we provided a new figure to explain the workflow (explained p9 & p10) (Appendix Figure S1). The main goal was to identify BRAFi-resistant genes, thus we pooled the results obtained with these two BRAF inhibitors.

Review#1 point 6: 3.Fig.3A: text says that majority of candidates were over-expressed in SDR. However, *BIRC3* is not overexpressed in the SDR of M229 and M238 (while it was upregulated in the DTP and DTPP states), although it is upregulated in SKMEL28-SDR and DDR. This should be discussed. Also, how about in the other cell lines from the Song 2017 study (M395, M397, M249) for which there is 2day, DTP, DTPP, SDR data? Did authors analyze protein levels of candidates (*BIRC3*..) as they did for SMAD3 in Fig.5L?

Response 1.6. In the revised version of the manuscript, we showed that *BIRC3* is overexpressed in almost all the samples from the study published by Song in 2017. In fact, *BIRC3* expression is induced in M229 in response to BRAFi at SDR stage (FC = 2.87 but it is difficult to see this weak induction using the blue scale). In order to help the reader, we added a graph showing the fold-change of *BIRC3* expression in cell lines described in the Song study (Resistant versus Parental cells). We grouped the Rx and Ra cell lines; Ra = Resistance and MAPK reActivation and Rr = Resistance and MAPK Redundant). In other words, the BRAFi Resistance observed in Ra cells is not determined by gene reprogramming but it is due to additional genetic alterations such as BRAF splicing events for M395 SDR. As attended, *BIRC3* expression did not increase in these Ra cell lines in contrast to the Rr cell lines (supplementary Fig. EV2A).

Review#1 point 7: 4.Fig.3F-G: graph should show all the patients (as in S2D-E), not only the ones in which there is an increase, otherwise it is misleading since it seems that all patients in the cohort had upregulation. Text says "majority of drug resistant patients", but it would be more accurate to state the percentage of patients with upregulation in resistant sample vs baseline. Last sentence of first paragraph in pag. 12 seems a bit too strong when it appears to conclude from just a correlation between expression and resistance state ("supporting their involvement in establishing drug tolerant and/or resistant phenotypes *in vivo*")

Response 1.7. We agree with the reviewer. For the revised version, we moved these items on Fig. 3 and we indicated the percentage of patients with an upregulation of the signature as

suggested (Fig. 3F & 3G). Moreover, we removed this overstatement (page 12) in the revised version.

Review#1 point 8: 5.Fig.4A: which BRAFi was used here? It is not clear. Legend says 12 mice were xenografted. However, in Methods pag.48, authors mention different BRAFi and different mouse cohorts "...three cell populations were subcutaneously xenografted on female NMRI nude mice flanks (3x10⁶ cells per mice); 501Mel (6 mice), 501Mel CRISPR-SAM vemurafenib resistant (10 mice), 501Mel CRISPR-SAM Paradox Breaker resistant (12 mice)".

Response 1.8. In the revised version, we clearly indicated the BRAF inhibitor used in each experiment to clarify the revised version of the manuscript. All the data were available in Table EV2.

For figure 4A, two BRAF inhibitors were used, the Vemurafenib and the paradox breaker (PLX8394). So, we xenografted three different populations: 501Mel cells (negative control, as shown in Fig. 1B), the VEM-persister cells and the PB-persister cells in immunocompromised mice (3x10⁶ cells/mice) as indicated in the workflow (Fig. 4A). Respectively, three and four tumors have been obtained with these two cell libraries (VEM- and PB-persister cells grafted on 11 and 12 nude mice). The cell library (CRISPR-SAM library not exposed to drug has been showed on Fig. 1, n=10 mice). The legends and the methods section have been improved by indicating these numbers.

Review#1 point 9: 6.Fig.4B: despite the interesting result, it appears that these tumours were grown in the absence of BRAFi. Even if the cell lines were resistant in vitro to BRAFi, it should have been demonstrated that they are able to grow in vivo under continuous BRAFi treatment, showing persistent resistance. Especially since tumours took quite a long time to develop, between 1 and 4 months, similar to the therapy-naïve setting (Fig.1D). Graph in 4B should also number the tumours (as they are identified in Table S7), it would be informative to analyze if tumours arising at 4 weeks had a different sgRNA enrichment vs tumours arising later (i.e. 12-16 weeks).

Response 1.9. We thank this reviewer for his/her positive comments on our work and for the suggestions made to improve its relevance. In absence of BRAFi, the tumor-growth monitoring took 5 months. So, we believe that in presence of BRAFi, the tumor growth will be slower, suggesting that the advised experience could require at least 6 months (and probably more) according to the literature (cf Response 1.3.). In conclusion, we think that this experience is not compatible with a rapid publication of our manuscript.

As suggested, the tumors have been numbered according to Tables EV7 and EV2 (Fig. 4B). Moreover, we formed two groups of tumors (early and late, n=4 and 3 respectively) and we looked for an sgRNA enrichment (early : Black dots and late : Blue dots). We confirmed that sgRNAs targeted our best candidates (*BIRC3*, *SLC9A5* & *SMAD3*) are more detected in "early tumors" than in 'late tumors'. Using this pipeline, we also identified *NTRK3*, *HARS2* and *PDGFRB* as interesting candidates.

Review#1 point 10: 7. When referring to ME1402, authors say in page 14 that they are intrinsically resistant to BRAFi, compared to acquired resistance of SKMEL28R. However, graphs 5G-H show that in both cases BRAFi decreased cell density by 50% in both cases, which is quite substantial. Was this 5 μ M? It is not stated in legend, but it is quite high. Was this the concentration used to make them resistant? It is not clear from Methods. As said above for original M229 and M238-resistant derivatives, 1 μ M was used (Nazarian et al 2010 Nature), as in other papers in the field (Obenauf et al 2015 Nature, Wang et al 2018 Cell, among other studies). If BRAFi treatment was 84h, it should be stated also in the legend (also in other figures).

Response 1.10 SKMEL28R cell line has been established by the team of Prof. D. PEEPER (NKI) (a gift to JC Marine's lab). The parental cell line was exposed to PLX-4720 during several months (up to 3 μ M). Cells are routinely cultivated with 0.1 μ M of PLX-4720. ME1402 cells are intrinsically resistant to BRAFi. In fact, the word "Resistant" is inaccurate in this context since we confirmed that SKMel28 S and R are "sensitive, at least in part", at 5 μ M to BRAFi (Vem) (Fig. 5F).

The 5 μ M (BRAFi, Vem) was an arbitrarily choice for the treatment of these cell lines (84h, Fig. 5). In the revised version, methods section and legends have been improved according to the advices. Moreover, for another query (Review#1 point 14), dose of 1 and 5 μ M have been compared in eight cell lines to define two groups: sensitive and resistant cell lines to BRAFi (Vem). These data have been added to the revised version (Fig. 5).

Review#1 point 11: 8. Fig.5I: this is in mel1402, but the knockdown in SKMEL28R (Fig.5G) is not shown. Same for Fig.S3F and S3G and corresponding knockdown.

Response 1.11. All the knock-down validations have been included in the revised version of the manuscript (Fig. EV3).

Review#1 point 12: 9. Fig.5J: text says that SMAD3i "strongly" decreased levels of p-SMAD3 induced by TGF β . However, from the blot it does not seem very strong. Do authors have a quantification from different blots? In line with this, authors should show p-SMAD3 blots from experiment in Fig.5K, and p-ERK and p-SMAD3 for Fig.5M to show that drugs are acting on-target.

Response 1.12. We agree with the comment. We removed this overstatement in the revised version. A quantification has been done: the inhibition is ~25% for this cell line overexpressing SMAD3 and stimulated by TGF β . To better illustrate the ability of SMAD3i (SIS3) to inhibit the P-SMAD3, we analyzed its inhibitory effect on endogenous SMAD3 protein (SKMel28R). This cell line expressed a high level of SMAD3 protein as showed in Fig. 5J. These experiments have been added in the revised version.

As suggested, the other WB experiments (controls) have been added to the revised version (Fig.5 & EV3K).

Review#1 point 13. 10. Fig.5L: was p-SMAD3 also increased? Or is it all due to increased total protein?

Response 1.13. As suggested, we examined the P-SMAD3 in the cell lines (Fig. 5L). The basal expression levels of P-SMAD3 are weak in general except for M238S cell line. P-SMAD3 is detectable in the resistant cell lines (SKMel28R, M229R & M238R). The western-blot experiments have been added to the revised version.

It is important to keep in mind that SMAD3 regulation by phosphorylation is not fully understood and the phospho-SMAD3 Ser423/425 status is not strictly correlated to SMAD3 transcriptional

activity (Ooshima et al, 2019) (p15). So, we also evaluated the effect of SMAD3i using a reporter assay (Fig. 5K).

Review#1 point 14. 11.Fig.5M: these experiments need a proper quantification (dose-response, IC50, synergy analyses) and including more BRAFi-resistant lines, since in the current experiment there is only one (SKMel28R). Comparing the pairs that authors have (M229S/R, M238S/R, SKMEL28S/R) would be much more informative, since they usually are sensitive=differentiated, resistant=undifferentiated. And as mentioned above, with current data, SMAD3 seems to be more potent in reducing viability of BRAFi-sensitive than BRAFi-resistant when used as monotherapy, which raises concerns over specificity and dependency of resistant cells on this pathway.

Also, 5uM BRAFi is quite high and may obscure synergy effects. For example, Fig.5A used 2uM, as in Lito 2012 Cancer Cell. Or even 1uM, as said above original M229 and M238 resistant derivatives (used in this study) were isolated after 1uM vemurafenib chronic treatment (Nazarian 2010 Nature).

Response 1.14. To further illustrate the capability of SMAD3i to target cutaneous melanoma, we performed the suggested experiments in more cell lines (3 pairs) (Fig. 5). We further characterized the inhibitory effect of SIS3 (SMAD3 inhibition) on these cell lines and on melanocytes (NHEMs, 3 donors). We showed that SIS3 reduced the cell density of all melanoma cells. Interestingly, the effect on NHEMs is weak and manageable (see Review#1 point 3) when compared to melanoma cells. The effect seems to be associated to the SMAD3 expression levels detected by WB in these cell lines (Fig 5L). Our results might suggest that these melanoma cells could be addicted to SMAD3 activity.

We also investigated the interest to combine BRAFi (Vem) and SMAD3i (SIS3) in the BRAFi-resistant cell lines (SKMel28R, M229R & M238R). We showed that SMAD3 inhibitor alone or in combination with BRAFi (Vem 5µM) or BRAFi+MEKi (Cobi 1µM) might be a promising treatment to reduce the amount of persister cells (melanoma). This is in accordance to the results of SMAD3 depletion experiments (Fig. 5) and an independent study using SMAD3 inhibitor *in vivo* (DOI: 10.1038/ncomms14677).

Review#1 point 15. 12.Fig.6B: here authors show induction of SMAD3 after treatment with AhR ligand for 10 days. Fig.6C-D experiments were for 7 days. It is unclear why such a long time was needed to show induction of transcription. In their previous paper Corre 2018 Nat Commun, authors showed transcriptomic changes after 48 h TCDD treatment in 501Mel cells. Was SMAD3 upregulated here?

Response 1.15. In our previous study (Corre et al., 2018), we demonstrated that AhR activation by the exogenous ligand of AhR (TCDD (dioxin) promoted the transcriptional activation of AhR in 501Mel. We observed a significant induction of SMAD3 mRNA in these experimental conditions (48h; log FC = 0.43 ; pvalue 0.00213). The induction was stronger after a longer treatment (7 days, Fig. 6B).

Review#1 point 16. 13.Fig.7B: text says that SMAD3 depletion further validated MMP2, AXL, EGFR and JUNB as SMAD3-regulated genes. Since the list was comprised of 9 genes, to clarify, were the other genes not tested or not regulated upon SMAD3 knockdown?

Response 1.16. In the submitted version, we arbitrarily selected the most “important genes” from this SMAD3-signature (MMP2, AXL, EGFR & JUNB). In the revised version, we showed the 9 genes as suggested and we updated the result section (Fig. 7H and p18).

Review#1 point 17. 14.Fig.7D: SMAD3-signature inducibility is higher in differentiated cells. However, basal expression is higher in dedifferentiated cells, which should be discussed. However, dedifferentiated cells here only include the M238 pair (S and R). Did authors

compare S and R of the pairs in which dedifferentiation increases from S to R (SKMEL28 and M229 pair, Fig.5L)? It would be a more appropriate comparison.

Response 1.17. We agree. This comparison has been performed for the revised version in six cell lines (three couples S & R; SKMel28, M229 and M238). We evaluated the basal expression level of the SMAD3-signature in basal condition and its inducibility by the TGF β (Fig. 7B and 7C). As attended, the inducibility is more pronounced in sensitive cell lines. The suggested comparison is more comprehensible by the reader. It is now easy to see that the basal SMAD3-signature is high in resistant cell lines.

It is important to keep in mind that the differentiation status of S & R cell lines is subtle: Parental cell lines are not all catalogued as "melanocytic" (SKMel28S, M229S and M238S are respectively M, T and NC). Conversely, the Resistant cell lines are not all undifferentiated (U) ((SKMel28R, M229R and M238R are respectively U, U and NC). These results are in accordance to Fig 5B and 5C showing the SMAD3 mRNA expression levels analyzed in 53 cells lines (in function of the 4 differentiation states). In addition, Fig. 7D showed that SMAD3-signature is high in undifferentiated (U) cell lines but we observed a disparity amongst the NC cell lines.

Review#1 point 18. 15.Regarding statistical analyses, some graphs should be revised and adjusted for multiple comparisons instead of t-test (using ANOVA or equivalent), like 5G-H, 6B-D, 7C,E. Also, in some cases like 6B-D, 7C,E it is unclear which comparison is being made, since there is a line on top of 2 bars. For example, 6B, line on top of TCDD and ITE, 2 asterisks vs dms0, does this mean that each comparison DMSO vs TCDD and DMSO vs ITE is 2 asterisks? While methods say that Anova was used in some cases and specified in figure legends, this is not found in legends.

Response 1.18. We agree that our graphical representation explaining the multiple comparison is not clear. We modified these representations as suggested and we improved the figure legends (stats). Moreover, we performed Anova tests as indicated in the figure legends and M&M section.

****Minor comments:****

Review#1 point 19. Some Supplementary Tables lack an explanation/legend of what the table and data show and the reader has to guess. For example, Table S7, what do numbers in each column mean? Is PB paradox breaker and V vemurafenib? Or Table S6, what is 0, 1, 2..? sgRNA counts? The other tables should be revised accordingly.

Response 1.19. We thank the reviewer for this comment. We apologize. As initially described in the submitted version, the numbers (0, 1 and 2, ...) mean number of significant sgRNAs enriched in each condition per gene : for Vem (Vemurafenib), PLX8394 (PB, Paradox Breaker) and the combination of the two BRAFi screens (PB+Vem; PBV). In the revised version, we improved the legends of the tables.

Review#1 point 20. Pag.7: when speaking about 501mel are unable to generate tumors in mice, it should say nude mice (used in ref.29) instead of immunocompromised.

Response 1.20. We thank the reviewer for this comment. In the revised version, we indicated the mouse strain (nude).

Review#1 point 21. Pag.11: differentiated cell line M229 (melanocytic).. However, Fig.5L says "T" for transitory.

Response 1.21. We thank the reviewer for this comment. This mistake has been fixed in the revised version of the manuscript.

Review#1 point 22. *Fig. 1K. BIRC3 tumors seem to be delayed (day 11) compared to the other two, especially SLC9A5 (day 4-5). This should be discussed given the strong phenotype of SLC9A5 even with a moderate overexpression (average 2fold, Fig. 1J) compared to the other two (6- and 20-fold). Legend should specify how many mice were injected with each cell line.*

Response 1.22. We thank the reviewer for this comment. In the Fig. 1K, we performed a proof-of-concept experiment showing that the increase of SMAD3 or BIRC3 or SLC9A5 promotes the tumor growth in contrast to parental cell line. It is interesting to keep in mind that SLC9A5 encodes a transporter. Thus, a moderate mRNA expression increase could significantly enhance its protein expression and thus the global activity of this transporter. This point has been discussed in the revised version of the manuscript (Discussion section, page 22).

In the submitted version, we indicated in legend of Figure 1 the number of mice and in Table EV2 (please see page 51: *Workflow depicting the validation step: 501Mel cells overexpressing SMAD3, BIRC3 or SLC9A5 (obtained by CRISPR-SAM) were xenografted on nude mice and tumor volume was monitored using caliper. 3x10⁶ cells/mouse. n=7, 6 and 6 mice, respectively.*). Due to the weak number of tumors per group and to avoid an overstatement, we did not interpret the difference of tumors onset.

Review#1 point 23. *Fig.2E: when referring to this dataset, it should be mentioned that these lines are therapy-naïve (never treated with drug) but they are intrinsically (partially) resistant to BRAFi when exposed to the drug, to distinguish from acquired resistance models (M229, M238..).*

Response 1.23. We thank the reviewer for this comment. This point has been fixed in the revised version of the manuscript.

Review#1 point 24. *Fig.2F: the axis is missing the numbers. Perhaps this should go in supplementary, or just indicate in the legend or elsewhere which quadrant of the graph shows the hits.*

Response 1.24. We thank the reviewer for this comment. We removed this item, as suggested.

Review#1 point 25. *Second paragraph in page 14 seems out of context, it should go earlier in page 13 when describing that SMAD3 high in dedifferentiated cells.*

Response 1.25. We thank the reviewer for this comment. We removed this paragraph.

Review#1 point 26. *Pag.17: when describing Hugo 2016 Cell patient samples, it should be specified that these are pre-treatment biopsies from responders and non-responders to anti-PD-1 treatment. Fig legend should be corrected too ("melanoma exposed to PD-1 therapy".. these samples were not exposed to anti-PD-1, they were collected before treatment). Therefore, higher SMAD3 signature would identify anti-PD1 non-responders*

Response 1.26. We apologize. This point has been fixed in the revised version of the manuscript (results p18 and discussion).

Review#1 point 27. *Fig.7I: text mentions that "comparing the Smad3-signature with the classical mesenchymal-like signature of melanoma (TCGA cohort) highlighted a significant correlation" but there is no correlation analysis here. So currently it would be an "overlap" or similar.*

Response 1.27. For the revised version, we provided the plot illustrating this correlation and the p-value (Fig. EV4A).

Review#1 point 28. *In discussion p.20 when referring to the transactivation obtained by CRISPR-SAM as "massive" (Fig.1I) authors should avoid using these subjective adjectives, especially since expression levels were not quantified.*

Response 1.28. We apologize. We fixed this issue in the revised version.

Review#1 point 29. *SLC9A5 is defined in p.21 but this should go the first time SLC9A5 is described in the paper.*

Response 1.29. We fixed this issue in the revised version of the manuscript.

Review#1 point 30. *Perhaps Table S8 is not needed since the list of genes (9) is already mentioned in the text (page 16).*

Response 1.30. We removed this superfluous table in the revised version.

Review#1 point 31. *Methods should specify formula used to calculate tumour volume.*

Response 1.31. We added the formula in the revised version of the manuscript (p28, Methods).

Authors should be commended for the detailed CRISPR-SAM protocol in Supplementary methods, it will be very useful in order for others to replicate/use this technology. All relevant prior studies are properly referenced.

We thank this reviewer for his/her positive comments on our manuscript.

Reviewer #1 (Significance (Required)):

The study builds upon previous findings of the group describing the role of AhR in BRAFi-resistance (Corre 2018 Nat Commun). The current study finds that the same genes that enable tumour growth of therapy-naïve melanoma cells are also enriched during acquisition of resistance to BRAFi, given that SMAD3 in particular is a target of AhR. The study suggests that this SMAD3-signature could be useful to find sub-populations of pre-existing BRAFi-resistant cells within therapy-naïve lesions, which should be evaluated.

This study would be important for researchers working on melanoma and also potentially on other cancers driven by EMT phenotype switching. The study hints that this SMAD3-signature could be operative in subsets of glioblastoma tumours, and also in the context of anti-PD-1 in melanoma.

Field of expertise: Melanoma, therapy resistance, invasion and metastasis, cytoskeleton.

Reviewer #2 (Evidence, reproducibility and clarity (Required)):

Gautron and colleagues search for novel genes that promote phenotypic plasticity and drug resistance in melanoma, discovered by in vivo CRISPR screens. The subject is highly topical, the technology is cutting edge, the presentation is excellent, and the work is technically sound. My critiques are relatively minor.

We thank this reviewer for his/her positive comments on our manuscript.

Review#2 point 1. *Side effects and toxicity: A key finding of this paper is SMAD3 as a potential therapeutic target in BRAFi-resistant melanoma. However, I did not find any discussion of how SMAD3-inhibition might result in toxicities or side effects in a treated individual. How widely-expressed is SMAD3? Is anything known about the effects of inhibition to a human or mouse? Could any strategies mitigate such toxicity?*

Response 2.1. We thank the reviewer for this comment. The possible “side effects and toxicity” have been also highlighted by Reviewer 1 (cf **Review#1 point 3.**).

Due to the fact that SMAD3 is expressed in many cell types (Fig. EV 2H), we agree that an eventual *in vivo* toxicity of SMAD3i (alias SIS3) may impair the use of this inhibitor for therapy. We selected this inhibitor accordingly to the literature¹⁻⁴. The most convincing manuscript, published in Nature Communications in 2015 (DOI: 10.1038/ncomms14677), demonstrated that inhibition of Smad3 prevents cancer progression. Tang et al., used the B16F10-luc melanoma model. These cells were s.c. inoculated into mice and mice have been treated with various dosages of SIS3 (0, 2.5, 5.0 or 10 mg.g⁻¹.day⁻¹, i.p.). Interestingly, the antitumoral effect of SIS3 is due to the decrease of B16F10-luc cell proliferation (dose-dependent effect of SIS3) and an increase of NK cell anti-cancer cytotoxicity. **Authors did not report toxicities despites 15 days of treatment (SMAD3i).**

In addition, an upgraded SIS3 version was published in 2020 (Discovery of a novel selective water-soluble SMAD3 inhibitor as an antitumor agent). So, we believe that this second generation of SMAD3i will be more appropriate for an *in vivo* usage. This molecule is not yet commercially available. **Authors did not report toxicities for these two generations of SMAD3i despites 20 days of treatment (SMAD3i).**

In the first version of the manuscript, we provided (Fig. S2G) expression levels of *SMAD3* in the TCGA dataset. In the revised version, we improved this figure with publicly available datasets (www.proteinatlas.org, Fig. EV 2H) to illustrate the *SMAD3* expression in cell types.

To evaluate a potential effect of SMAD3i on normal cells, we exposed normal melanocytes (three donors) in primary culture to SMAD3i (dose-response) to define the “therapeutic window”. These experiments clearly indicated that melanocyte survival is weakly affected by SMAD3 inhibition (up to 20µM), in agreement with the non-toxicity of this inhibitor observed *in vivo* (DOI: 10.1038/ncomms14677). In addition, we confirmed that BRAF inhibition (Vemurafenib 1 or 5µM) did not alter NHEM cell density as expected (Fig. 5). These latest results have been added to the revised version. Altogether, our data and previous reports strongly suggest that SMAD3 inhibition is relevant for cutaneous melanoma and the toxicity is manageable.

Review#2 point 2. *Figure 1C: If I understand correctly, the authors present here the correlation of sgRNA counts between replicates. This is not a valid measure of experiment quality. The authors should present the correlation of FOLD CHANGES. See Hanna and Doench (PMID: 32284587) Box2 for an explanation why.*

Response 2.2. We agree with the reviewer. This point has been fixed in the revised version of the manuscript (Fig. 1C removed). In the revised version, we also removed the sentence

“The quality of the two cell library replicates was evaluated by estimating the distribution of the guides (Fig. 1C, right panel).”

Based on the recommendations (PMID: 32284587), fold changes have been calculated for each replicate (see below). Hanna and Doench explained that *“positive-selection screens, in which a drug treatment or other stringent selective pressure is applied, often have lower replicate correlation because the majority of genes are not involved in the phenotype and are therefore expected to have little signal”*. *“The low correlation is not a cause for concern as long as some portion of guides enrich in both replicates”*. Here, we have the same conclusion with our data (see below and Tables EV5 and EV6). We also confirmed that the *“identification of guides that do enrich across replicates should constitute interesting candidates”*.

Review#2 point 3. p8 *“Thirty-six other genes were recurrently retrieved in the tumors but not in all”. Here and elsewhere the authors are rather tough with candidates that do not appear universally across all replicates. Were any of these candidates validated? It could be a rich source of additional drug targets, and perhaps even ones for where small molecule inhibitors already exist and may give less side effects than SMAD3 etc.*

Response 2.3. We thank this reviewer for his/her sagacious comments. In the first version of the manuscript, we investigated if our melanoma growth-promoting genes (Table EV4) are essential genes (their inhibition trigger cell death). In fact, only YAP1, SLC25A41 and TGIF1 are described as essential genes, suggesting that the inhibition of our melanoma growth-promoting genes (except YAP1, SLC25A41 and TGIF1) may be safe (and confirming the recent publications demonstrating that SMAD3 inhibition is feasible *in vivo* without toxicity).

In the revised version, we examined the candidates (alias genes promoting tumor growth, Tables EV3 and EV4) which can be inhibited using chemical inhibitors. Among these candidates, we validated the role of BIRC3 using the chemical inhibitor Birinapant (Fig EV1B, C). Our results are in accordance with another manuscript available on BioRxiv (doi: <https://doi.org/10.1101/843185>). Using a two-cell type (2CT) whole-genome CRISPR-Cas9 screen, authors identified BIRC2 as a target to increase the immunotherapy efficiency on melanoma cells. They found that the inhibition of BIRC2 (by CRISPR-Cas9 or chemical inhibitor (Birinapant)) enhanced melanoma cell destruction by T cells. Mechanistically, they showed that BIRC2 promoted immunotherapy resistance through inhibiting non-canonical NF- κ B signaling and limiting inflammatory chemokine production. In 2013, another publication (PMID: 23403634) demonstrated that the Birinapant in combination with TNF- α is effective on melanoma cells. **In conclusion, these results confirmed that BIRC3 and BIRC2 inhibition using chemical inhibitor (Birinapant) may represent an interesting way to restrict tumor growth and therapy-resistance.**

In the revised version, the BIRC3 inhibition experiments have been included (two cell lines, Fig. EV1B and EV1C)

Review#2 point 4. *Degree of activation by CRISPRa: The method gives highly variable degrees of activation. This is shown in the literature where fold changes range from 1.5 to many hundreds. Can the authors comment on how much this might affect their results? Figure 1J shows that SMAD3 gives very high activation. Could this partially explain why it consistently appears in the screens? Could this also explain why the ignored other candidates (in above comment) appear in less than all the screens? Could it also explain why some of the 3 sgRNAs for each candidate do not appear as hits in the screen (because they do not activate expression strongly enough)? I'd encourage the authors to discuss this in the paper somewhere, and even consider a few more validation experiments (qRT-PCR) to look at fold activation by the various sgRNAs for the target genes, and check if it correlates with those sgRNAs enrichment in the screen.*

Response 2.4. We thank this reviewer for his/her comments. In the first version of the manuscript, we already (in part) discussed this point (relationship between fold transactivation, basal expression levels and hits, please see Discussion section, second paragraph). For the revised version, we generated an additional figure (Fig. EV5) and we modified the discussion based on these complementary results.

As suggested, we investigated the ability of sgRNAs to drive *SMAD3* expression to better understand why some of the 3 sgRNAs for each candidate do not appear as hits in the screens (Fig. EV5). Firstly, we showed that the sgRNAs contained in the library (12 sgRNAs promoting *SMAD3* expression) target four different *SMAD3* promoters according to NCBI. By examining the sgRNAs targeting *SMAD3*, we found that 2 sgRNA are enriched (BRAFi resistance) (Fig. EV5). These two sgRNAs promote the expression of the longest *SMAD3* isoform; the *SMAD3* mRNA expressed in our model (501Mel cells) (Fig. EV5A-E). It is important to note that the nine other sgRNAs targeting *SMAD3* are not able to confer the BRAFi-resistance since they target other *SMAD3* isoforms. From RNA-seq experiments (501Mel used for the screen and SKMel28 used for the validation steps), we observed that exon 1 is clearly detected in these two melanoma cell lines, suggesting that the longest isoform of *SMAD3* (NM_005902) seems to be associated with the BRAFi resistance.

Next, we investigated why the sgRNA g71 was not associated to the resistance in our screens since it targets the exon 1 too. We hypothesized that this sgRNA could be less potent to transactivate *SMAD3* expression when compared to g92 and g116. We compared the transactivation ability of these three guides to two other guides (g50 and g155, not available in the library) also targeting the exon 1 of the longest isoform. As negative control, we used another guide; gIntron3 (targeting the intron 3 of NM_005902 (and the exon 1 of NM_001145104)). gIntron3 was not associated to BRAFi resistance in our screens. By transient transfection, we compared the ability of these 6 guides to transactivate *SMAD3* expression. The quantification has been performed by RT-qPCR (exon 1 or 6). As attended, the guide conferring the BRAFi-resistance are potent to promote *SMAD3* expression. We also observed that in function of the binding site (position), the transactivation ability is variable as initially demonstrated by the Zhang lab. We also showed that gIntron3 induced *SMAD3* expression but not the right isoform to confer resistance to BRAFi (it promotes the shortest isoform). Surprisingly, in these experiments, the guide g71 efficiently transactivate *SMAD3* gene.

To conclude, we confirmed that 3 guides per isoform is probably not enough to maximize the chance to identify hits. The current sgRNA libraries contain more than 3 sgRNA per target probably to compensate this issue. Moreover, we highlighted the need to analyze the effect of each sgRNA and not solely the z-score per gene for CRISPR screens. Here, we showed that 2 sgRNA targeting *SMAD3* gene conferred BRAFi resistance (2/12 for *SMAD3* gene) but 2 among 3 sgRNAs targeting the *SMAD3* isoform expressed in our model (501Mel cells) are efficient. For our analyses, we selected genes with at least two efficient sgRNA. It is highly probable that we missed interesting candidates (false negative) due to the weak number of sgRNA/target in the library and we solely performed two replicates.

Review#2 point 5. *Targeting AhR and SMAD3 by small molecules: The authors discuss some issues with the small molecule inhibitors they used. Why not simply design antisense oligonucleotides / gapmers targeting these mRNAs?*

Response 2.5. We thank this reviewer for his/her comments. Since the first version of the manuscript, a study clearly demonstrated that AHR inhibition is feasible and efficient *in vivo* (mice and B16 cells) using a new AhR-antagonist (Kyn 101, Ikena Oncology) and CH-223191 +/- anti-PD-1 (Campesato et al., Nature Comms 2020).

To the best of our knowledges, gapmer targeting AhR has not been yet used *in vivo* to reduce the tumor growth. This strategy is promising since we and others demonstrated that ASOs can efficiently reduce melanoma tumor growth by targeting RNAs such as *TYRP1* or *SAMMSON*. In the first version of the manuscript, we demonstrated that AhR silencing abrogates *SMAD3* expression, suggesting that AhR knockdown is also a promising strategy.

In the revised version of the manuscript, we included these two points (AhR antagonists for therapy and ASO targeting *AhR* or *SMAD3*) in the discussion.

Review#2 point 6. *Significance of AhR: I was a little puzzled about how AhR fits with the SMAD3 story. Likely this could be fixed by some explanation at an appropriate point in the paper, about the motivation for looking into AhR. For example, is it because AhR would have some advantage as a drug target over SMAD3 itself? Or is it simply because the authors studied this gene previously?*

Response 2.6. We thank this reviewer for his/her comments. In our previous study (Corre, 2018), we identified AhR as a potential target for melanoma therapy. However, our AhR antagonist (Resveratrol) was poorly efficient *in vivo* due to its poor bioavailability. Thus, we performed these CRISPR screenings to identify a new druggable-target associated to the AhR pathway. During the review process, another team published an efficient AhR antagonist for *in vivo* usage (cf response 2.5).

Moreover, in function of the cell type, sustained AhR activation or its inhibition could be safe or deleterious. It is also important to keep in mind that AhR plays a role in the modulation of the adaptive and innate immune systems and AHR is downregulated in autoimmune diseases (cf review Aryl hydrocarbon receptor ligands in cancer: friend and foe (PMID: 25568920)). In addition, AhR antagonism demonstrates tumor cell intrinsic AhR dependence in certain cancers (melanoma and Glioblastoma). The AhR antagonist promotes the T cell expansion (as well as IL2 and IFN- γ), and reduces functional regulatory T-cells stimulated by kynurenine, and decreases suppressive cytokines (and function of myeloid-derived suppressor cells).

Due to the duality of AhR signaling pathway, we thought that it might “be risky” to inhibit AhR in human with cutaneous melanoma. Nonetheless, AhR antagonist validated in mice are currently evaluated in clinical trials (NCT04200963, Ikena Oncology) and (NCT04069026, Bayer). To date, we don't know the safety of these antagonists in human. Thus, we looked for a downstream target of AhR instead of targeting AhR itself. We identified *SMAD3* (and other candidates). According to studies performed in murine model by two independent teams, *SMAD3* inhibition seems safe and efficient to decrease tumor growth. These two strategies seem useable.

In the revised version of the manuscript, the AhR-*SMAD3* inhibition has been revisited accordingly to these new elements.

Reviewer #2 (Significance (Required)):

My expertise: cancer and CRISPR screening.

29th Jan 2021

Dear Dr. Gilot,

Thank you for the submission of your revised manuscript to EMBO Molecular Medicine. We have now received the enclosed reports from the two referees who re-reviewed your manuscript. As you will see, they are both supportive of publication, and I am therefore pleased to inform you that we will be able to accept your manuscript, once the following minor points will be addressed:

1) Referees' comments:

- we propose following referee #1' suggestion and including the figure from the rebuttal response 1.9 in the Appendix file.
- please add "SMADi" in Figure 5J.

2) Main manuscript text

- Please answer/correct the changes suggested by our data editors in the main manuscript file (in track changes mode). This file will be sent to you in the next few days. Please use this file for any further modification.
- Thank you for providing a "Conflict of Interests" section. Please remove "The authors declare no potential conflicts of interest." on pages 1 and 2.
- Abstract, last sentence: remove "expands" (highlights)
- Material and methods:
 - o We note that you often refer to previously published methods. Please make sure that enough information is nevertheless provided for reproducibility and transparency purposes.
 - o Please indicate the origin and age of the mice.
- Statistics: Please indicate in the figures or in the legends the exact n= and exact p= values, not a range, along with the statistical test used. You may provide these values as a supplemental table in the Appendix file, but not as part of the Source Data.
- Data Availability Section:

Please note that the Data Availability Section is restricted to new primary data generated in this study, and should not list previously generated datasets. Please note that accession to new primary datasets has to be made public before acceptance of the manuscript. Please update the checklist accordingly (section F/18).

3) Figures, Appendix and Source Data:

- The EV tables need their legends removed from the Appendix and added directly to each respective file in a separate tab.
- The source data currently uploaded as Fig. S3 and Table EV9 should be uploaded as Source Data files, 1 file per figure.
- Appendix: Appendix Text (with updated table of content) should be merged with Appendix Figure S1 and Appendix Figure S2.

4) Please note that all corresponding authors are required to supply an ORCID ID for their name upon submission of a revised manuscript. An ORCID identifier is missing for Prof. Marie-Dominique Galibert.

5) Thank you for providing The Paper Explained section. Your manuscript was cross-checked for

similarities with other manuscripts, and a resemblance was found between the first paragraph of The Paper Explained and a previously published study. Please modify this paragraph accordingly (see the parts of your text highlighted in green in the attached document).

6) Thank you for providing a section "For more information". Please add a title (or short description) for the weblink provided.

7) Thank you for providing a nice synopsis picture. When resized to 550px width, some parts of the text/figure are a bit small or difficult to read. Could you please resize some of the elements to make sure everything is readable/visible?

8) As part of the EMBO Publications transparent editorial process initiative (see our Editorial at <http://embomolmed.embopress.org/content/2/9/329>), EMBO Molecular Medicine will publish online a Review Process File (RPF) to accompany accepted manuscripts.

In the event of acceptance, this file will be published in conjunction with your paper and will include the anonymous referee reports, your point-by-point response and all pertinent correspondence relating to the manuscript. Let us know whether you agree with the publication of the RPF and as here, **IF YOU WANT TO REMOVE OR NOT ANY FIGURES** from it prior to publication.

I look forward to receiving your revised manuscript.

Yours sincerely,

Lise Roth

Lise Roth, PhD
Editor
EMBO Molecular Medicine

To submit your manuscript, please follow this link:

Link Not Available

Photos 400-800 DPI

*Additional important information regarding figures and illustrations can be found at <https://bit.ly/EMBOPressFigurePreparationGuideline>

The system will prompt you to fill in your funding and payment information. This will allow Wiley to send you a quote for the article processing charge (APC) in case of acceptance. This quote takes into account any reduction or fee waivers that you may be eligible for. Authors do not need to pay any fees before their manuscript is accepted and transferred to our publisher.

***** Reviewer's comments *****

Referee #1 (Comments on Novelty/Model System for Author):

This revised manuscript has been improved by validating key in vitro experiments in additional resistant models, and also by providing immunostainings on PDX samples. All my concerns have been addressed and/or adequately discussed.

Referee #1 (Remarks for Author):

EMM-2020-13466-V2

Authors have done a thorough and careful job of responding to the concerns of the previous version. I support publication and congratulate the authors for this excellent study.

I would suggest including the figure "Tumor onset early vs late" (from Response 1.9) in the manuscript, at least the data for SMAD3, BIRC3 and SLC9A5 (perhaps as a supplementary) since it further supports the findings.

In Fig.5J, it looks like "SMAD3i" is missing from panel next to "- +"

Referee #2 (Comments on Novelty/Model System for Author):

This is a technically sound and high impact study.

Referee #2 (Remarks for Author):

I'm satisfied with the corrections.

Reply to the reviewers

Rebuttal_ EMM-2020-13466-V3

We thank the editor for handling our manuscript and both reviewers for their constructive evaluations. We provide below a detailed list of corrections that we performed to address the reviewers' comments and improve the quality of our manuscript.

Reviewer #1

Authors have done a thorough and careful job of responding to the concerns of the previous version. I support publication and congratulate the authors for this excellent study.

We thank this reviewer for his/her positive comments on our manuscript.

I would suggest including the figure "Tumor onset early vs late" (from Response 1.9) in the manuscript, at least the data for SMAD3, BIRC3 and SLC9A5 (perhaps as a supplementary) since it further supports the findings.

As suggested, we included this item as a new figure: please see Appendix Figure S3.

In Fig.5J, it looks like "SMAD3i" is missing from panel next to "- +"

As suggested, we modified the figure 5J.

Reviewer #2

This is a technically sound and high impact study.

I'm satisfied with the corrections.

We thank this reviewer for his/her positive comments on our manuscript.

The authors performed the requested editorial changes.

11th Feb 2021

Dear Dr. Gilot,

Thank you for sending the revised files. I am now very pleased to accept your manuscript for publication in EMBO Molecular Medicine!

Please note that I removed the following sentence from the legend of Fig. EV3K: "Each histogram represents the mean + s.d."

Please contact us immediately if this is not correct.

Your manuscript will now be sent to our publisher to be included in the next available issue of EMBO Molecular Medicine.

Please read below for additional important information regarding your article, its publication and the production process.

Congratulations on a nice study!

Yours sincerely,

Lise Roth

Lise Roth, Ph.D
Editor
EMBO Molecular Medicine

Follow us on Twitter @EmboMolMed
Sign up for eTOCs at embopress.org/alertsfeeds

*** ** IMPORTANT INFORMATION ** **

SPEED OF PUBLICATION

The journal aims for rapid publication of papers, using using the advance online publication "Early View" to expedite the process: A properly copy-edited and formatted version will be published as "Early View" after the proofs have been corrected. Please help the Editors and publisher avoid delays by providing e-mail address(es), telephone and fax numbers at which author(s) can be contacted.

Should you be planning a Press Release on your article, please get in contact with embomolmed@wiley.com as early as possible, in order to coordinate publication and release dates.

LICENSE AND PAYMENT:

All articles published in EMBO Molecular Medicine are fully open access: immediately and freely available to read, download and share.

EMBO Molecular Medicine charges an article processing charge (APC) to cover the publication costs. You, as the corresponding author for this manuscript, should have already received a quote with the article processing fee separately. Please let us know in case this quote has not been received.

Once your article is at Wiley for editorial production you will receive an email from Wiley's Author Services system, which will ask you to log in and will present you with the publication license form for completion. Within the same system the publication fee can be paid by credit card, an invoice, pro forma invoice or purchase order can be requested.

Payment of the publication charge and the signed Open Access Agreement form must be received before the article can be published online.

PROOFS

You will receive the proofs by e-mail approximately 2 weeks after all relevant files have been sent to our Production Office. Please return them within 48 hours and if there should be any problems, please contact the production office at embopressproduction@wiley.com.

Please inform us if there is likely to be any difficulty in reaching you at the above address at that time. Failure to meet our deadlines may result in a delay of publication.

All further communications concerning your paper proofs should quote reference number EMM-2020-13466-V3 and be directed to the production office at embopressproduction@wiley.com.

Thank you,

Lise Roth, Ph.D
Scientific Editor
EMBO Molecular Medicine

Corresponding Author Name: David Gilot

Journal Submitted to: Embo molecular Medicine

Manuscript Number: EMM-2020-13466 V3